# Validation of the Sentinel-5 Precursor TROPOMI cloud data with Cloudnet, Aura OMI $O_2$-$O_2$, MODIS and Suomi-NPP VIIRS

Steven Compernolle[1], Athina Argyrouli[2,3], Ronny Lutz[3], Maarten Sneep[4],
Jean-Christopher Lambert[1], Ann Mari Fjæraa[5], Daan Hubert[1], Arno Keppens[1], Diego Loyola[3],
Ewan O'Connor[6,7], Fabian Romahn[3], Piet Stammes[4], Tijl Verhoelst[1], and Ping Wang[4]

[1]Royal Belgian Institute for Space Aeronomy (BIRA-IASB), Ringlaan 3, 1180 Uccle (Brussels), Belgium
[2]Technical University of Munich, TUM Department of Civil, Geo and Environmental Engineering, Chair of Remote Sensing Technology, Munich, Germany
[3]German Aerospace Center (DLR), Münchener Straße 20, 82234 Weßling, Germany
[4]Royal Netherlands Meteorological Institute (KNMI), Utrechtseweg 297, 3730 AE De Bilt, The Netherlands
[5]Norsk Institutt for Luftforskning (NILU), Instituttveien 18, 2007 Kjeller, Norway
[6]Finnish Meteorological Institute (FMI), Helsinki, Finland
[7]University of Reading, Whiteknights, PO Box 217, Reading, Berkshire, RG6 6AH, United Kingdom

**Correspondence:** Steven.Compernolle@aeronomie.be

**Abstract.** Accurate knowledge of cloud properties is essential to the measurement of atmospheric composition from space. In this work we assess the quality of the cloud data from three Copernicus Sentinel-5 Precursor (S5P) TROPOMI cloud products: (i) S5P OCRA/ROCINN_CAL (Optical Cloud Recognition Algorithm/Retrieval of Cloud Information using Neural Networks;Clouds-As-Layers), (ii) S5P OCRA/ROCINN_CRB (Clouds-as-Reflecting Boundaries) and

5 (iii) S5P FRESCO-S (Fast Retrieval Scheme for Clouds from Oxygen absorption bands - Sentinel). Target properties of this work are cloud top height and cloud optical thickness (OCRA/ROCINN_CAL), cloud height (OCRA/ROCINN_CRB and FRESCO-S) and radiometric cloud fraction (all three algorithms). The analysis combines: (i) the examination of cloud maps for artificial geographical patterns, (ii) the comparison to other satellite cloud data (MODIS, NPP-VIIRS and OMI $O_2$-$O_2$), and (iii) ground-based validation with respect to correlative observations (2018-04-30 to 2020-

10 02-27) from the Cloudnet network of ceilometers, lidars and radars. Zonal mean latitudinal variation of S5P cloud properties are similar to that of other satellite data. S5P OCRA/ROCINN_CAL agrees well with NPP VIIRS cloud top height and cloud optical thickness, and with Cloudnet cloud top height, especially for the low (mostly liquid) clouds. For the high clouds, S5P OCRA/ROCINN_CAL cloud top height is below the cloud top height of VIIRS and of Cloudnet, while its cloud optical thickness is higher than that of VIIRS. S5P OCRA/ROCINN_CRB and S5P

FRESCO cloud height are well below the Cloudnet cloud mean height for the low clouds, but match on an average better with the Cloudnet cloud mean height for the higher clouds. As opposed to S5P OCRA/ROCINN_CRB and S5P FRESCO, S5P OCRA/ROCINN_CAL is well able to match the lowest CTH mode of the Cloudnet observations.

Peculiar geographical patterns are identified in the cloud products, and will be mitigated in future releases of the cloud data products.

## Contents

## 1 Introduction

Since decades the global distribution of atmospheric constituents has been monitored by ultraviolet/visible/near-infrared (UV/VIS/NIR) spectrometers measuring at the nadir of a satellite the radiance scattered by the Earth's atmosphere and reflected by its surface. The multi-channel UV Backscatter instrument BUV started the monitoring of the ozone column and profile in 1970-1976, continued since 1978 with the SBUV(/2) series (McPeters et al., 2013), and further extended nowadays with the OMPS-nadir series aboard the Suomi-NPP and JPSS platforms. In the late 1980s first maps of tropospheric ozone were derived from UV satellite measurements of the total ozone column (Fishman et al., 1990). In 1995, the first UV/VIS/NIR hyperspectral spectrometer in space, ERS-2 GOME (Burrows et al., 1999), paved the way to satellite observations of other species besides ozone: e.g., nitrogen dioxide ($NO_2$), bromine monoxide (BrO), formaldehyde (HCHO), glyoxal (CHOCHO), sulfur dioxide ($SO_2$), water ($H_2O$). In 2002-2012 Envisat SCIAMACHY (Bovensmann et al., 1999) added to the GOME capabilities short-wave infrared (SWIR) channels enabling the detection of methane ($CH_4$), carbon monoxide (CO), and carbon dioxide ($CO_2$). Since, the GOME and SCIAMACHY UV/VIS/NIR data records have been extended by Aura OMI (Levelt et al., 2018) and by three GOME-2 instruments aboard EPS/MetOp-A/B/C meteorological platforms. In the framework of the EU Earth Observation programme Copernicus (Ingmann et al., 2012) they will be further extended beyond horizon 2040 by the Sentinel-4, Sentinel-5 and CO2M missions, with enhanced capabilities like unprecedented spatial resolution. As a gap filler between heritage satellites and the Sentinel-5 series, Sentinel-5 Precursor (S5P) was launched in October 2017 with the TROPOspheric Monitoring Instrument (TROPOMI, Veefkind et al. (2012)) aboard. Since April 2018 this UV/VIS/NIR/SWIR hyperspectral imaging spectrometer provides daily, high-resolution, global measurements of atmospheric species (www.tropomi.eu) related to air quality ($NO_2$, $SO_2$, CO, tropospheric $O_3$, aerosols), ozone depletion, climate change, UV radiation, and volcanic hazards to aviation.

Atmospheric composition measurements from space can be affected by the presence of clouds. Clouds can not only mask underlying parts of the atmosphere, but they can also modify the radiative transfer of sunlight within and around the field-of-view of the instrument and increase the sensitivity to atmospheric constituents above and between clouds (e.g., Wang et al. (2008)). Therefore, all atmospheric composition data processors include a treatment of cloud interferences and S5P is no exception. The effect of clouds on atmospheric constituent retrievals depends mainly on the effective fractional cloud coverage of the field of view (or cloud fraction) and the cloud top height, but other parameters play a role like the cloud optical thickness, the albedo, their altitude distribution and their horizontal

patterns. Since GOME, all UV/VIS nadir sounders with an exception for OMI include measurements of the oxygen A band around 760 nm, from which two independent cloud parameters can be retrieved (Schuessler et al., 2014) - in addition to cloud height, either cloud fraction (Stammes et al., 2008) or cloud optical thickness (Loyola et al., 2010). Additional parameters like the cloud fraction (when not derived from the $O_2$ A band observations) can be retrieved

from UV spectral measurements (van Diedenhoven et al., 2007) or from broadband polarization monitoring devices (Loyola, 1998; Lutz et al., 2016; Grzegorski et al., 2006; Sihler et al., 2020). Its spectral range being limited to 500 nm, the effective cloud fraction and effective cloud pressure for OMI are retrieved using a DOAS (Differential Optical Absorption Spectroscopy) fit of the $O_2$-$O_2$ absorption feature around 477 nm (Acarreta et al., 2004; Veefkind et al., 2016).

The OCRA/ROCINN algorithms have a long-standing history and have already been applied to a set of operational instruments starting with GOME on ERS-2 (Loyola et al., 2010). A continuous development and the flexibility of OCRA/ROCINN allowed their easy adaptation to subsequent missions like SCIAMACHY on Envisat (Loyola, 2004) and the GOME-2 instruments on-board MetOp-A/B/C (Lutz et al., 2016). Recently, the algorithms have also been adapted to the EPIC instrument onboard the DSCOVR satellite, which is located at the Lagrangian point L1

(Molina García et al., 2018). Now being operational for TROPOMI on Sentinel-5 Precursor (Loyola et al., 2018), the OCRA/ROCINN cloud retrieval scheme will also be used operationally for the upcoming UVN instrument on Sentinel-4, the first mission for a geostationary view of air quality over Europe.

FRESCO (Fast Retrieval Scheme for Clouds from the $O_2$ A-band) is a fast algorithm to retrieve cloud fraction and cloud height by fitting the spectral reflectance inside and outside the $O_2$ A-band at 760 nm by a Lambertian

cloud model. The FRESCO retrieval method has been applied to GOME, SCIAMACHY, GOME-2 and TROPOMI. The method with its refinements over the years, like inclusion of Rayleigh scattering and directional surface albedo, has been described by Koelemeijer et al. (2001); Wang et al. (2008, 2016); Tilstra et al. (2020). FRESCO data are mainly used to correct for the cloud effect in trace gas retrievals, and to filter clouds in trace gas and aerosol retrievals. FRESCO-S (FRESCO for Sentinel) has been implemented in the L2 processor of TROPOMI as a support product for

KNMI and SRON level 2 products.

We note that applications of the S5P cloud data are not limited to atmospheric composition measurements. As demonstrated by Loyola et al. (2010) for GOME, OCRA/ROCINN can be successfully applied to study global and seasonal patterns and trends of cloud amount, cloud-top height, cloud-top albedo and cloud type, and compares well with the multisatellite international satellite cloud climatology project (ISCCP) D-series cloud climatology. While

developed primarily for cloud correction of trace gas retrievals, a secondary goal of S5P FRESCO is the determination of long-term cloud height trends by adding to the $O_2$ A-band observations that started with the measurements by GOME in 1995. The advantage over thermal infrared cloud height measurements is its independence of temperature.

The OCRA/ROCINN_CAL, S5P OCRA/ROCINN_CRB and S5P FRESCO cloud properties are input to several other S5P products: total and tropospheric ozone column, ozone profile, stratospheric, tropospheric and total $NO_2$

column, tropospheric HCHO column, total $SO_2$ column, aerosol layer height and $CH_4$ column (Fig. 1). Hence, given

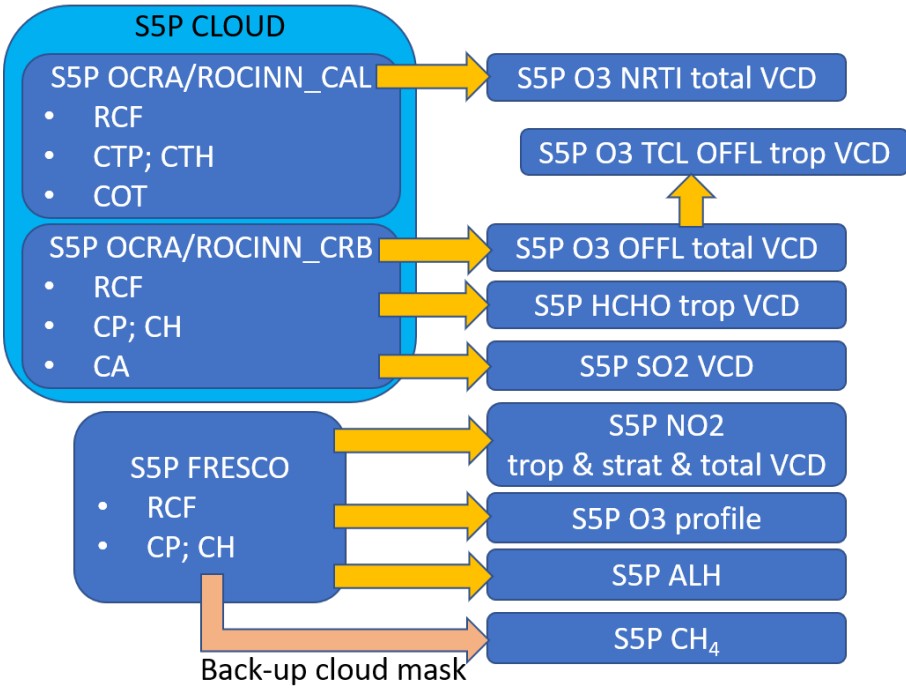

**Figure 1.** Flow chart indicating which S5P products use the cloud properties from S5P OCRA/ROCINN_CAL, S5P OCRA/ROCINN_CRB and S5P FRESCO. Note that S5P OCRA/ROCINN_CAL and S5P OCRA/ROCINN_CRB cloud properties are contained in the same S5P CLOUD product files. At the time of submission of this work, S5P $O_3$ profile is not yet operational. Note that regarding to S5P $CH_4$, NPP-VIIRS is used as the main cloud mask, while S5P FRESCO is merely a backup.

the central role of the S5P cloud products, their validation is key. In this work, a comprehensive validation is performed using ground-based data from Cloudnet as well as cloud data from other instruments: NPP-VIIRS, OMI and MODIS.

Section 2.1 gives an overview of the different cloud data products and the cloud properties discussed in this work, and establishes terminology. In Sections 2.2 and 2.3 the different satellite and ground-based data sets are described in more detail. Notes on previous assessments of the cloud algorithms and on intercomparability of cloud parameters are provided in Sections 2.4 and 2.5 respectively. Section 3 discusses briefly the S5P mission requirements for the cloud data. The latitudinal variation of zonal means of cloud fraction and of cloud height of different satellite cloud products is compared in Sect. 4.1, while the across-track dependence is studied in 4.2. Section 4.3 compares specifically cloud top height and cloud optical thickness of S5P OCRA/ROCINN_CAL with those of NPP VIIRS (not the official production release but a prototype one). Cloud height of the S5P products is compared with ground-based Cloudnet data in Section 4.4; here also a link is made to the OMI OMCLDO2 vs. Cloudnet comparison (Veefkind et al., 2016). Section 5 discusses peculiar geographical patterns that can occur in S5P OCRA/ROCINN version 1 and S5P FRESCO

**Table 1.** Properties, abbreviation and mathematical symbol.

| Parameter | abbreviation | mathematical symbol |
|---|---|---|
| geometrical cloud fraction[a] | GCF | $f_{gc}$ |
| radiometric cloud fraction | RCF | $f_{rc}$ |
| RCF, scaled to a fixed cloud albedo $a = 0.8$ | sRCF[b] | $f_{rc,0.8}$ |
| cloud top height & pressure | CTH, CTP | $h_{ct}, p_{ct}$ |
| cloud height & pressure[c] | CH, CP | $h_c, p_c$ |
| cloud optical thickness | COT | $\tau_c$ |
| cloud albedo | CA | $A_c$ |
| surface albedo | SA | $A_s$ |
| cloud mean height[d] | CMH | $h_{cm}$ |

a. GCF is not derived from S5P measurements.

b. The relation between sRCF and RCF is $f_{rc,a} = f_{rc} A_c / a$. In this work always $a = 0.8$ is taken.

c. These refer to the position of the optical centroid. See e.g., Stammes et al. (2008). This depends on the optical thickness of the cloud.

d. Calculated as the mean of the positions of the cloudy altitude bins in a vertical cloud profile. This does not depend on cloud optical thickness.

version 1.3, and how these are improved in recently released (but not yet reprocessed) upgraded versions. Conclusions are given in Sect. 6.

## 2  Description of the data sets

### 2.1  Overview of cloud data products, properties, and related terminology

In this work several cloud data products and cloud properties are discussed; here we provide an overview and terminology conventions. Table 1 contains an overview of properties discussed in this work (either as subject for validation or as important influence quantity) and the corresponding abbreviation and mathematical symbol. Table 2 contains an overview of cloud data products and main cloud properties.

The S5P cloud products we validate here (S5P OCRA/ROCINN, S5P FRESCO) provide a radiometric cloud fraction[1] (RCF; $f_{rc}$). Note that a RCF is related to, but different from, a geometrical cloud fraction (GCF; $f_{gc}$) as provided by e.g., NPP/VIIRS and MODIS. The RCF is not the geometric cloud fraction of the true cloud, but can be defined as the fraction that has to be attributed to the model cloud to yield (in combination with non-cloud contributions) a top-of-atmosphere (TOA) reflectance that agrees with the observed reflectance. In OCRA, the clear-sky (RCF=0) reflectance is taken from composite maps created from satellite measured reflectances and the fully cloudy (RCF=1) reflectance is defined as 'white' in the color diagram. OCRA then determines the radiometric cloud fraction using the

---

[1]The term 'effective cloud fraction' is sometimes also used in the literature (e.g., Stammes et al., 2008). Note that radiometric cloud fraction has to be clearly distinguished from the 'cloud radiance fraction' found in e.g., the S5P $NO_2$ data product and which is a different quantity.

**Table 2.** Overview of cloud products, algorithms and main properties discussed in this work. The property abbreviations are explained in Table 1.

| Product | Platform\sensor | Algorithm | Property | Ref |
|---|---|---|---|---|
| S5P OCRA/ROCINN_CAL[a] | S5P\TROPOMI | OCRA | RCF | Loyola et al. (2018) |
| | | ROCINN_CAL | CTH, CTP, COT[b] | |
| S5P OCRA/ROCINN_CRB[a] | S5P\TROPOMI | OCRA | RCF | Loyola et al. (2018) |
| | | ROCINN_CRB | CH, CP, CA | |
| S5P FRESCO | S5P\TROPOMI | FRESCO-S | sRCF[c], CP, CA | KNMI (2019) |
| VIIRS[d] | SNPP\VIIRS | | GCF, CTP, COT | |
| MODIS MYD08_D3[e] | Aqua\MODIS | | GCF, CTP | Platnick et al. (2017) |
| OMCLDO2 | Aura\OMI | OMCLDO2 | sRCF[f], CH[g], CP | Veefkind et al. (2016) |
| Cloudnet | ground-based | | CTH, CMH[h] | Illingworth et al. (2007) |

a. S5P OCRA/ROCINN_CAL and S5P OCRA/ROCINN_CRB subproducts are within the same S5P CLOUD product files.

b. Before comparing with VIIRS COT, the S5P OCRA/ROCINN_CAL is first converted to an *effective* COT using RCF×COT. More detail is provided in Sect. 4.3.

c. In S5P FRESCO, for most pixels, CA is fixed at 0.8. However, CA>0.8 is allowed to avoid cloud fractions larger than 1, so this is not strictly a sRCF. When doing actual comparisons between S5P FRESCO and other products, we therefore first convert to a strict sRCF with CA fixed at 0.8.

d. A prototype was used in the comparison and not the official (Platnick et al., 2017) VIIRS product.

e. Daily gridded L3 product, based on the L2 MYD06 product.

f. sRCF with CA fixed at 0.8.

g. CH is not provided as such in the OMCLDO2 product. It is calculated here using the OMCLDO2 CP and a scale height of 7668 m, (see Eq. (2)).

h. CMH is not provided as such in the Cloudnet product. We calculate it here considering classification labels 1-7 as cloudy grid cells and labels 0 and 8-10 as non-cloudy, following Veefkind et al. (2016).

differences between the reflectance (defined as colors in OCRA) of a measured scene and its corresponding clear-sky values. In FRESCO, the radiometric cloud fraction is the cloud fraction value which, in combination with the assumed cloud albedo (CA) and the input surface albedo, yields a TOA reflectance that agrees with the observed reflectance. In most cases one has RCF ≤ GCF; an example is a scene that is fully cloud covered ($f_{gc} = 1$) with an optically thin cloud ($f_{rc} < 1$) (Stammes et al., 2008). Note that the GCF, as opposed to the RCF, does not depend on cloud optical thickness (COT).

As the cloud models of S5P OCRA/ROCINN and S5P FRESCO differ, their RCFs are not directly comparable. Therefore, we scale the RCF to the corresponding cloud fraction of a Lambertian reflector with fixed CA equal to 0.8 (sRCF; $f_{rc,0.8}$). This is explained in more detail in Sections 2.2.1 and 2.2.2. Note that the OMI OMCLDO2 product already assumes a CA of 0.8 (Sect. 2.2.5).

S5P OCRA/ROCINN_CRB, S5P FRESCO and OMCLDO2 all model a cloud as a Lambertian reflector. The retrieved cloud height (CH; $h_c$) pertains to the optical centroid of the cloud rather than to the cloud top. On the other hand, S5P OCRA/ROCINN_CAL, SNPP\VIIRS and Aqua\MODIS provide cloud top heights.

## 2.2   Satellite data sets

### 2.2.1   S5P TROPOMI CLOUD OCRA/ROCINN

Here we provide technical information on the S5P CLOUD OCRA/ROCINN product. For more detail we refer the reader to the Product Readme File (PRF), Product User Manual (PUM) and Algorithm Theoretical Basis Document (ATBD), all available at https://sentinels.copernicus.eu/web/sentinel/technical-guides/sentinel-5p/products-algorithms.

    **Versioning and dissemination.** The S5P CLOUD product is one of the S5P UPAS products, other S5P UPAS products being HCHO, $SO_2$, $O_3$ (total column) and $O_3\_TCL$ (tropospheric column). It is available from the Pre-Operations data hub. All UPAS products share a common processor version numbering. As is the case for most S5P data products, the nominal operational processing produces a near-real time (indicated with 'NRTI' in the file name) and an offline ('OFFL' in the file name) data product, with the UPAS processor release that is active at the time. There have been a few reprocessing campaigns to produce a consistent data-set. The resulting reprocessed ('RPRO' in the file name) files can be combined with OFFL data for longer time series. By combining RPRO and OFFL data, a consistent version 1 CLOUD data record (with processor version numbers 1.1.7-1.1.8; note that for CLOUD there were no changes between 1.1.7 and 1.1.8.) is available from 2018-04-30 up to 2020-07-12, after which version 2 was introduced.

    Note that for processor version 1, NRTI CLOUD uses the same processor as RPRO and OFFL, and therefore has nearly the same output (for the same processor version). For version 2 there are algorithmic differences between NRTI and OFFL data.

    Below, we first describe processor version 1 (version number up to 1.1.8), and then the main changes introduced with version 2 (version number starting with 2.1.3). Version 1, for which a +2-year record is available, is the target of the bulk of the analysis in this work. There has been no version 2 reprocessing to date; this data record starts at 2020-07-13 for OFFL, and at 2020-07-16 for NRTI. Some of the main impacts of the processor version upgrade are described in Sect. 5. A full quality assessment of version 2 is out of scope of the current work.

    **Version 1.** The S5P CLOUD OCRA/ROCINN retrieval (Loyola et al., 2018) is a two-step algorithm where the OCRA (Optical Cloud Recognition Algorithm) (Loyola, 1998; Lutz et al., 2016) computes the RCF using a broad-band ultraviolet/visible (UV/VIS) color space approach and ROCINN (Retrieval of Cloud Information using Neural Networks) retrieves the CTH, CH, COT and CA from near-infrared (NIR) measurements in and around the oxygen A-band (~760nm).

    OCRA derives the RCF from UV–VIS reflectances by separating the sensor measurements into two components: a cloud-free background and a remainder expressing the influence of clouds. A color-space approach is used, where broad-band UV-VIS reflectances are translated to blue and green colors. The underlying assumption is that clouds appear white in the color-space, meaning that the spectrum of a cloud is wavelength independent across the UV-VIS wavelength range. The actual radiometric cloud fraction is then determined as the distance between the fully cloudy "white" color and the clear-sky colors taken from the reflectance background composite maps. For version 1 of the

algorithm, the cloud-free background is based on three years of OMI data and consists of global monthly composite maps per color with a spatial resolution of 0.2 x 0.4 degrees. This relatively coarse and asymmetric spatial grid choice is due to the relatively large and asymmetric (especially near the swath edge) OMI pixels. Thanks to the monthly temporal resolution, seasonal changes can be covered. For each given day a linear interpolation between two adjacent monthly maps is used. In a pre-processing step, scan angle dependencies of the colors are addressed by fitting low-order polynomials to monthly mean reflectance data as a function of color, time, across-track pixel position (i.e. viewing zenith angle) and latitude. Instrumental degradation is currently not addressed in OCRA itself since the updated L1b data will themselves include a degradation correction.

ROCINN is a machine learning algorithm for retrieving two additional cloud parameters from the measured NIR radiances around the $O_2$ A-band; the fitting window covers the full spectral range from 758 to 771 nm. The forward problem refers to the simulation of sun-normalized radiances for different cloud configurations using the VLIDORT Radiative Transfer Model (RTM) (Spurr, 2006). A significant set (~200000 samples) of simulated radiances, which satisfies the conditions of the smart sampling (Loyola et al., 2016), is used for the training of the operational Neural Network (NN). The replacement of the exact RTM by a NN, which is a well tested approximation for complex operational algorithms like ROCINN, is in particular beneficial for gaining computational efficiency. The CH/CTH and CA/COT are the cloud parameters which can be retrieved simultaneously using the Tikhonov regularization technique from two independent pieces of information (Schuessler et al., 2014). Note that during the inversion a wavelength shift for the earthshine spectrum is fitted additionally (Loyola et al., 2018). Two cloud models are handled in ROCINN: (i) Clouds-as-Reflecting-Boundaries (CRB) which considers the cloud as a Lambertian reflector and (ii) Clouds-as-Layers (CAL) which considers the cloud as a homogeneous cluster of scattering liquid water spherical particles using Mie theory. ROCINN_CRB retrieves an effective cloud height and a cloud albedo, while ROCINN_CAL retrieves a cloud top height and a cloud optical thickness. The CAL cloud base height is not a retrieved quantity but it is fixed by assuming a constant cloud geometrical thickness of 1 km. In version 1, other complementary information about the surface properties have been initially estimated from the MERIS monthly climatology ($0.25°$x$0.25°$ spatial resolution).

Note that both ROCINN_CAL and ROCINN_CRB re-retrieve the RCF with the OCRA RCF as a priori. This is done with a strong regularization such that the values do not differ much from the OCRA RCF. After the ROCINN retrieval, cloud (top) pressure is obtained from the retrieved cloud (top) height using ECMWF profiles.

There is no separate treatment for snow/ice pixels, but it is known that cloud retrieval is more challenging in these conditions.

To summarize, there are two cloud products stored in the S5P CLOUD data files: S5P OCRA/ROCINN_CAL (providing RCF $f_{rc}$, CTH $h_{tc}$ and COT $\tau_c$) and S5P OCRA/ROCINN_CRB (providing RCF, CH $h_c$ and CA $A_c$) (Fig. 1 and Table 2).

To be able to do RCF comparisons with S5P FRESCO and OMCLDO2, we first convert the S5P OCRA/ROCINN_CRB RCF to an sRCF with a cloud albedo fixed at 0.8. In ROCINN_CRB, the cloud albedo $A_c$ is attributed a fill value when

the RCF $f_{rc} = 0$. Therefore, the conversion is done as follows

$$f_{rc,0.8} = f_{rc} * A_c/0.8, \text{ if } f_{rc} > 0 \tag{1}$$
$$f_{rc,0.8} = 0, \text{ if } f_{rc} = 0$$

The last line of Eq. (1) is needed to prevent cases where $f_{rc} = 0$ and $A_c = \text{NaN}$ would lead to $f_{rc,0.8} = \text{NaN}$.
Note that when both RCF and CA reach unity, sRCF reaches 1.25 rather than 1. This conversion is not possible for
ROCINN_CAL which does not provide a CA, but as the RCFs of ROCINN_CAL and ROCINN_CRB are both close
to the OCRA RCF anyway, a separate evaluation is deemed unnecessary.

**Changes in version 2.** Two of the more major changes in version 2 are the following. In the new S5P CLOUD
version 2, the OMI-based cloud-free background maps have been replaced by maps based on TROPOMI data and,
thanks to the better spatial resolution of this instrument, the maps could be refined to 0.1 x 0.1 degrees, while keeping
the monthly temporal resolution. Furthermore, the surface properties are no longer based on a monthly climatology.
Instead, the geometry-dependent surface properties are retrieved directly from TROPOMI measurements within the
ROCINN fitting window using the GE_LER (geometry-dependent effective Lambertian equivalent reflectivity) algo-
rithm (Loyola et al., 2020), daily dynamically updated on a $0.1°$ x $0.1°$ grid. For a full overview of the changes in
version 2, we refer the reader to the PRF and the ATBD.

### 2.2.2 S5P TROPOMI FRESCO-S

Here we provide technical information on the S5P FRESCO-S support product. FRESCO-S specific information can
be found in the S5P $NO_2$ ATBD (KNMI, 2019) and in the S5 CLOUD ATBD (KNMI, 2018). Information about earlier
FRESCO algorithms can be found in Koelemeijer et al. (2001); Wang and Stammes (2014).

**Versioning and dissemination.** S5P FRESCO is one of the products generated by the S5P NL-L2 processor. Other
S5P NL-L2 products are CO, $CH_4$, $NO_2$, AER_AI (aerosol absorbing index) and AER_LH (aerosol layer height).
Note that S5P FRESCO is a support product and its data files are not publicly released, but its cloud parameters can
be accessed via the ALH or $NO_2$ data files, which are available from the pre-operations data hub. As is the case for
most S5P data products, the nominal operational processing produces an NRTI and an OFFL data product, with the
NL-L2 processor release that is active at the time. Reprocessing is applied to obtain a consistent data record. The
main focus of our analysis is the RPRO+OFFL 1.3 data record which extends from 2018-04-30 to 2020-11-29. Very
recently, version 1.4 was introduced. The corresponding OFFL data starts at 2020-11-29, but to date no reprocessing
is available.

Below, we first describe processor version 1.3 (NL-L2 version numbers 1.3.0 to 1.3.2; note that the FRESCO
version is identical for these numbers), and then the main changes introduced with version 1.4. Version 1.3, for which
a 2.5-year record is available, is the target of the bulk of the analysis in this work. As there has been no FRESCO
reprocessing after version 1.3, the 1.4 data record is still short, starting at 2020-11-29 for OFFL, and at 2020-12-02 for

NRTI. Some of the main impacts of the processor version upgrade are described in Sect. 5. A full quality assessment of version 1.4 is out of scope of the current work.

**Version 1.3.** FRESCO-S models a cloud as a Lambertian reflector, similar to S5P OCRA/ROCINN_CRB. FRESCO-S retrieves the information on cloud pressure $p_c$ and RCF $f_{rc}$ from the reflectance in and around the $O_2$ A band. FRESCO uses three about 1 nm wide wavelength windows, namely $758 - 759$ nm (continuum, no absorption), $760 - 761$ nm (strong absorption), and $765 - 766$ nm (moderate absorption). So both retrieved parameters $p_c$ and $f_{rc}$ are consistently retrieved from the same spectral region. As opposed to S5P OCRA/ROCINN_CRB, where cloud albedo is retrieved, in FRESCO-S, the cloud albedo is assumed to be fixed at 0.8 (see Koelemeijer et al., 2001; Stammes et al., 2008, for the justification), except when this assumption would lead to a cloud fraction larger than 1. In those cases the RCF is set to 1, and the cloud albedo is fitted instead, but only if the cloud height is well separated from the surface. In FRESCO, the basic retrieved quantity is cloud height (in km), which is converted to pressure using the AFGL mid-latitude summer (MLS) profile (Anderson et al., 1986).

Due to the increase in the spectral resolution in the TROPOMI instrument, the different spectral grid for each viewing direction, and small wavelength shifts introduced by inhomogeneous illumination of the spectral slit due to spatial variation of the brightness of the scene, some changes were introduced in the FRESCO-S algorithm compared to previous FRESCO versions. The spectral resolution of the reflectance database was increased to allow for interpolation of the database to the wavelengths of the observation[2]. This is in marked contrast to previous FRESCO versions (for the instruments GOME, SCIAMACHY, GOME-2), where the observed wavelengths were interpolated to the wavelengths of the database. Each viewing direction has its own reflectance database, to adjust to the different nominal wavelength grids and the variation of the instrument spectral response function.

The FRESCO-S algorithm uses a surface albedo monthly climatology based on GOME-2 (Tilstra et al., 2017). An important advantage over the MERIS black-sky albedo climatology (based on 2002-2006 data, i.e., about 15 years ago) (Popp et al., 2011) is that it is more recent. On the other hand, it is affected by the GOME-2 resolution and the solar zenith angle at overpass time. Due to the difference in overpass time between GOME-2 (in the morning) and S5P (in the afternoon), and the large discrepancy in the spatial resolution of both instruments, the surface albedo climatology is currently considered as one of the larger sources of error for the FRESCO-S algorithm. To compensate, some adjustments are made to suppress negative effective cloud fraction due to a climatological surface albedo value that is higher than reality. Also note that for scenes with a surface albedo higher than the assumed cloud albedo, the cloud parameters are less reliable. Treatment of snow/ice surfaces is described in the S5P $NO_2$ ATBD (KNMI, 2019). The version of FRESCO validated here is 1.3, with the same time range as for the S5P OCRA/ROCINN product. The FRESCO processing in all these versions is identical, as the changes only applied to other NL-L2 products.

Before comparing the S5P FRESCO RCF with those of S5P OCRA/ROCINN_CRB and OMCLDO2, we first convert them to an sRCF using RCF*CA/0.8. This only makes a difference for those pixels where CA was fitted and not fixed at 0.8. Note that the RCF of a scene viewed in a certain direction (e.g., at large viewing zenith angle and

---

[2]The database is tored with a four-fold spectral oversampling so that spline interpolation can be used for this step.

forward scattered light) can exceed unity if the reflectance of the cloud is larger than unity in that direction. This does not violate flux conservation since that holds for the average over all directions.

**Version 1.4.** From previous validation efforts we know that FRESCO retrieves a height near the optical extinction weighted mean height of the cloud, at least for scenes with a significant cloud cover. For scenes with low clouds, i.e., close to the surface, a height that is even closer to the surface will be retrieved. This also holds for low aerosol layers, since the algorithm does not discriminate between the two types of scatterers (Wang et al., 2012). In many cases FRESCO then retrieves the surface height, which is incorrect. This defect can be remedied by using a wider window with low to moderate absorption in the $O_2$ A-band. Instead of $765 - 766$ nm, a 5 nm wide window $765 - 770$ nm increases the sensitivity to low clouds. This new look-up table is used for FRESCO versions 1.4 and later.

### 2.2.3  Suomi-NPP VIIRS

The Visible/Infrared Imager/Radiometer Suite (VIIRS) is one of the five instruments onboard the Suomi National Polar-orbiting Partnership (NPP) satellite platform launched at the end of October 2011. The spectral coverage expands from the visible (VIS) to infrared (IR) with 22 channels from 0.41 µm to 12.01 µm at two different spatial resolutions of 375 m and 750 m. Five channels are high-resolution image bands (I1-5 at 375m) and sixteen are moderate-resolution bands (M1-16 at 750m). The optical/microphysical property (i.e., CLDPROP_L2_VIIRS_SNPP) cloud product refers to the pixel resolution of 750m. This Level-2 (L2) product was developed by NASA (Platnick et al., 2017) to ensure continuity for the long-term records of Moderate Resolution Imaging Spectroradiometer (MODIS) and VIIRS heritages. Note that the VIIRS data used in this work are not part of NASA VIIRS production release files and potential differences cannot be ruled out. Within the CLDPROP algorithm, the cloud top properties are derived from NOAA's operational algorithms, the so-called Clouds from AVHRR Extended (CLAVR-x) processing system, in which the algorithm is based primarily on IR spectral channels, with the additional information of shortwave infrared (SWIR) channels. In particular, the cloud top height is derived from the AWG (Algorithm Working Group) Cloud Height Algorithm (ACHA) (Heidinger and Li, 2019; Heidinger et al., 2019). Moreover, the cloud optical and microphysical property product inherits the MOD06 cloud optical/microphysical property retrieval algorithm from Platnick et al. (2017). The cloud optical thickness COT is retrieved simultaneously with the cloud effective radius (CER) based on a two channel retrieval introduced in Nakajima and King (1990). The COT information is primarily derived from the reflectance in a non-absorbing VIS, near infrared (NIR), or SWIR spectral channel which depends on the surface type. The CER information is provided by the reflectance in an absorbing SWIR or mid-wave infrared (MWIR).

### 2.2.4  Aqua MODIS

There is a Moderate Resolution Imaging Spectroradiometer (MODIS) instrument on board of both the Terra and the Aqua satellites, with Terra in descending mode passing the equator in the morning and Aqua in ascending mode passing the equator in the afternoon, respectively. MODIS has 36 spectral bands ranging in wavelengths from 0.4 $\mu$m to 14.4 $\mu$m and data products are retrieved in three different spatial resolutions of 250 m, 500 m, and 1 km. The

comparison with TROPOMI can be done for the ascending MODIS/Aqua and only in a daily basis using the Level-3 (L3) MODIS gridded atmosphere daily global joint MYD08_D3 product (Platnick et al., 2015). It contains daily 1 x 1 degree grid average values of atmospheric parameters among others also cloud properties. Cloud-top temperature, height, effective emissivity, phase and cloud fraction are produced using infrared channels with 1-km-pixel resolution and stored in the L2 MODIS cloud data product file MYD06_L2, which is one of the four L2 MODIS atmosphere products used for the L3 MODIS atmosphere daily global parameters.

### 2.2.5 Aura OMI OMCLDO2

The OMI OMCLDO2 product (Veefkind et al., 2009, 2016) is retrieved from Level-1B VIS channel from the Dutch-Finnish UV-Vis nadir viewing spectrometer OMI (Ozone Monitoring Instrument) on NASA's EOS-Aura polar satellite. The nominal footprint of the OMI ground pixels is $24 \times 13 \ \mathrm{km}^2$ (across $\times$ along track) at nadir to $165 \times 13 \mathrm{km}^2$ at the edges of the 2600 km swath, and the ascending node local time is 13:42 hrs. The OMI instrument covers the UV and visible wavelength range (270–500 nm). This means that the oxygen A-band that is used by FRESCO and ROCINN is not available in the spectral range measured by OMI, and an alternative cloud retrieval algorithm is required. For OMI a cloud retrieval algorithm was developed that uses the $O_2$–$O_2$ collision induced absorption feature at 477 nm. This is done by using a cloud model that is very similar to the model used in FRESCO (Acarreta et al., 2004; Veefkind et al., 2016). Similar to FRESCO, it fixes the cloud albedo at 0.8, retrieves a radiometric cloud fraction and cloud height, and is sensitive not only to cloud but also to aerosol. The sensitivity of the $O_2$–$O_2$ cloud retrieval algorithm differs from FRESCO, because of the different wavelength range with a generally much lower surface albedo, especially over vegetated land, and a reduced sensitivity for (very) high clouds due to the reduced absorption at low pressures due to the density-squared nature of the absorption feature itself (Acarreta et al., 2004). Otherwise, both FRESCO and $O_2$–$O_2$ Cloud are expected to retrieve a height around the mid-level of the cloud (Sneep et al., 2008; Stammes et al., 2008).

The OMCLDO2 data product contains a cloud pressure, but not a cloud height. Therefore, the cloud pressure is converted to a cloud height using a scale height of $h_{\mathrm{scale}} =$7668 m, (see Eq. (2))

$$h_c^{\mathrm{OMCLDO2}} = -h_{\mathrm{scale}} \ln\left(\frac{p_c}{p_s}\right) + h_s \tag{2}$$

with $p_s$ the surface pressure and $h_s$ the surface altitude of the OMCLDO2 pixel. The value of 7668 m was obtained by fitting to the AFGL Mid latitude summer (MLS) profile (Anderson et al., 1986), which is used as reference profile in the FRESCO algorithm.

Following Veefkind et al. (2016) we include in Sect. 4.4 a comparison of OMCLDO2 with Cloudnet data, to judge how this is different from the S5P comparisons with Cloudnet, using the same comparison settings.

## 2.3 Ground-based data sets: Cloudnet

Europe operates a network of ground-based cloud-profiling active remote sensing stations as part of the Aerosol, Clouds and Trace Gas Infrastructure Network (ACTRIS). These stations operate vertically-pointing cloud radar and lidar/ceilometer and use the Cloudnet processing scheme (Illingworth et al., 2007) for the continuous evaluation of cloud profile properties. The Cloudnet scheme combines the cloud radar and lidar measurements at a temporal resolution of 30 s and a vertical resolution of 30 m to create a target categorization product which diagnoses the presence or absence in each altitude bin of: aerosol, insects, drizzle, rain, liquid cloud droplets, supercooled liquid droplets, ice cloud particles, melting ice cloud particles. Note that multiple targets can be diagnosed within a single altitude bin. The Cloudnet L2 classification product then takes the target categorization product and simplifies the possible combinations into 11 main atmospheric target classifications at the same resolution (30 s and 30 m). From the classifications also a cloud base height and cloud top height are derived and stored in the product. The Cloudnet processing scheme was also applied to similar cloud-profiling measurements from the US Department of Energy Atmospheric Radiation Measurement (ARM) sites. The sites included in this validation dataset are provided in Table 3 and displayed in Fig. 2. Cloudnet products are freely available for download from the Cloudnet database (http://cloudnet.fmi.fi/).

The physical horizontal extent of the cloud radar measurements is on the order of 20 m at 1 km altitude, and 200 m at 10 km altitude; for the lidar, the physical horizontal extent of the measurements is about an order of magnitude smaller. Horizontal advection of clouds by the wind during the 30-s averaging time implies an effective horizontal extent that is usually larger than the physical horizontal extent; for example, a $30 \mathrm{~m\,s}^{-1}$ wind at 10 km yields an effective horizontal extent of 900 m for both instruments.

We use the CTH provided by Cloudnet directly in our validation work. Furthermore, following Veefkind et al. (2016), we convert Cloudnets classification to a vertical cloud profile, by considering altitude bins with target classification type 1-7 as cloudy, and bins with the remaining classification types (0 and 8-10) as cloud-free. The cloud mean height (CMH; $h_{cm}$) is then calculated as the mean of the vertical positions of the cloudy altitude bins.

## 2.4 Note on previous assessments of OCRA/ROCINN and FRESCO algorithms

Here follows a brief description of previous assessments of the S5P TROPOMI cloud product algorithms, mainly available from Sect. 3 of the S5P Science Verification Report (S5P-SVP Richter and the Verification Team, 2015). It must be noted however that in this work pre-launch versions of the algorithms were tested and that the current algorithms underwent significant changes since then.

Pre-launch versions of the S5P FRESCO and S5P OCRA RCF algorithms were compared to the MICRU algorithm using GOME-2 data (see the S5P-SVP Richter and the Verification Team, 2015, section 13.3), (Sihler et al., 2020). This study includes the across-track dependence of the difference between OCRA and MICRU.

The operational S5P cloud retrieval and trace gas retrieval algorithms are all based on 1D radiative transfer. The neglect of three-dimensional radiative transfer (3D-RT) effects becomes relevant due to the small ground pixel size

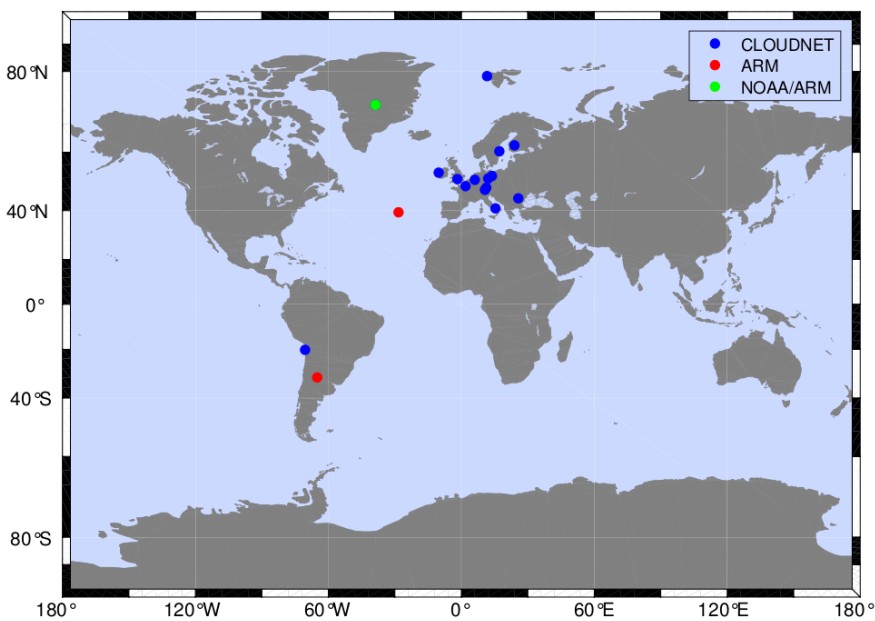

**Figure 2.** Selection of Cloudnet and ARM sites considered in this work.

**Table 3.** Selection of Cloudnet and ARM sites considered in this work.

| Station | lat[°],lon[°] | Location | Network |
|---|---|---|---|
| Ny-Alesund | 78.93, 11.92 | Svalbard | Cloudnet |
| Summit | 72.60, -38.42 | Greenland | NOAA/ARM |
| Hyytiala | 61.84, 24.29 | Finland | Cloudnet |
| Norunda | 60.85, 17.48 | Sweden | Cloudnet |
| Mace Head | 53.33, -9.90 | Ireland | Cloudnet |
| Lindenberg | 52.21, 14.13 | Germany | Cloudnet |
| Leipzig | 51.35,12.43 | Germany | Cloudnet |
| Chilbolton | 51.14,-1.44 | United Kingdom | Cloudnet |
| Juelich | 50.91, 6.41 | Germany | Cloudnet |
| Palaiseau | 48.71, 2.21 | France | Cloudnet |
| Munich | 48.15, 11.57 | Germany | Cloudnet |
| Schneefernerhaus | 47.42, 10.98 | Germany | Cloudnet |
| Bucharest | 44.35, 26.03 | Romania | Cloudnet |
| Potenza | 40.60, 15.72 | Italy | Cloudnet |
| Graciosa | 39.09, -28.03 | Azores | ARM |
| Iquique | -20.54, -70.18 | Chile | Cloudnet |
| Villa Yacanto | -32.13, -64.73 | Argentina | ARM |

of TROPOMI. The impact of cloud shadow was simulated in the S5P-SVP (Richter and the Verification Team, 2015, section 13.3.3). Other 3D-RT effects, for vertically extended clouds, are the dependencies of observed cloud fraction (Minnis, 1989) and of COT (Liang and Girolamo, 2013) on viewing zenith angle (VZA).

From the results of Joiner et al. (2010), it can be assumed that on global average ~10% of the TROPOMI pixels contain multilayer clouds. Furthermore, the assumption of a homogeneous cloud field "is never valid" (Rozanov and Kokhanovsky, 2004). However, such information cannot be obtained from single view observations[3] of the $O_2$ A-band and is therefore necessarily neglected in the S5P cloud algorithms. Loyola et al. (2018) present simulated OCRA/ROCINN_CAL and CRB retrievals for double-layer scenarios.

As shown in the S5P-SVP (Richter and the Verification Team, 2015, section 13.4.2.3), cloud height comparisons between ROCINN_CAL or FRESCO with SACURA (Rozanov and Kokhanovsky, 2004) show larger disagreement at scenes with a low RCF and a higher surface albedo, indicating a larger uncertainty in these conditions. In agreement with this, simulations with FRESCO (Wang and Stammes, 2014) have shown that for optically thick clouds, the cloud height is near the optical midlevel, while for optically thin clouds and higher surface albedo, a FRESCO cloud height above the cloud can be found.

## 2.5 Note on intercomparability of cloud properties

An issue when comparing cloud properties from different cloud products is that they are often not exactly comparable, for example because they do not exactly represent the same quantity, or because the sensitivities are at a different wavelength range. Below we give a short overview.

**Geometrical CF vs radiometric CF.** See also Sect. 2.1. The GCF usually exceeds the RCF. Furthermore, the GCF is independent of the cloud optical thickness while the RCF can be related to COT. E.g., if the OCRA RCF over- or underestimates the GCF in individual cases strongly depends on the cloud optical thickness. Finally, the UV/VIS spectrometer data from TROPOMI are usually less sensitive to optically very thin clouds, which might be easier detectable with imager data like VIIRS and MODIS that also include bands in the infra-red.

**Radiometric CF.** The RCF of OCRA, FRESCO and OMCLDO2 are based on different model assumptions and/or use different wavelength ranges. To make these quantities more comparable the sRCF was introduced (see Eq. (1)).

**Cloud (top) height.** The different products use different wavelength ranges and/or are based on different models. E.g., Lambertian reflector in the case of ROCINN_CRB, FRESCO, and OMCLDO2, Mie theory in the case of ROCINN_CAL. Furthermore, Cloudnet's radar-based CTH will have a higher sensitivity to optically thin ice clouds than ROCINN_CAL's CTH. The Cloudnet CMH, which is compared to ROCINN_CRB CH and FRESCO CH in this work, does not take into account the optical thickness of the layers.

**Cloud optical thickness.** VIIRS COT is independent of the GCF, while the ROCINN_CAL COT is inversely related to the RCF. ROCINN_CAL RCF x COT is therefore compared with the VIIRS COT (more detail in Sect. 4.3).

---

[3]Multilayer information can be obtained in combination with another sensor like MODIS (Joiner et al., 2010) or from multidirectional $O_2$ A-band observations (Desmons et al., 2017).

## 3 Mission requirements

The mission requirements applicable to the cloud data product from the atmospheric composition Sentinels were first stated in ESA (2017a, b). Adapting the terminology[4] to be compliant with the international metrology standards VIM (International vocabulary of metrology) (JCGM, 2012) and GUM (Guide to the expression of uncertainty in measurement) (JCGM, 2008); these are as follows: (i) the bias on cloud fraction, cloud height and cloud optical thickness may not exceed 20% and (ii) the uncertainty requirement is 0.05 for cloud fraction, 0.5 km for cloud height, and 10 for cloud optical thickness. We understand here that cloud fraction refers to the RCF (possibly scaled with a fixed cloud albedo), while cloud height can refer to both the cloud height at the optical centroid (as provided by S5P OCRA/ROCINN_CRB and S5P FRESCO) or the cloud top height (as provided by S5P OCRA/ROCINN_CAL). Since the beginning of its nominal operation in April 2018, in-flight compliance of S5P TROPOMI with these mission requirements has been monitored routinely by means of comparisons to ground-based reference measurements in the Validation Data Analysis Facility (VDAF) of the S5P Mission Performance Centre (MPC) and by confrontation with satellite data from MODIS, VIIRS and OMI.

Mission requirements relate to deviations of the satellite data from an (unknown) true value. But in comparisons with real-life reference data, deviations occur also due to imperfect reference measurements and moreover, because of different temporal/spatial/vertical sampling and smoothing properties (Loew et al., 2017). Frameworks and terminology related to comparisons are developed in Lambert et al. (2013); Verhoelst et al. (2015); Verhoelst and Lambert (2016); Keppens et al. (2019).

It should be noted that single numbers as requirement are necessarily a simplification. The impact of cloud parameter errors depends on the cloud height and on the application (e.g., cloud correction of trace gases which may or may not be well-mixed, cloud slicing to obtain tropospheric ozone, ...).

## 4 Results

### 4.1 Comparison of zonal means between cloud products

In this section zonal mean comparisons are presented for the different cloud products. Figure 3 presents comparisons at one day (2018-04-28) between S5P OCRA/ROCINN_CAL (processor version 1) and VIIRS of RCF and GCF (left panel), CTH (middle panel) and COT (right panel).

S5P OCRA/ROCINN_CAL RCF and VIIRS GCF show a similar latitudinal variation, but, as expected, the geometrical cloud fraction is higher than the S5P OCRA radiometric cloud fraction (Loyola et al., 2010). While the CTH variation is similar, variations are stronger for VIIRS. Finally, COT latitudinal variations are similar for S5P

---

[4]In the ESA documentation 'bias' and 'random error' is used. The term 'random error' is not retained here as several components contribute to the uncertainty that are not random. Here we use the VIM/GUM terms bias (estimate of a systematic error) and uncertainty (non-negative parameter that characterizes the dispersion of the quantity values).

OCRA/ROCINN_CAL and VIIRS, but with an offset (S5P OCRA/ROCINN_CAL higher than VIIRS). Further detail about the comparison between S5P OCRA/ROCINN_CAL and VIIRS is provided in Sect. 4.3.

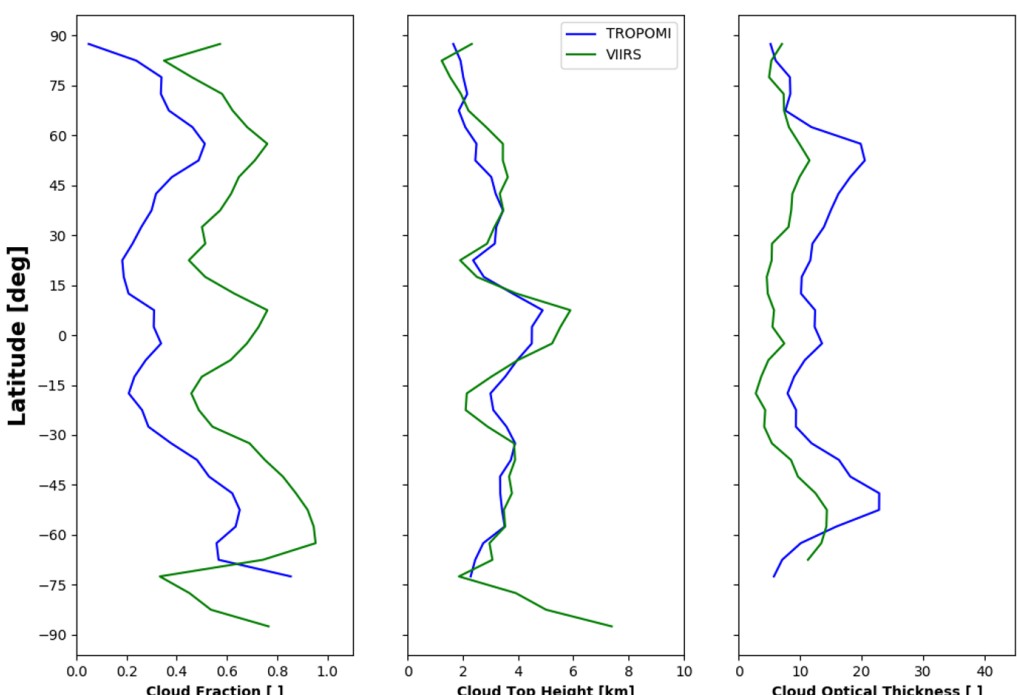

**Figure 3.** Zonal means for S5P OCRA/ROCINN_CAL processor version 1 (blue) and NASA VIIRS (green). The comparison refers to data from 28th April 2018 (VIIRS cloud fraction is a geometrical cloud fraction whereas the S5P OCRA cloud fraction is a radiometric one).

Figure 4, left panel, presents a comparison of the zonal means of sRCF of S5P OCRA/ROCINN_CRB (version 1), S5P FRESCO (version 1.3) and OMCLDO2, and of GCF of MODIS, in function of latitude, at day 2020-02-29. Similar
5  results have been obtained at other days (see Fig. S1). There is a good correspondence between the three products between approximately -60° and +40° latitude. OMCLDO2 and S5P OCRA/ROCINN_CRB have a mean difference in $f_{rc,0.8}$ of 0.005 in this region, while S5P FRESCO $f_{rc,0.8}$ is ~0.03 higher than S5P OCRA/ROCINN_CRB $f_{rc,0.8}$. Beyond this latitude range, the sRCF diverge, with $f_{rc,0.8}$ becoming larger for OMCLDO2 and especially for S5P FRESCO, where the sRCF reaches values up to 1.2. This can likely be attributed to the different treatment of snow-ice
10  cases by the different cloud products. Also indicated on the same figure panel is the GCF of MODIS. The latitudinal variation show roughly similar variations as that of the S5P OCRA/ROCINN_CRB, S5P FRESCO and OMCLDO2, but, again as expected, the GCF is larger than the sRCF of the other cloud products. Note that at the extreme latitudes,

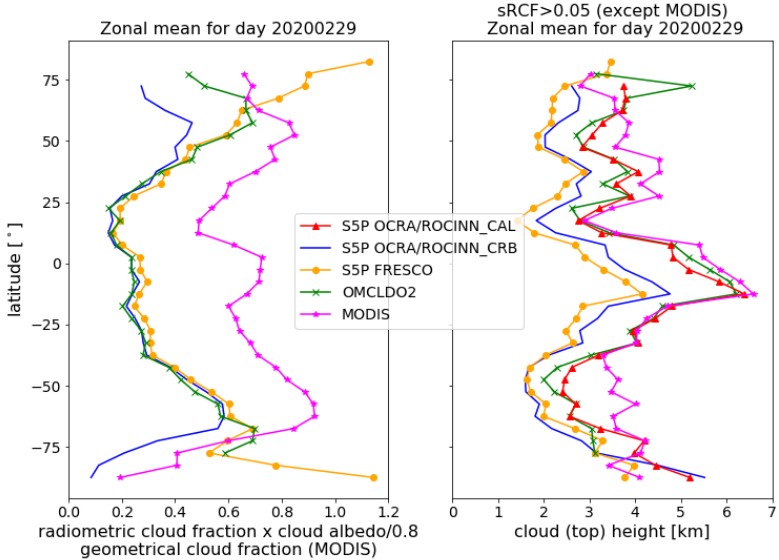

**Figure 4.** Zonal means at 2020-02-29. Left: scaled radiometric cloud fraction $f_{rc,0.8}$ of S5P OCRA/ROCINN_CRB version 1, S5P FRESCO version 1.3 and OMCLDO2, and geometric cloud fraction of MODIS. Right: cloud height of the same data products, and in addition cloud top height of S5P OCRA/ROCINN_CAL version 1. Here, pixels with $f_{rc,0.8} < 0.05$ are excluded (not applicable to MODIS), as the cloud height becomes highly uncertain at very low cloud fraction. Note that S5P OCRA/ROCINN automatically assigns a fill value to the cloud height when RCF<0.05.

the latitudinal variation of MODIS GCF is rather comparable to that of S5P OCRA/ROCINN than to that of S5P FRESCO.

Figure 4, right panel, presents a comparison of the zonal means of cloud height of S5P ROCINN_CRB, S5P FRESCO and OMCLDO2, and of the cloud top height of S5P ROCINN_CAL and MODIS. Pixels with $f_{rc,0.8} < 0.05$
are removed (except for MODIS, where RCF is not applicable), as the cloud height uncertainty becomes very high at these low cloud fractions. Note that for S5P ROCINN_CRB and S5P ROCINN_CAL, pixels with RCF $< 0.05$ are automatically assigned a fill value. While the latitudinal variation of cloud (top) height of the different cloud products are similar, there are also offsets. S5P FRESCO CH is on average a few hunderd meter below S5P OCRA/ROCINN_CRB CH, while S5P OCRA/ROCINN_CAL CTH and OMCLDO2 CH are ~1 km above S5P OCRA/ROCINN_CRB CH.
MODIS CTH is mostly higher than S5P OCRA/ROCINN_CAL CTH, being about 0.5 to 1 km higher between latitudes $[-60°, -40°]$ and $[+30°, -50°]$. Similar conclusions can be drawn for other days (Fig. S1). The consistently higher MODIS CTH compared to S5P OCRA/ROCINN_CAL CTH is also observed when comparing 1 month of data (Fig. S2). These results are consistent with Schuessler et al. (2014) showing that the ROCINN_CAL model retrieves higher clouds than the ROCINN_CRB model, and also consistent with the results from Loyola et al. (2010) showing
higher clouds from infrared sounders compared to ROCINN.

## 4.2 Across-track dependence

The across-track dependence of sRCF of S5P CLOUD CRB and S5P FRESCO, and of C(T)H of S5P CLOUD CAL, S5P CLOUD CRB and S5P FRESCO, for day 2020/02/29, is shown in Fig. 5. Note that only latitudes between 60°N and 60°S are selected to limit the impact of snow/ice. For sRCF, pixels with qa_value > 0.5 (this is the quality indicator for the S5P cloud products) are selected. For C(T)H, in addition sRCF>0.05 is required. Note that (i) S5P FRESCO pixels were first remapped to the S5P CLOUD grid, and (ii) we also show the common subset (FRESCO and ROCINN_CRB) of valid pixels. The sRCF is higher towards the edges of the swath due to enhanced cloud scattering along a slant path, and there is a maximum in the sun glint region, west of the middle row. The cloud height increases towards the edges of the swath, due to the longer slant path in the $O_2$-A absorption, and there is a minimum in cloud height at the location of the sun glint region. These are effects known from other sensors (e.g., Tuinder et al., 2010).

sRCF and C(T)H have a similar shape for the different products, although there are offsets. S5P FRESCO sRCF is slightly higher (about 0.05) than S5P OCRA/ROCINN_CRB, in agreement with Fig. 4 (excluding the extreme latitudes). When for each product only the own screenings are applied, S5P FRESCO CH is lower than S5P OCRA/ROCINN_CRB CH by roughly 200 m. However, it should be noted that the filter settings for S5P FRESCO are less restrictive than for S5P OCRA/ROCINN. When taking the common subset of pixels for both S5P FRESCO and S5P OCRA/ROCINN_CRB, a small shift in FRESCO sRCF towards lower values is visible. Of more significance is the CH shift of S5P FRESCO towards higher values, becoming close to S5P OCRA/ROCINN_CRB CH.

## 4.3 Comparison between S5P OCRA/ROCINN_CAL and NPP VIIRS

### 4.3.1 Data selection and processing

For the current study, six days of NASA VIIRS data have been provided to DLR for the initial validation of S5P OCRA/ROCINN_CAL processor version 1. To enable a pixel-by-pixel comparison, the original 750 m NASA VIIRS pixels have been regridded to the TROPOMI footprints as explained in the S5P-NPP Cloud Processor ATBD (Siddans, 2016). The VIIRS cloud mask is converted to a GCF. This is done by counting the co-located pixels within a TROPOMI footprint and dividing the number of 'confidently cloudy' pixels by the total number of pixels ('confidently cloudy' + 'probably cloudy' + 'probably clear' + 'confidently clear'). The datasets have been filtered according to several criteria to ensure that the comparison is meaningful. Only data with a OCRA/ROCINN_CAL qa_value above 0.5 were selected. The S5P CLOUD snow/ice flag was used to exclude data over such high reflective surfaces because the cloud retrievals are particularly challenging in these conditions. Furthermore, only pixels where both VIIRS GCF>0.9 and OCRA RCF>0.9 contributed to the comparison. This filter step mitigates artefacts of the regridding process especially at the cloud boundaries and at scattered small-scale clouds. Also, this high CF threshold was chosen because there one expects the least deviations between the S5P OCRA RCF and the VIIRS GCF, justifying the intercomparability between the ROCINN and VIIRS (CTH, COT) cloud parameters. Finally, only pixels which obey to the threshold criteria of CTH<15 km and 1<COT<150 were used for the validation exercise. Those thresholds have

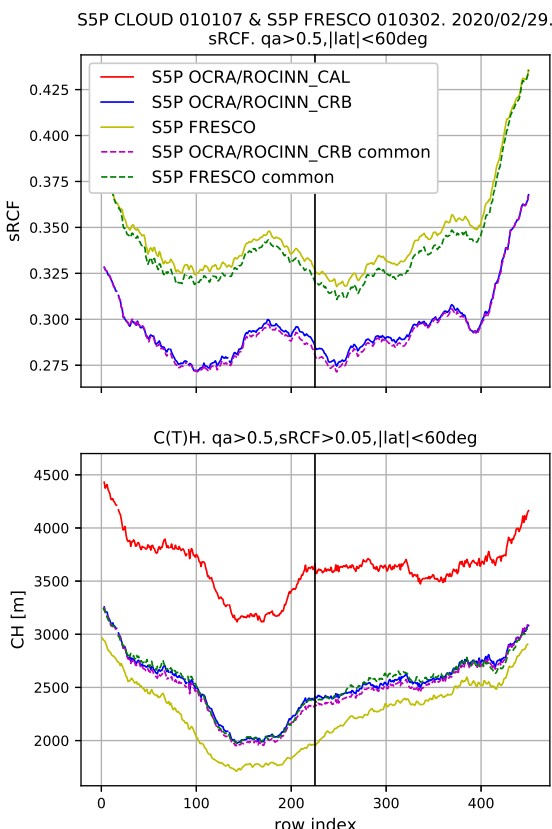

**Figure 5.** (a) Mean pixel value per row index of sRCF of S5P OCRA/ROCINN_CRB and S5P FRESCO. Only pixels with qa_value>0.5 are selected. Note that FRESCO pixels are mapped to those of OCRA/ROCINN. The label 'common' indicates that the common subset of valid pixels are taken. The black vertical line indicates the middle row. Note that index 0 corresponds to the westernmost pixel. (b) C(T)H of S5P OCRA/ROCINN_CAL, S5P OCRA/ROCINN_CRB and S5P FRESCO. Only pixels with qa_value>0.5 and sRCF>0.05 are selected. Other conventions as in (a).

been set because the S5P OCRA/ROCINN algorithm in CLOUD version 1 can retrieve clouds up to a maximum CTH of 15 km and with an optical thickness not lower than 1 while the re-gridded VIIRS COT has a maximum of 150. After the aforementioned filtering and harmonization process of the two datasets, the total number of valid pixels for comparison exceeded the number of 30.000.000.

5       The COT from VIIRS is not comparable directly to the S5P COT because VIIRS has a geometric cloud fraction and not a radiometric one. For optically thin clouds, the radiometric cloud fraction is smaller than the geometric one and this results in a higher associated COT. This is demonstrated in Fig. S3 a and b. For this reason we have introduced

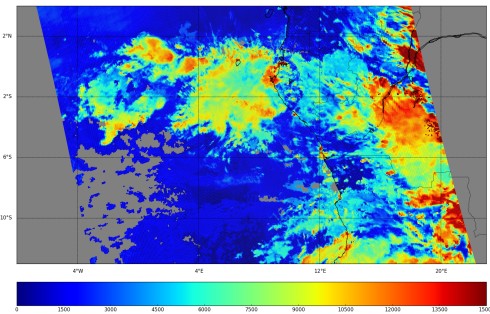 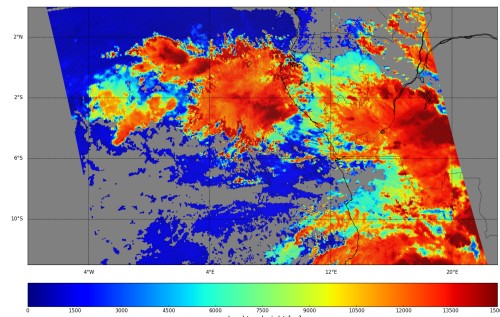

**Figure 6.** Cloud top height of S5P OCRA/ROCINN_CAL processor version 1 (left) and of regridded NPP VIIRS (right), for orbit 01080.

an effective COT equal to COT x RCF. In this way, the lower RCF/higher COT retrieval with respect to a GCF/COT retrieval as in VIIRS are compensated for (Fig. S3, b and c).

As demonstrated by Nakajima and King (1990), the reflection function at 0.75 μm is in principle sensitive to the COT and the reflection function at 2.16 μm is sensitive to the effective radius, but this two-channel retrieval method can also be applied with a slightly different combination of wavelengths. In the case of VIIRS, the exact wavelength combination in use is 2.2 μm and, depending on the surface type, either 0.65 μm, 0.86 μm or 1.24 μm. S5P OCRA/ROCINN_CAL COT is retrieved at the continuum of the Oxygen A-band (outside the absorption band). Therefore, the wavelength coverage should have no significant impact on the COT.

It should be noted that the performance of OCRA/ROCINN is optimal for high geometric cloud fractions and optically thick clouds. The combination of low geometric cloud fraction and optically thin cloud is the most challenging and would be interesting to assess. However, given the intercomparability limitations noted above for lower cloud fractions, we consider a deeper analysis beyond the scope of this paper.

### 4.3.2 Results

Figure 6 presents, for part of orbit 01080, the cloud top height of S5P OCRA/ROCINN_CAL and of NPP VIIRS, after regridding to the same pixel size as S5P OCRA/ROCINN. Figure S3 shows the same for cloud optical thickness. While similar cloud features can be discerned in the S5P OCRA/ROCINN_CAL and the NPP VIIRS plots, there are also quantitative differences.

The daily distribution and statistical characteristics do not vary significantly between the different days, as it can be seen from the box plots of Fig. S4. In particular, as far as the COT is concerned, the distribution for TROPOMI is much wider than the one of VIIRS (with a standard deviation of TROPOMI being 22.8 compared to 12.7 for VIIRS)

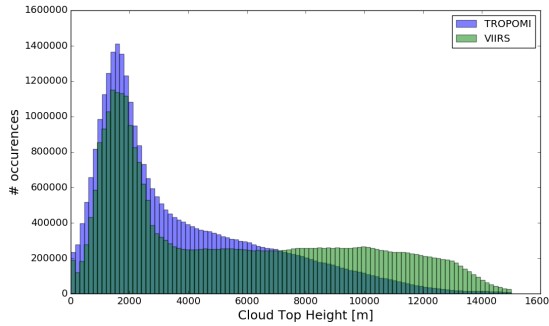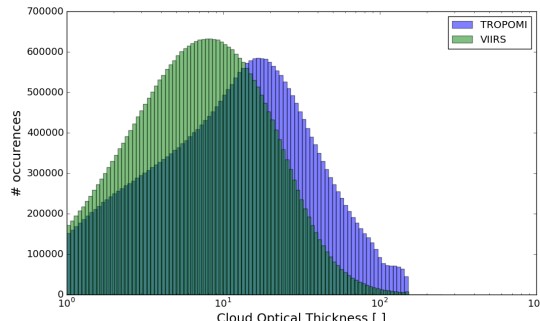

**Figure 7.** Histograms of the CTH (top) and COT (bottom) for TROPOMI OCRA/ROCINN_CAL processor version 1 and VIIRS. The complete dataset is considered. Note that for TROPOMI the effective COT (i.e., original COT x CRF) is used.

and the median for TROPOMI is about 12 while for VIIRS it is about 7. The first quartile Q1 for both instruments is 3, but the third quartile Q3 is higher in TROPOMI than in VIIRS (i.e., 24 and 15, respectively). For the CTH, the distribution for TROPOMI is narrower than the one for VIIRS with a standard deviation of 2.9 km and 4.0 km for TROPOMI and VIIRS, respectively. The median for TROPOMI is 2.5 km and for VIIRS it is 4 km. Similarly to the
COT, the first quartile Q1 is for both sensors the same around 1.8 km. However, the third quartile is for TROPOMI at 6 km and for VIIRS at 9 km.

   The general features of the statistical measures can be drawn from Fig. 7, which depict the histograms of the CTH and COT, respectively. The complete dataset including both surface types (land and ocean/water) for the 6 days is used. First of all, one can see that both instruments capture the same CTH mode at ~1.8 km. This mode is mainly dominant
over the ocean (see Fig. S5 in the supplementary information) and it refers to the low-level marine stratocumulus clouds. Differences at the tails for the CTH distributions are present. TROPOMI seems to underestimate the high level clouds with CTH larger than 8 km. The mean TROPOMI CTH is lower than the one from VIIRS (3.8 km and 5.4 km, respectively). The observed negative bias is in the order of 1.6 km. As will be seen in Sect. 4.4, a negative bias is also observed from the Cloudnet comparison. Therefore, this is a general outcome of the CTH validation using independent
sensors. Regarding the COT, TROPOMI seems to overestimate this cloud parameter meaning that the clouds appear optically thicker than in VIIRS. The S5P ROCINN overestimation is consistent with the GOME ROCINN results from (Loyola et al., 2010). The mean COT from VIIRS was found at 11.4 whereas the mean COT of TROPOMI is found at 19.2 leading to a positive bias of 7.9. The positive bias in the COT and the negative bias in the CTH is also seen from the zonal means (see Fig. 3). The explanation for those biases are mainly related to the fact that S5P
OCRA/ROCINN_CAL assumes liquid-water clouds and their properties stem from the Mie scattering theory. The ice clouds are not parameterized with the current S5P ROCINN_CAL algorithm in CLOUD version 1 but they will be included in future versions. While water clouds are assumed to be composed entirely by spherical droplets, ice clouds consist of a variety of habits (i.e., mixtures of randomly-oriented hexagonal plates and columns, two dimensional

bullet rosettes and aggregates). The direct impact of the cloud microphysics to the retrieved COT is discussed in Zeng et al. (2012).

The similarity of both datasets is summarized with a Taylor diagram (Taylor, 2001) in Fig. S6. The correlation coefficients for the cloud parameters CTH and COT are shown based on the surface type. The CTH is highly correlated

for both surface types with the correlation coefficient r being 0.86 and 0.74 over water and land, respectively. Similarly, the COT appears with a higher correlation coefficient r=0.66 over water in comparison to 0.48 over land. The low correlation coefficients for the COT over land might be due to non-realistic surface albedo values (extracted from a climatology). Usually, over land the surface albedo might change more rapidly than over water. The CTHs for land and water, which are lying in the inner area of the dashed arc in the Taylor diagram, imply that the corresponding S5P

dataset has a lower standard deviation than VIIRS, indicating S5P pattern variations with a decreased amplitude. The COTs for land and water, which are lying in the outer area, imply that the S5P pattern variations are higher than those of VIIRS. Moreover, the CTH appears with a lower root-mean-square (RMS) error than the COT.

The agreement between NASA VIIRS and S5P OCRA/ROCINN_CAL seems to be much better for the low-level clouds which usually consist of liquid water particles. The main question is how well the two sensors agree for the

several cloud types. For identifying in which type of clouds the differences are larger we follow the ISCCP (International Satellite Cloud Climatology Project) classification (Schiffer and Rossow, 1983). The scheme depicted in Fig. S7 classifies the clouds based on their CTH and COT combinations. Cumulus, stratocumulus and stratus are the low-level clouds, the altocumulus, altostratus and nimbostratus are the mid-level clouds and finally, the cirrus, cirrostratus and deep convective are classified as high-level clouds. The biases for all low-, mid- and high-level clouds are summarized

in Table 4; note that we classify based on the VIIRS cloud properties as this is the selected reference.

**Table 4.** Mean difference (S5P OCRA/ROCINN_CAL minus VIIRS) of COT and CTH, classified according to the ISCCP scheme. In brackets, the mean values for both sensors are included. Note that for TROPOMI COT an effective COT is used (=original COT x CRF). The classification is done using the VIIRS cloud properties.

| Low-level | Cumulus | | Stratocumulus | | Stratus | |
|---|---|---|---|---|---|---|
| | Land [2.5%] | Water [13%] | Land [10%] | Water [32%] | Land [3.4%] | Water [3.5%] |
| COT [ ] | +0.1 [VIIRS 2.3, TROPOMI 2.4] | +0.5 [VIIRS 2.3, TROPOMI 2.8] | +2.4 [VIIRS 10.7, TROPOMI 13.1] | +5.6 [VIIRS 9.6, TROPOMI 15.2] | +1.7 [VIIRS 38.8, TROPOMI 40.5] | +11.2 [VIIRS 33.5, 44.7] |
| CTH [km] | -0.1 [VIIRS 1.8, TROPOMI 1.7] | -0.5 [VIIRS 1.3, TROPOMI 1.8] | -0.7 [VIIRS 2.2, TROPOMI 1.5] | -0.2 [VIIRS 1.7, TROPOMI 1.5] | -0.9 [VIIRS 2.2, TROPOMI 1.3] | -0.4 [VIIRS 1.9, TROPOMI 1.5] |

| Mid-level | Altocumulus | | Altostratus | | Nimbostratus | |
|---|---|---|---|---|---|---|
| | Land [2.8%] | Water [2.9%] | Land [14%] | Water [10%] | Land [4.8%] | Water [2.6%] |
| COT [ ] | +0.7 [VIIRS 2.4, TROPOMI 3.1] | +3.3 [VIIRS 2.3, TROPOMI 5.6] | +3.9 [VIIRS 10.8, TROPOMI 14.7] | +10.7 [VIIRS 10.8, TROPOMI 21.5] | +0.5 [VIIRS 41.4, TROPOMI 41.9] | +13.2 [VIIRS 37.6, 50.8] |
| CTH [km] | -0.8 [VIIRS 4.6, TROPOMI 3.8] | -0.5 [VIIRS 4.5, TROPOMI 4.0] | -1.5 [VIIRS 4.4, TROPOMI 2.9] | -1.6 [VIIRS 4.5, TROPOMI 2.9] | -1.7 [VIIRS 4.3, TROPOMI 2.6] | -1.9 [VIIRS 4.6, TROPOMI 2.7] |

| High-level | Cirrus | | Cirrostratus | | Deep convective | |
|---|---|---|---|---|---|---|
| | Land [15%] | Water [10%] | Land [38%] | Water [21%] | Land [9%] | Water [5%] |
| COT [ ] | +1.6 [VIIRS 2.3, TROPOMI 3.9] | +3.3 [VIIRS 2.2, TROPOMI 5.5] | +8.3 [VIIRS 9.9, TROPOMI 18.2] | +15.5 [VIIRS 10.2, TROPOMI 25.7] | +10.1 [VIIRS 42.0, TROPOMI 52.1] | +21.8 [VIIRS 39.2, 61.0] |
| CTH [km] | -2.4 [VIIRS 10.1, TROPOMI 7.7] | -2.5 [VIIRS 10.2, TROPOMI 7.7] | -3.7 [VIIRS 10.0, TROPOMI 6.3] | -3.6 [VIIRS 9.5, TROPOMI 5.9] | -4.1 [VIIRS 9.9, TROPOMI 5.8] | -3.8 [VIIRS 9.6, TROPOMI 5.8] |

Cumulus and stratocumulus (low-level clouds) over water appear 50% of the time with small negative CTH biases of a few hundred meters. The largest COT bias among the low-level clouds appears for the stratus type over water, but these type of clouds are not so frequent. From the mid-level clouds (see Table 4) the altocumuli show a low CTH bias, but the other two types (among which the most frequent is the altostratus), have a negative bias of about 1.5 km. Altostratus and nimbostratus over water appear with a high positive bias in the COT. Finally, from the high-level clouds, the cirrostratus and cirrus, which appear with a frequency higher than 80%, show high biases in both CTH and

COT. All in all, the agreement between VIIRS and TROPOMI cloud properties is certainly best for low level clouds and worse for high-level clouds.

## 4.4 Comparison of S5P cloud height with Cloudnet

In this section we discuss the comparison of S5P OCRA/ROCINN_CAL CTH, S5P OCRA/ROCINN_CRB CH and S5P FRESCO CH with ground-based Cloudnet data. Moreover, we compare also OMI OMCLDO2 with Cloudnet, using the same methodology, as this allows to make the connection with the work of Veefkind et al. (2016). By comparing the S5P products with the Cloudnet data on one hand, and OMCLDO2 with Cloudnet data on the other hand, one learns better how the effective cloud heights of these different products relate to the (vertically resolved) lidar/radar cloud observations of Cloudnet, and where they are different. Aura/OMI has a similar overpass time as S5P/TROPOMI. Like S5P OCRA/ROCINN and FRESCO, OMCLDO2 provides rather effective cloud heights which are used as input in the retrieval of atmospheric gases. It should be noted that other OMI cloud products could have been taken here for comparison, like the OMCLDRR which is based on the Fraunhofer filling signatures (346-354 nm) to derive effective cloud fraction and cloud optical centroid pressure (Joiner and Vasilkov, 2006), but this is beyond the scope of the current work.

In this comparison, we have used S5P CLOUD RPRO and OFFL files with processor version 1.1.7, and S5P FRESCO RPRO+OFFL files with processor version 1.3, from 2018-04-30, to 2020-02-27.

Satellite - Cloudnet comparison pairs are established as follows.

– Where applicable, the satellite RCF is converted to an sRCF, at CA=0.8.

– Satellite pixels are selected only if they cover the Cloudnet site, have a qa_value>50% (S5P) or no error flag (OMCLDO2), and have sRCF > 0.05.

– All Cloudnet measurements within $\pm 600$ s of a S5P or OMCLDO2 overpass are considered. From these, the Cloudnet cloud occurrence fraction (COF), mean Cloudnet CTH $h_{ct}$ and CMH $h_{cm}$, and standard deviation of Cloudnet CTH $\sigma(h_{ct})$ and CMH $\sigma(h_{cm})$ are calculated.

– To limit temporal variability, co-locations are selected only if Cloudnet COF > 50%, $\sigma(h_{ct}) < 0.5$ km and $\sigma(h_{cm}) < 0.5$ km.

Despite these filter criteria, some comparison error due to co-location mismatch will persist. Another, more fundamental, problem is that the cloud (top) heights obtained from the radar-lidar based Cloudnet data on one hand, and the more 'effective' cloud (top) heights of the S5P and OMI products are not fully comparable. For example, while the Cloudnet CMH is obtained as a simple mean of all 'cloudy' grid positions (regardless of the local optical thickness), the cloud (top) heights returned by the S5P and OMI cloud products do depend on optical thickness. Moreover, none of the S5P or OMI cloud products take into account the possibility that a cloud can be multi-layered.

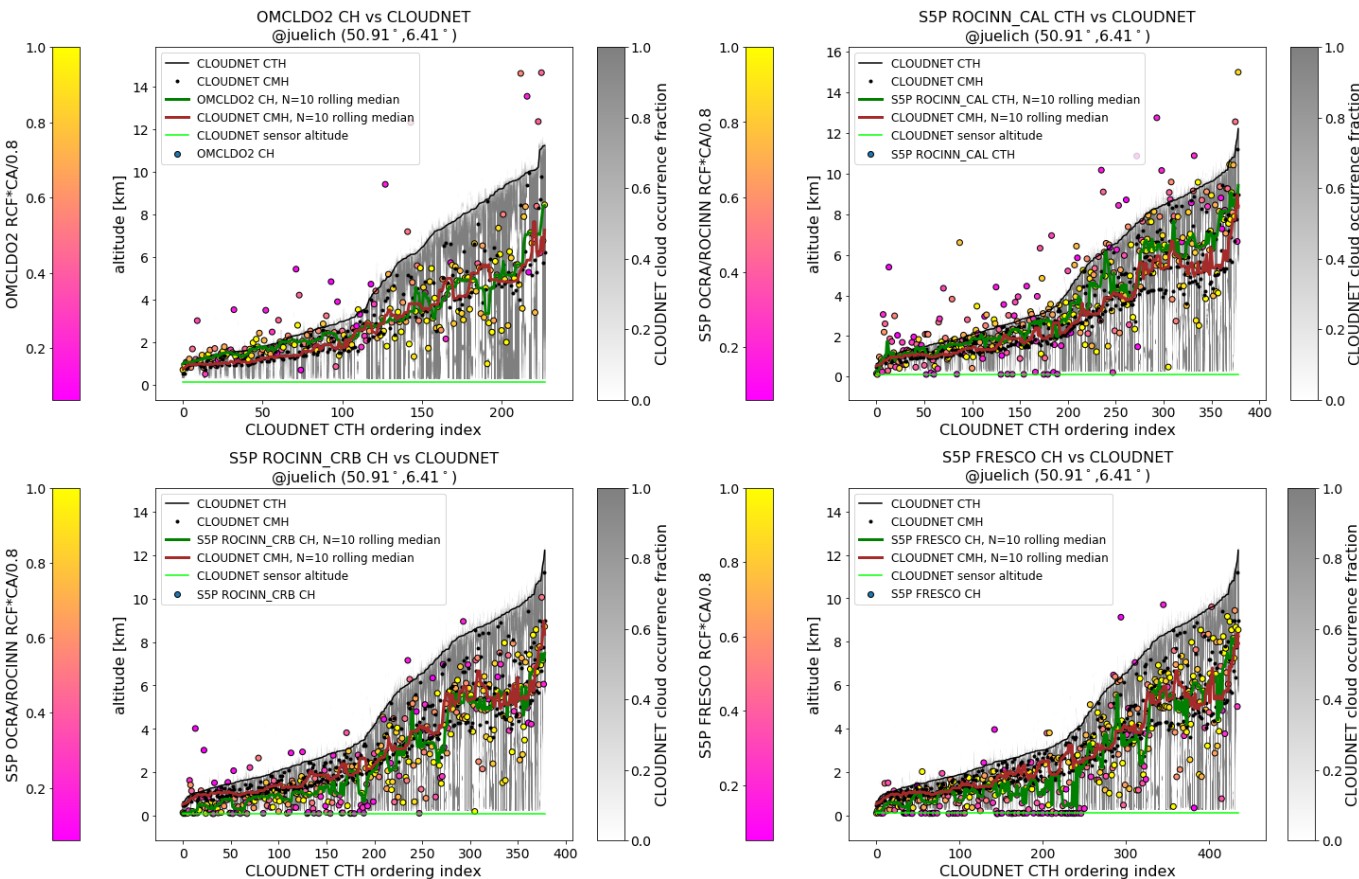

**Figure 8.** Comparisons of OMCLDO2 CH (top left), S5P OCRA/ROCINN_CAL CTH (top right), S5P OCRA/ROCINN_CRB CH (bottom left) and S5P FRESCO CH (bottom right) with Cloudnet, at the site Juelich. Along the x-axis, the cases are ordered according to the Cloudnet CTH. Black line indicates the Cloudnet CMH. Satellite data points are coloured based on the sRCF (left colour bar). 10-point window rolling medians based on the satellite data (green line) and on the Cloudnet CMH data (brown line) are added as well. The grey background is the vertically resolved cloud occurrence fraction derived from the Cloudnet data for the period ±600 s of the satellite overpass (right colour bar).

Figure 8 presents comparisons between satellite (OMCLDO2 CH, S5P OCRA/ROCINN_CAL CTH, S5P OCRA/ROCINN_CRB CH, S5P FRESCO CH) and Cloudnet, at the site Juelich. Co-location pairs are ordered along Cloudnet CTH. Similar plots are provided in the supplement for the other sites.

Furthermore, normed histograms and associated estimated probability density distributions of OMCLDO2 CH vs Cloudnet CTH, and S5P OCRA/ROCINN_CAL CTH vs Cloudnet CTH, are provided in Fig. 9. Similar plots are provided in the supplement (Sect. S3) for the other sites, with the exclusion of sites with less than 70 co-located data pairs, and of the site Summit, where most satellite cloud height retrievals are problematic.

Following conclusions can be drawn:

- Co-location pairs with a low satellite RCF are more scattered, in line with the higher cloud height uncertainty.

- **OMCLDO2 vs Cloudnet.** There are far fewer co-locations with Cloudnet available for OMCLDO2 than for the S5P cloud products, showing a clear advantage for S5P. For the lowest (Cloudnet CTH$\lesssim$2 km; mostly liquid) clouds, OMCLDO2 CH corresponds to the Cloudnet CTH. At Cloudnet CTH $\gtrsim$3 km, OMCLDO2 CH rather corresponds to the Cloudnet CMH (e.g., Juelich, Palaiseau, Mace Head) or is below the Cloudnet CMH (e.g., Munich, Schneefernerhaus, Leipzig). The latter pattern was also seen at Cabauw by Veefkind et al. (2016) (their figure 10). At Juelich, both Cloudnet CTH and OMCLDO2 CH distributions (Fig. 9) have a low-altitude local mode at a well-matching $\approx$2 km. Also, for several other sites, the low-altitude mode agrees within 20% (Fig. 10), but there are also exceptions (Lindenberg, Norunda, Ny Alesund).

- **S5P OCRA/ROCINN_CAL vs Cloudnet.** On average, Cloudnet CTH $\gtrsim$ S5P OCRA/ROCINN_CAL CTH $\gtrsim$ Cloudnet CMH. For the higher clouds (Cloudnet CTH$\gtrsim$4 km) it is in most cases closer to Cloudnet CMH. The low-altitude local CTH modes of Cloudnet and S5P OCRA/ROCINN_CAL are reasonably well matched: in most cases they agree within 20% (Fig. 10). At several sites (e.g., Graciosa island, Juelich, Chilbolton) a higher altitude CTH mode is also captured by S5P OCRA/ROCINN_CAL, but shifted towards lower altitude compared to Cloudnet CTH. Mean and median S5P OCRA/ROCINN_CAL CTH are lower than those of Cloudnet, mainly due to the CTH mismatch of the higher altitude clouds which have an ice component.

- **S5P OCRA/ROCINN_CRB and S5P FRESCO vs Cloudnet.** S5P OCRA/ROCINN_CRB CH and S5P FRESCO CH are on average below Cloudnet CMH for clouds with Cloudnet CTH$\lesssim$4 km. For higher clouds the satellite CH rather corresponds to Cloudnet CMH.

- It can be seen from Fig. 8 that in a number of cases S5P FRESCO and S5P OCRA/OROCINN_CRB retrieve a cloud height equal to the surface altitude. This contributes at least partly to the on average lower S5P OCRA/ROCINN_CRB CH and S5P FRESCO CH compared to Cloudnet CMH. Ground height retrievals occur also for S5P OCRA/ROCINN_CAL, but to a far less extent. It does not occur for OMCLDO2. For S5P FRESCO, these low cloud height retrievals can be attributed to the low sensitivity of the selected window of the $O_2$ A band to low clouds (see Sect. 2.2.2; this will be improved for FRESCO 1.4 with the new window selection), while the $O_2$-$O_2$ band employed by OMI OMCLDO2 has a better sensitivity for low clouds (Acarreta et al., 2004). Regarding S5P ROCINN_CRB and ROCINN_CAL, deeper investigations are needed to conclude under which particular situations these low retrievals happen, or why they are less prevalent for ROCINN_CAL.

Local conditions can impact the comparison:

- The Cloudnet station at Schneefernerhaus is located at a mountain, at 2.7 km, while the surface altitude attributed to the relatively coarse satellite pixels is generally lower. This causes a mismatch in cloud height for low altitude clouds, where the cloud height observed by the satellite can be below the station altitude.

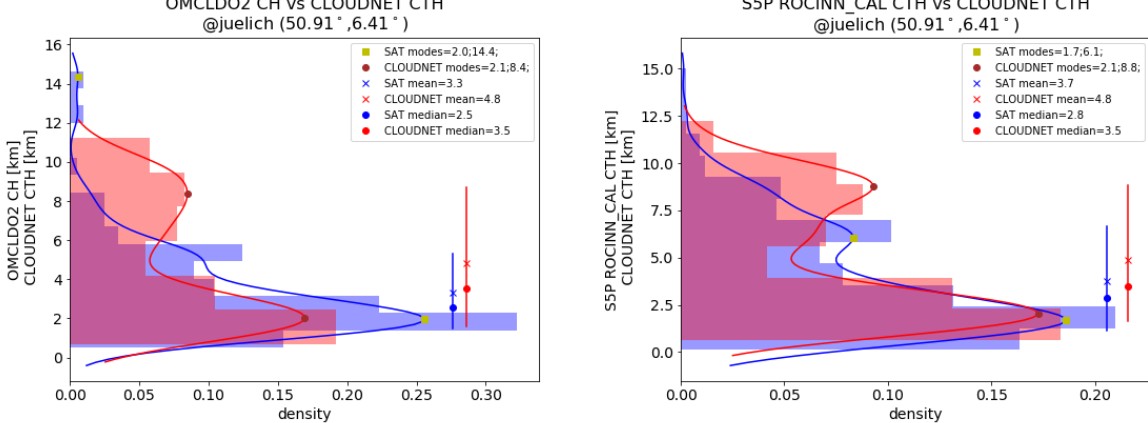

**Figure 9.** Normed histograms of satellite-Cloudnet co-located cloud height or cloud top height, with superimposed density estimates using Gaussian kernels (from the python scipy.stats package), at the site Juelich. OMCLDO2 CH vs Cloudnet CTH (left) and S5P OCRA/ROCINN_CAL CTH vs Cloudnet CTH (right). The most important local modes of the satellite and Cloudnet distributions are indicated, as well as mean, median and the central 68% interval.

- Ny-Alesund can be affected by snow-ice conditions and as a consequence a high surface albedo, which is a challenge for satellite cloud retrievals. Different satellite products can be affected in different ways. OMCLDO2 is characterized by many high-sRCF data points with a cloud height equal to the surface altitude. For S5P OCRA/ROCINN_CAL, S5P OCRA/ROCINN_CRB and S5P FRESCO, one notices a significant number of low-sRCF data points where the retrieved cloud height is significantly overestimating the Cloudnet CTH.

- Summit is covered by permanent ice. OMCLDO2 overestimates the Cloudnet CTH. For most data points of S5P OCRA/ROCINN_CAL, S5P OCRA/ROCINN_CRB and S5P FRESCO, cloud heights equal to the surface altitude are obtained.

Figs. 11 and 12 present boxplot comparisons between the S5P cloud products cloud (top) height and Cloudnet height, with indications of the mission requirements on bias and uncertainty; the latter is compared here with a robust dispersion estimator: 0.5 of the central 68 interpercentile interval (0.5 IP68), which amounts to one standard deviation in the ideal case of a Gaussian error distribution. It should be noted that apart from satellite error, several other components contribute to the bias and dispersion: measurement error in the ground-based data, temporal and spatial co-location mismatch, and the fact that the effective cloud heights from satellite, and those from Cloudnet, are not fully comparable. In particular regarding the calculated dispersion vs the stated uncertainty requirement, it must be clear that this can only serve as a partial quality test. If the dispersion is *lower* than the uncertainty threshold, one can be confident that the satellite uncertainty is within the threshold. If, on the other hand, the dispersion is *higher* than the uncertainty threshold, it is by no means a proof that the satellite data exceeds the uncertainty threshold.

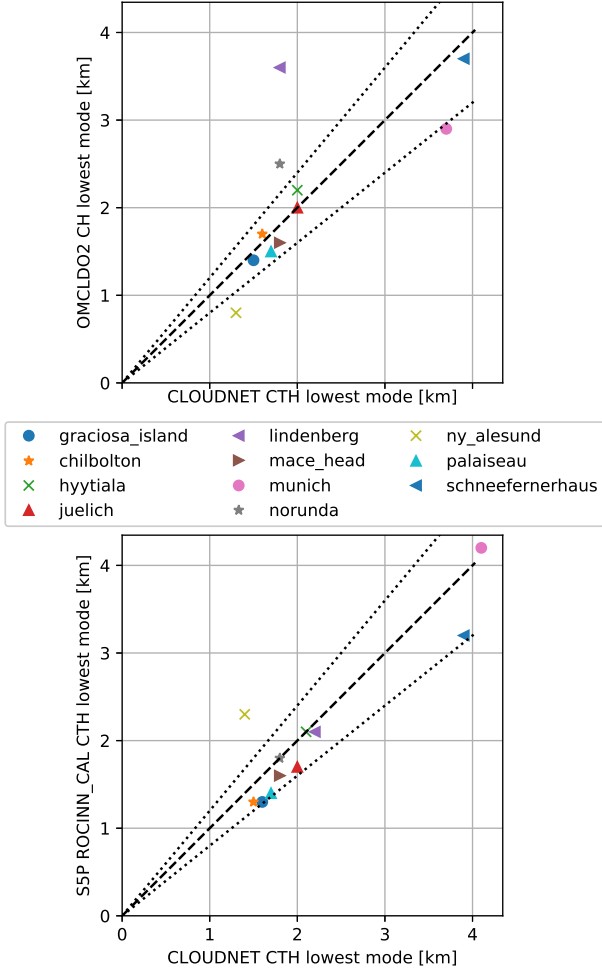

**Figure 10.** Top. Correlation plot between the lowest CTH modes of OMCLDO2 and Cloudnet. Only sites with more than 70 co-location pairs are considered. Dashed line is the 1:1 line, dashed lines are the $\pm 20\%$ deviations from the 1:1 line. Bottom. The same, but between the lowest CTH modes of S5P OCRA/ROCINN_CAL and Cloudnet.

Figure 11 presents boxplot comparisons between S5P OCRA/ROCINN_CAL CTH and Cloudnet CTH, and between S5P OCRA/ROCINN_CAL CTH and Cloudnet CMH. In agreement with the above results, S5P OCRA/ROCINN_CAL CTH minus Cloudnet CTH is characterized by a negative bias, bordering to or exceeding the 20% bias requirement. However, if one compares S5P OCRA/ROCINN_CAL CTH with the Cloudnet CMH, the bias requirement is fulfilled in most cases. The dispersion exceeds the 0.5 km uncertainty threshold in almost all cases (but see the note on the calculated bias and dispersion vs. bias and uncertainty thresholds above).

Figure 12 presents boxplot comparisons between S5P OCRA/ROCINN_CRB CH and Cloudnet CMH, and between S5P FRESCO CH and Cloudnet CMH. A negative bias is observed at almost all sites, often exceeding the 20% bias requirement. Again, the dispersion exceeds the 0.5 km uncertainty threshold in almost all cases (but see the note on the calculated bias and dispersion vs. bias and uncertainty thresholds above).

## 5   Impact of processor version upgrades

In this section the impact of recently released processor version upgrades is shortly discussed. Note that the plots of the upgrades in this work are not based on operational data, but on pre-release processings.

### 5.1   S5P OCRA/ROCINN: version 2 vs version 1

Geographical or swath related patterns may appear for some S5P OCRA/ROCINN parameters in S5P OCRA/ROCINN CLOUD version 1 (Lutz et al., 2016; Richter and the Verification Team, 2015). Their appearance is not fully deterministic and is mainly related to the clear-sky background reflectance maps and scan-angle dependency correction that are both using OMI data in S5P CLOUD version 1. These OMI-based auxiliary data are functions of several parameters, e.g. time, wavelength, latitude, viewing zenith angle etc. The patterns listed below are not a general issue seen at all times and geolocations but rarely appear only for some combinations of (time, geometry, geolocation). With the update to CLOUD version 2, these OMI-based auxiliary data are replaced based on the TROPOMI data themselves and the effects listed below are largely reduced.

The following patterns may appear in S5P OCRA/ROCINN CLOUD version 1:

– an enhanced radiometric cloud fraction and cloud height mainly at the east edge of the swath at some months at some latitudes. Most pronounced effects seem to appear in the bands [40,60]°N and [30,40]°S. Figure 13 illustrates the issue for an example in the cloud top height in S5P CLOUD version 1 and the improvement in S5P CLOUD version 2, while Fig. S28 shows the issue for an example in the cloud fraction.

– A gradient in the cloud albedo with higher values in the northern hemisphere compared to the southern hemisphere (Fig. S29).

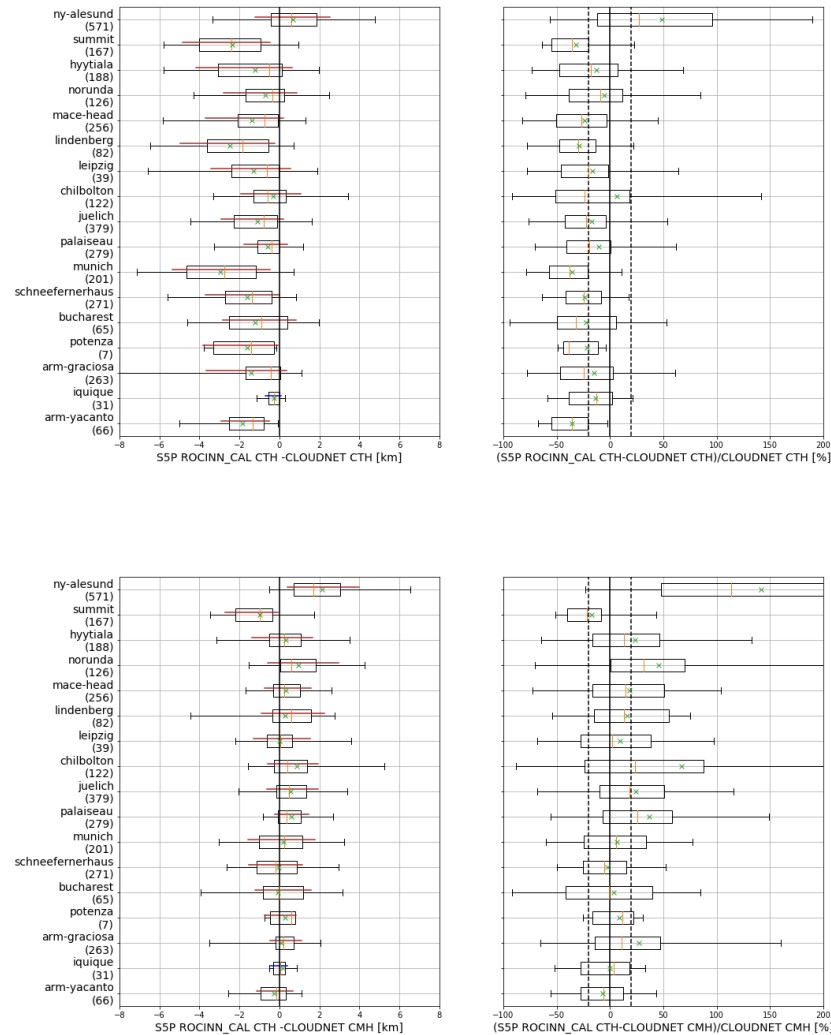

**Figure 11.** Top. Boxplot of the difference (left), and relative difference (right) between S5P OCRA/ROCINN_CAL CTH and Cloudnet CTH. Bottom. The same but between S5P OCRA/ROCINN_CAL CTH and Cloudnet CMH. Figure conventions are as follows. Box edges: first and third quartile; line: median; whiskers: 5 and 95 percentiles; cross: median. Furthermore, a line corresponding to the central 68% interval is indicated. Its color is brown when the 0.5 IP68 exceeds the dispersion requirement, and green otherwise.

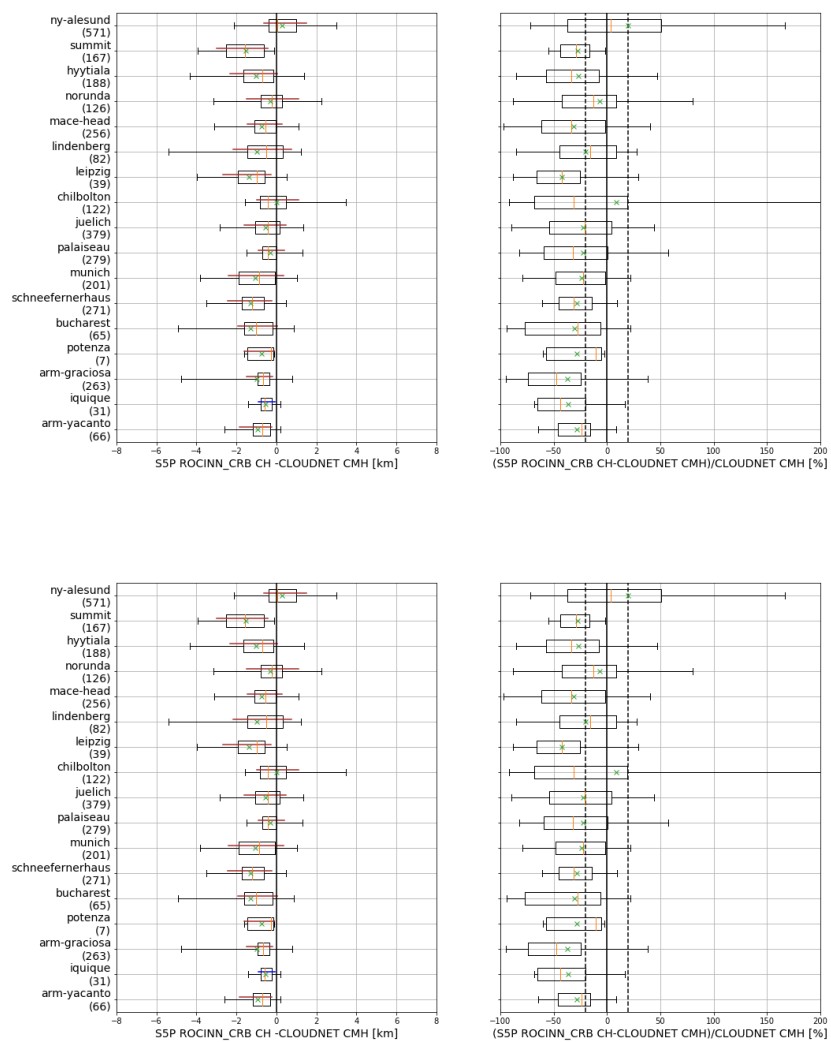

**Figure 12.** Top. Boxplot of the difference (left), and relative difference (right) between S5P OCRA/ROCINN_CRB CH and Cloudnet CMH. Bottom. The same but between S5P FRESCO CH and Cloudnet CMH. The same conventions as in Fig. 11 are followed.

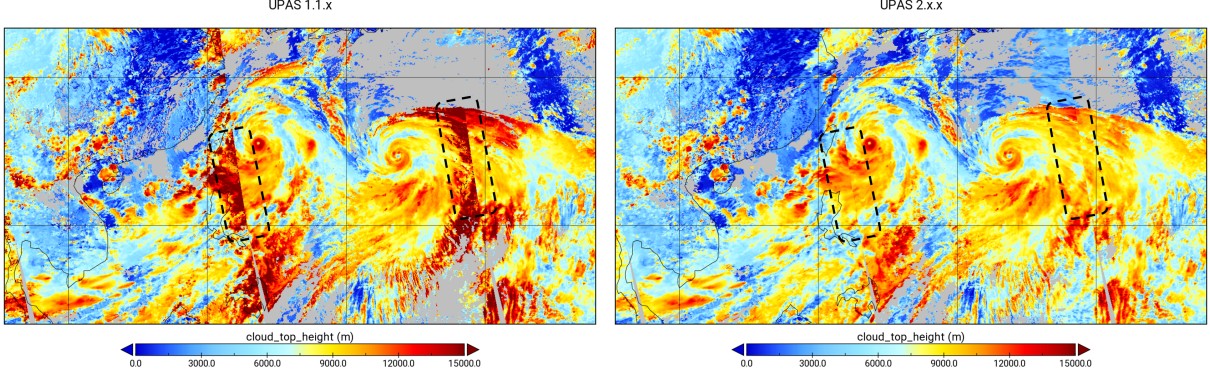

**Figure 13.** S5P OCRA/ROCINN_CAL CTH of parts of orbits 09416, 09417, 09418 on 2019-08-08 for CLOUD OFFL 1.1.7 (left) and CLOUD version 2 (right). Note the regions with sharper contrast in CTH across orbit edges for version 1 which have largely disappeared for version 2 (indicated with black dashed rectangles).

## 5.2 S5P FRESCO: version 1.4 vs version 1.3

An issue in S5P FRESCO 1.3 is that, at low radiometric cloud fraction, there is a tendency to retrieve a cloud height (or aerosol height, as the algorithm does not discriminate between aerosol and cloud, (Wang et al., 2012)) equal to the surface altitude. Errors in the cloud (or aerosol) height can have an important impact on the retrieval of tropospheric

$NO_2$ columns by TROPOMI.

As an example, we discuss here a cloud-aerosol event captured by TROPOMI over China at 2019-02-23 (Fig. 14 and Fig. S30). The aerosol is observed by the S5P Absorbing Aerosol Index (AAI) product (Fig. 14, top left), and attributed a height of 300 to 500 m above the surface by the S5P Aerosol Layer Height (ALH) product (Fig. S30, bottom right). A low RCF cloud (RCF~0.3) of approximately the same shape is perceived by S5P FRESCO (Fig. 14,

top right). Although a cloud is detected by S5P FRESCO, it is attributed zero offset from the surface (Fig. 14, bottom left).

A new version of S5P FRESCO, with a more wide fit window ('FRESCO-A wide') is very recently released (S5P FRESCO version 1.4). For this new product, the sensitivity to low clouds in the low atmosphere is improved. Figure 14, bottom right, shows that FRESCO-A wide places the cloud at 300-500 m above the surface. The steps in this figure

are an artefact caused by the spectral smile effect of the TROPOMI 2D-spectrometer.

The improvement in $NO_2$ column retrieval by using FRESCO-A wide instead of FRESCO version 1.3 is discussed in more detail by Eskes et al. (2020). Note that for scattering aerosols with little absorption the improvement of FRESCO algorithm is expected to have the same effect as for clouds: the raising aerosol layer height will improve the $NO_2$ column. For (strongly) absorbing aerosols the radiative transfer is more complicated and its effect on $NO_2$

retrievals has to be analysed separately (see e.g., Chimot et al., 2019).

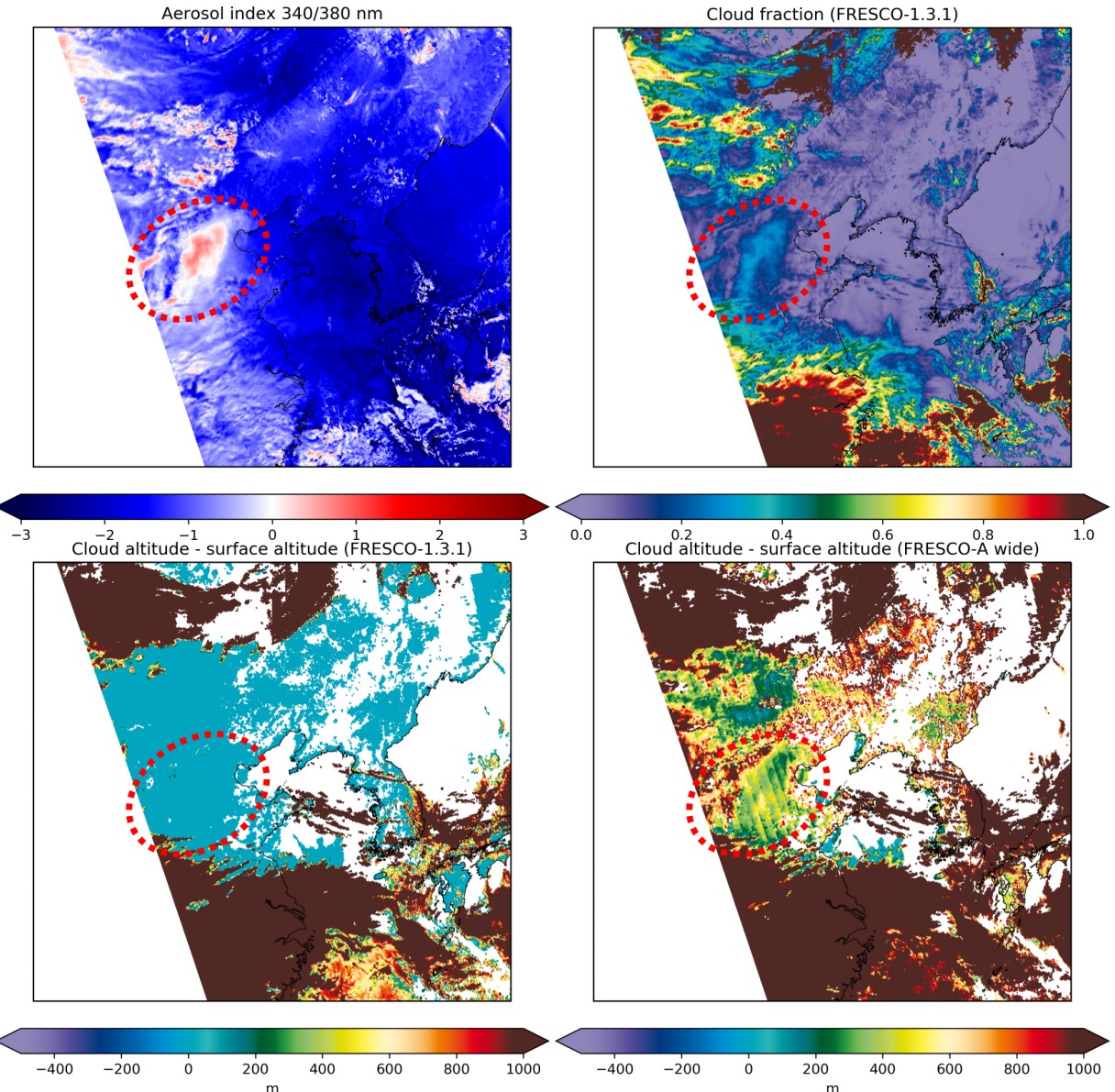

**Figure 14.** S5P Aerosol Index OFFL 1.2.2 (top left), S5P FRESCO OFFL 1.3.1 cloud fraction (top right), S5P FRESCO OFFL 1.3.1 cloud height offset from the surface (bottom left) and S5P FRESCO-A wide (version 1.4) cloud height offset from the surface (bottom right). Orbit 7062 at 2019-02-23, 1200x1200 $km^2$ square centered at 38°N, 120°E. The cloud height products are filtered using qa_value > 0.5 and CF > 0.05. The region of interest is indicated by the red-dashed ellipse.

# 6 Discussion and conclusions

The TROPOMI cloud products S5P OCRA/ROCINN_CAL, S5P OCRA/ROCINN_CRB and S5P FRESCO are validated in this work, using independent satellite data (NPP VIIRS, MODIS and OMI OMCLDO2) and ground-based Cloudnet data. The following conclusions are obtained.

- In the comparison of zonal means between different satellite products, similar latitudinal variations in cloud fraction, cloud (top) height and cloud optical thickness are obtained, sometimes with offsets. Radiometric cloud fractions, scaled to a fixed cloud albedo, between S5P OCRA/ROCINN, S5P FRESCO and OMCLDO2 agree well, except at extreme latitude where S5P FRESCO diverges.

- The across-track dependence of sRCF and C(T)H shows a similar variation for S5P OCRA/ROCINN and FRESCO. Cloud height offsets between ROCINN_CRB and FRESCO are largely reduced when the common set of pixels is taken.

- CTH and COT of S5P OCRA/ROCINN_CAL and NPP VIIRS CTH agree best for low-level liquid clouds, while they disagree for the high-level clouds. A similar conclusion is reached when comparing S5P OCRA/ROCINN_CAL CTH with ground-based Cloudnet CTH. In a future release of S5P CLOUD, liquid clouds and ice clouds will be retrieved separately. Then an improved agreement with NPP VIIRS and Cloudnet can be expected.

- The different S5P cloud products, and OMCLDO2, track different vertical portions of a cloud (as observed by Cloudnet). For low clouds, OMCLDO2 CH corresponds to the Cloudnet CTH, while S5P OCRA/ROCINN_CRB CH and S5P FRESCO CH are below the Cloudnet CMH. For higher clouds, OMCLDO2 CH is sometimes at, but also sometimes below, the Cloudnet CMH. This is in line with expectations: there is a reduced sensitivity for high clouds due to the reduced absorption at low pressures due to the density-squared nature of the absorption feature (Acarreta et al., 2004). On the other hand, S5P OCRA/ROCINN_CRB CH and S5P FRESCO CH rather follow the Cloudnet CMH for high clouds. S5P OCRA/ROCINN_CAL CTH is mostly somewhere between the Cloudnet CMH and the Cloudnet CTH.

- As opposed to ROCINN_CRB and FRESCO (both based on a Lambertian model), ROCINN_CAL (based on Mie scattering cloud model), is well able to match the lowest CTH mode of the Cloudnet observations. At several Cloudnet sites, ROCINN_CAL also observes a second high mode, but shifted towards smaller CTH compared to the Cloudnet CTH. Furthermore, S5P OCRA/ROCINN_CAL CTH has far less a tendency to retrieve a cloud height equal to the surface altitude.

- S5P OCRA/ROCINN RCF and C(T)H can exhibit enhanced values at the east swath edge and a N-S gradient in the cloud albedo. This is improved with the recently released S5P CLOUD version 2.

- S5P FRESCO has a tendency to retrieve at low cloud fraction a cloud height equal to the surface altitude. This is improved with the recently released S5P FRESCO 1.4 version.

Typical applications of the TROPOMI cloud products are in the context of cloud impact on atmospheric composition measurement, such as masking of a measurement scene, accounting for modification in radiative transfer (e.g., the air mass factor) or cloud slicing (e.g., to estimate the tropospheric component of ozone). The study of seasonal patterns and trends is another potential application (Loyola et al., 2010).

The recently released upgrades (S5P OCRA/ROCINN version 2, S5P FRESCO version 1.4) were not the main focus of this paper as there has been not yet a reprocessing of the full time series. Moreover, other improvements in the cloud products are foreseen in upcoming version releases. These new data versions should be validated with the same system as used in the current paper, allowing the necessary independent assessment of the S5P data product evolution.

*Data availability.* Sentinel-5p CLOUD OCRA/ROCINN RPRO (reprocessed) and OFFL (offline) data 1.1.7-1.1.8 can be obtained from the Sentinel-5P Pre-Operations Data Hub (https://s5phub.copernicus.eu/dhus/). Sentinel-5p FRESCO data files are not publicly available, but the FRESCO cloud properties are available in Sentinel-5p $NO_2$ data files, also at the Sentinel-5P Pre-Operations Data Hub. Cloudnet data is available from http://cloudnet.fmi.fi/ or from EVDC (ESA Atmospheric Validation Data Centre, https://evdc.esa.int/). Aqua MODIS MYD08_D3 data can be obtained from https://ladsweb.modaps.eosdis.nasa.gov. Aura OMI OMCLDO2 data can be obtained from https://disc.gsfc.nasa.gov/datasets/OMCLDO2_003/summary.

*Author contributions.* SC and AA carried out the global validation analysis, with support from RL and MS. JCL, DH, AK and TV contributed input and advise at all stages of the analysis. AA, DL, RL and FR developed the OCRA/ROCINN retrieval algorithms and the corresponding operational UPAS processors for TROPOMI, GOME-2 and GOME at DLR. MS, PS and PW developed the TROPOMI and OMI cloud data processors at KNMI. AMF and EOC post-processed Cloudnet data tailored to S5P validation and contributed ground-based scientific expertise. All authors revised and commented on the manuscript.

*Competing interests.* The authors declare that they have no conflict of interest.

*Acknowledgements.* Part of the reported work was carried out in the framework of the Copernicus Sentinel-5 Precursor Mission Performance Centre (S5P MPC), contracted by the European Space Agency (ESA/ESRIN, Contract No.4000117151/16/I-LG) and supported by the Belgian Federal Science Policy Office (BELSPO), the Royal Belgian Institute for Space Aeronomy (BIRA-IASB), the Netherlands Space Office (NSO), and the German Aerospace Centre (DLR). Part of this work was also supported by the S5P Validation Team (S5PVT) AO project CHEOPS-5p (ID #28587, Co-PIs J.-C. Lambert and A. Keppens, BIRA-IASB) with national funding from the BELSPO/ProDEx project TROVA-E2 (PEA 4000116692). The authors express special thanks to B. Langerock, J. Granville, S. Niemeijer and O. Rasson for post-processing of the network and satellite data and for their dedication to the S5P operational validation.

   This work contains modified Copernicus Sentinel-5 Precursor data (2018-2020) processed by DLR and KNMI, and post-processed by BIRA-IASB. We acknowledge the ACTRIS RI and the ACTRIS-2 project (European Commission contract H2020-INFRAIA, grant no. 654109) for providing the ground-based data from the Cloudnet sites in this study, which was produced by the Finnish Meteorological Institute, and is available for download from http://cloudnet.fmi.fi/. The cloud radar, ceilometer and

microwave radiometer data for the ARM sites used in this study (Graciosa and Villa Yacanto) were obtained from the Atmospheric Radiation Measurement (ARM) user facility, managed by the Office of Biological and Environmental Research for the U.S. Department of Energy Office of Science. The cloud radar, ceilometer and microwave radiometer data for the Summit Station were obtained from NOAA; overall programmatic and logistical support for was provided by the US National Science Foundation, with additional instrumental support provided by the NOAA Earth System Research Laboratories, the DOE Atmospheric Radiation Measurement

Program, and Environment Canada. We warmly thank the PIs and staff at all stations for their sustained effort in maintaining high quality measurements and for valuable scientific discussions.

   We also thank Steven Platnick and co-workers for the provision of prototype 2018 files of NPP-VIIRS to DLR. The Aqua/Modis Aerosol Cloud Water Vapor Ozone Daily L3 Global 1Deg CMG dataset (http://dx.doi.org/10.5067/MODIS/MYD08_D3.061) was acquired from the Level-1 and Atmosphere Archive & Distribution System (LAADS) Distributed Active Archive Center (DAAC),

located in the Goddard Space Flight Center in Greenbelt, Maryland (https://ladsweb.nascom.nasa.gov/).

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
