# Peer review of "Validation of the Sentinel-5 Precursor TROPOMI cloud data with Cloudnet, Aura OMI O2-O2, MODIS and Suomi-NPP VIIRS"

_Atmospheric Measurement Techniques, 2020_

## Referee Comment (RC1) · Anonymous Referee #1 · 29 Jul 2020

This paper presents an evaluation of cloud products retrieved using the Sentinel-5 Precursor TROPOMI instrument. Cloud properties from the Cloudnet, Aura OMI O2-O2, MODIS and Suomi-NPP VIIRS datasets are used for this evaluation. The TROPOMI cloud products are mainly used to correct of filter trace gas retrievals that are performed with TROPOMI, but it is useful to evaluate the products with more established datasets. The paper is of interest for AMTD.

The paper is generally well written and structured, although information is somewhat scattered. I recommend publication after the general and specific comments listed below are addressed.

[Figure]

General comments:

1)The different algorithms and datasets are described throughout the paper. Information is somewhat scattered through the introduction, section 2 and section 3. Also, some information is given for one algorithm but not for others. I suggest merging this information in a more consistent manner in a single section. For example, quite specific information on the FRESCO algorithm on the adjustment of the spectral resolution of the database is in the introduction, but is better placed in a section, while it also raises the question how/if these wavelengths shifts are handled in the OCRA/ROCINN.

2) A distinction is made between the radiative cloud fraction (RCF) and the scaled RCF (sRCF). However, the RCF should be better defined in the paper. If I understand correctly, the RCF generally is the cloud fraction that leads to the observed reflectance under the assumption of a certain cloud reflection. If the assumed cloud reflection is the correct one then the RCF is equal to the geometric cloud fraction. For FRESCO it is clear what the assumed cloud reflection is (0.8), but it is less clear for OCRA/ROCINN. From the literature I get the impression that OCRA/ROCINN is capable of determining whether a pixel is fully cloudy ('white') or not. But for fractional clouds, I suspect it is biased towards assuming a high cloud reflectance similar to FRESCO. That is also supported by the RCF in Figure 5 being very similar to the sRFC in Fig 6, which are both similarly low-biased compared to VIIRS cloud fraction. The histograms in Fig. 8 are filtered to have "only pixels with a VIIRS geometrical cloud fraction, which originates from a cloud mask, above 0.9 contributed to the comparison." Although this sentence is unclear (see specific comments), my interpretation is that any TROPOMI pixel is included if it is 90% covered by cloud as determined by VIIRS. As mentioned above, these are situations where OCRA/ROCINN is probably doing a reasonable job on identifying a high cloud fraction and retrieving the cloud optical thickness. It masks the performance of determining cloud optical depths for low cloud fractions. Please discuss the interpretation of the retrieved RCF and optical thickness for cases with fractional cloud cover. A 2D histogram of retrieved cloud optical thickness (or albedo)

and radiometric cloud fraction of all the OCRA/ROCINN results would be helpful.

Specific comments:

Page 2, line 5: Remove the extra periods after "(SO2)". Page 2, line 27-28: The two references are both relevant for using the PMDs but no reference for using the UV spectral measurements is given. A suggestion is "Van Diedenhoven, B., O.P. Hasekamp, and J. Landgraf, 2007: Retrieval of cloud parameters from satellite-based reflectance measurements in the ultraviolet and the oxygen A-band. J. Geophys. Res., 112, D15208, doi:10.1029/2006JD008155."

Equation 1: In the statement about "if fc=0", I guess that should be "if f_{rc}=0", but this follows from the main equation so seems not needed.

Equation 2: The other algorithms are using O2 absorption lines and thus are also expected to retrieve cloud (top) pressure instead of height. I guess the conversion represented by Eq. 2 is already done in these products. Please discuss in the paper.

Page 9, line1 11-13: The sentence starting with "For scenes with clouds" is very fragmented and hard to follow. I suggest rewriting.

Page 10, line 27-31: I am a bit confused what is meant with "pixel" in these sentences. Do you mean "altitude bin" or "range gate" of the radar/lidar measurements or collocated satellite pixel. I suggest rewriting to make this clearer.

Page 17, line 16-17: It is stated that "only pixels with a VIIRS geometrical cloud fraction, which originates from a cloud mask, above 0.9 contributed to the comparison". Are the pixels in this sentence TROPOMI pixels? That is, are only (near) fully cloud covered TROPOMI pixels included? Although the reason given is to reduce artifacts, it is quite a limitation of the comparisons as the performance of OCRA/ROCINN to retrieve optical thickness and height for fractional cloud cover is not evaluated.

Page 17, line 25: An effective COT is introduced. Please make sure to identify where this effective COT is used. Are the histograms in Fig 9 showing the effective COT from

TROPOMI? If so, I think that important detail is easily missed (If so, it was by me.). If so, how does Fig. 9 look like without that scaling? Also, I would think that this scaling might work with an albedo, but not with COT, since reflectance scales highly non-linear with COT. Please discuss in the paper.

Page 17, lines 28-30: Please check the wavelengths mentioned here. VIIRS does not have a 2.16 micron band, so I am quite sure that should be 2.25 micron. Also, I would think the optical thickness is inferred from 0.67 micron over land and 0.86 over ocean.

Page 19, line 8: "Aggregated" should be "aggregates".

Page 19, line 22: Looking at table S2, I notice that the mean CTH determined by TROPOMI for high clouds is often lower than the threshold for high clouds. I guess the classification is done using the VIIRS cloud properties only. If so, please state this in the paper or otherwise explain. I also suggest moving table S1 in the main text.

Page 21, Fig 9: This is a very complicated and busy figure. I had some trouble understanding what the x-axis represents, so please explain this better. I still do not understand how the CLOUDNET classes are represented with the pink, white and blue colors. I think a bit more text explaining all the different and dots would be helpful.

Page 23, line 14: What are "zero offset clouds"?

---

## Referee Comment (RC2) · Anonymous Referee #2 · 8 Aug 2020

This paper analyzes properties of clouds derived from TROPOMI measurements and conducts a validation exercise comparing them to retrieval inferred with sensors and algorithms based on different physical approaches.

The objectives, methods used and the results must be considered as provisional, as the described algorithms seem not yet mature enough, as often declared by the authors themselves throughout the presentation of the results.

Many improvements are undergoing and newer reprocessing of the records are to be expected in the near future. As such, the paper documents the ongoing effort to make TROPOMI cloud data reliable.

[Figure]

The science is sound and the text is well written. However, the paper still needs moderate-to-major revisions. I therefore recommend publication only after my comments are addressed.

I will collapse all my remarks in one general comment below and leave some minor comments later on.

Main general comment

- First and foremost, the present paper is about clouds from TROPOMI. Correction of trace gas retrievals is only one of the many applications.

Even if the Sentinel-4 and Sentinel 5 missions, extending the Sentinel 5P record with the same algorithms, share the obvious goal of monitoring atmospheric composition, cloud research is unfortunately shadowed and left in the background.

Let us immagine a data user who wants to conduct own cloud search using European data sets instead of American data sets, and specifically TROPOMI data. Is he sufficiently informed about the range of applicability of the retrievals for cloud research itself? I do not think so.

Therefore, I find it misleading to begin the introduction by dedicating the entire first paragraph to past missions endeavouring the study of atmospheric constituents only. I understand the logic, but I find it overkill. Clouds are firstly mentioned only at line 19.

This does not mean that the paragraph should be removed, but at the end of the introduction I expect a paragraph of equal importance and length that would enable the reader to judge whether TROPOMI's cloud data can do the job for research objectives as: cloud trends, aerosol-cloud interactions, climatology generation, hydrological cycle, climate extremes, input for aviation safety (is a radiative cloud height, or centroid, of any importance to aviation? No. Cloud top is.).

It looks like that the first comparison of TROPOMI cloud algorithms has been already reported in the TROPOMI S5P Science Verification Report (S5P-SVP). As far as I can

judge comparing that outcome with that of this paper, the main conclusions are the same. But some methodological approaches deployed for the S5P-SVP, delivering valuable insights, have not been followed up here (e.g., across-track errors, surface errors as function of low cloud fractions, three-dimensional RT).

So, I invite the authors to elaborate and make explicit the following algorithmic aspects:

- Errors arising from a plane-parallel approximation (neglection of 3D RT). We know that the improved spatial resolution does have an impact on RT. We are not talking about GOME-2 or SCIAMACHY anymore. Please also link your results to those of the MICRU algorithm when talking about cloud fraction.

- Errors arising from the neglection of cloud multi-layeredness. Are the algorithms capable to flag this? Can a data user expect to be able to use TROPOMI data to investigate turbulent atmospheres? Joiner et al. (2010) shows that the fraction of multi-layer cloudy pixels can be up to 50% or more at OMI spatial resolution.

- Vertical inhomogeneity of clouds. To what extent are the presented algorithms capable to follow it? Can they be improved to encapsulate different vertical extinction profiles? Will the algorithms be able to reproduce cloud distributions inferred, e.g., from CloudSAT? Ziemke et al., 2009 show that average cloud extinction profiles for tropical deep convective clouds that peak at different pressures depending in general on the total optical thickness. This implies that the ISCCP diagrams from TROPOMI can not be fully reliable because of the following remark:

- Are the CTH/CH retrievals dependent on COT? This is a matter of great concern for cloud research. This aspect is hastily mentioned by the authors only once, but it got my attention. this has to be read and understood in connection with bullet (2) by the first reviewer, which I support. Please, elaborate and make explicit.

- Surface influence. Looking at the S5P-SVP, Figures 13.28 and 13.29, pages 285-286, there is a clear CH-dependency on surface reflectivity and cloud fraction. In the

present paper only at P23, bias dependence on surface reflectivity is mentioned. So, I appreciate a similar exercise, where the accuracy of CH is subset after surface reflectivity and cloud fraction. It can be done within a Taylor diagram or by other means, but this interdependency must be made explicit.

- Across-track dependence of cloud retrievals. The authors are encouraged to compare their results with those of Fasnacht et al., 2019. Fasnacht et al., A geometry-dependent surface Lambertian-equivalent reflectivity product for UV-Vis retrievals – Part 2: Evaluation over open ocean, Atmos. Meas. Tech., 12, 6749–6769, 2019.

Minor comments:

P3 L18: "used by FRESCO". There should also be "... and ROCINN"?

P5 L5: "Note that at maximum RCF and CA, sRCF reaches 1.2 rather than 1." Do the authors have an explanation why a CF must exceed the limit of 1? Clearly the value is not physical. So, please, elaborate and make explicit that a CF=1.2 is needed as an ad-hoc correction for surface and/or trace gas retrieval.

P8 L28: "Due to the difference in overpass time between GOME-2 (in the morning) and Sentinel 5 precursor (in the afternoon)" Why is the overpass time a source of discrepancy for the surface albedo climatology? I can understand the difference in footprint size, but not a difference of some hours when building a climatology of an object barely evolving within few hours.

For the spatial resolution, it is not clear to me why the GOME-2 climatology is used and not the MERIS black-sky albedo climatology, which would be a much better choice. Please, elaborate, make explicit and justify.

P13 L2 and ff: Section 4.1.2 S5P FRESCO. I find this section unnecessary in the context of this manuscript. The purpose of the paper is to present a validation and comparison between different cloud products derived from TROPOMI measurements. However, in this section, the inability of FRESCO to discriminate aerosols from clouds

is presented. This result, besides not being new (Wang et al, ACP, 2012), is not surprising given the spectral range used by the algorithm which handles cloud and aerosol radiances similarly.

Moreover, the authors swiftly interchange between clouds and aerosols in the narrative and this is inconsistent: line 9 should read "For this new product, the sensitivity to low ___aerosols___ in the low atmosphere is improved" and not "the sensitivity to ___low clouds___ is improved".

This is because based on the very same evidence provided by the authors themselves in the paper you are not retrieving clouds.

So wouldn't it be better to filter out all those pixels that are reasonably aerosol from the data set? I would like to stress that, although one of the possible applications of these data sets is the correction for trace gas retrievals, a cloud data set should serve cloud research too. What if a data user is going to average and assess long-term tendencies or climatology of cloud properties? How much of such missflagged aerosols will be present in the record?

Additionally, please collapse all FRESCO algorithmic details in one section, as pointed out also by the first referee.

References

Joiner, J., Vasilkov, A. P., Bhartia, P. K., Wind, G., Platnick, S., and Menzel, W. P.: Detection of multi-layer and vertically-extended clouds using A-train sensors, Atmos. Meas. Tech., 3, 233–247, 2010.

Wang P, Tuinder ONE, Tilstra LG, De Graaf M, Stammes P (2012) Interpretation of FRESCO cloud retrievals in case of absorbing aerosol events. Atmos Chem Phys 12(19):9057–9077, doi:10.5194/acp-12-9057-2012

Sentinel-5P TROPOMI Science Verification Report, S5P-IUP-L2-ScVR-RP, Issue 2.1", Sect. 13.4-14.4, https://earth.esa.int/documents/247904/2474724/Sentinel-5P-

TROPOMI-Science-Verification-Report, 2015

Ziemke, J. R., Joiner, J., Chandra, S., Bhartia, P. K., Vasilkov, A., Haffner, D. P., Yang,K., Schoeberl, M. R., Froidevaux, L., and Levelt, P. F.: Ozone mixing ratios inside tropical deep convective clouds from OMI satellite measurements, Atmos. Chem. Phys., 9, 573–583, 2009.

Fasnacht et al., A geometry-dependent surface Lambertian-equivalent reflectivity product for UV-Vis retrievals – Part 2: Evaluation over open ocean, Atmos. Meas. Tech., 12, 6749–6769, 2019.

---

## Referee Comment (RC3) · Anonymous Referee #3 · 9 Aug 2020

This paper compares different OCRA/ROCINN and FRESCO cloud products to several auxiliary satellite data from other sensors as well as ground based data. It furthermore describes the ongoing efforts to improve the reliability of both official product branches. The content of the paper is clearly in line with the topics of AMT. Publication, however, is only encouraged after the following comments and those detailed by the Referees #1 and #2 are considered.

Major comments:

1) Actually, I got the impression to read two merged papers. On the one hand, the actual operational algorithms are described and somehow verified – validation without

knowing the actual truth may be exaggerating - and, on the other hand, significant problems of the algorithms are identified and possible fixes are presented. What I miss is the link: Why are the future algorithm versions that are proposed to fix many issues of the actual algorithms not included in the verification exercises? From the manuscript as is, I get the impression that this paper is actually an algorithm presentation of somehow improved algorithms within a paper extensively using outdated data.

In order to bridge this gap, I would like to suggest to options: a) Split the paper in two, one verifying the actual operational algorithms and another one (or two for both algorithm branches each) introducing the future versions of the algorithms – then with a verification as well. b) Treat both algorithm versions (actual and future) of both OCRA/ROCINN and FRESCO similarly in the verification section so that an actual user may judge for himself either to use what is already available or to wait for an algorithm update being applied in the future without knowing, when this will be.

2) I found the structure of manuscript quite confusing. Descriptions of the algorithms appear at several locations. I suggest to first introduce all data, then describe the conducted studies and finally discuss the results.

Minor comments

1) The abstract contains many acronyms, which are not described at their first appearance, e.g. the difference between ROCINN_CRB and ROCINN_CAL are not clear from the beginning. I sugest to restructure the whole abstract. Maybe it is possible to collapse both ROCINN branches to one. Furthermore, the outlook on future mitigations (page 1, line 8) should be moved to the end the abstract as suggested in the guidelines.

2) Please add more references: page 2, line 6: for the trace gas products, please also add H2O page 2, line 16: Please add reference and name pollution page 2, line 28: both HICRU and MICRU are also using PMD data, please add references below page 3, line 7: Please add a reference to the FRESCO algorithm applying "a directional surface albedo". I could not find this feature described in the cited references. page

3, line 13: Ref to FRESCO-S page 4, line 10: Reference to OCRA page 7, line 17: Reference to OCRA page 12, line 2: Are these "geographical patters" also discussed in Lutz et al., 2016? Please add a reference to the S5P verification report.

3) Please homogenize the acronyms and formulae. For example, O2 is sometimes written with subscript and sometimes not. Also TROPOMI/S5P is not consistently spelled. Sometimes its lower case and sometimes it is in caps.

4) page 3, line 2: What about GEMS and TEMPO? It what respect is S4 the first of its kind?

5) page 7, line 27: The choice of resolution seems a bit arbitrary. Please provide a rationale for both. Page 7, line 28: What is the rationale behind taking monthly mean reflectance data.

6) Please include a description of the treatment of snow and ice surfaces in the description of the OCRA/ROCINN and FRESCO.

7) page 9, line 3: "no official documentation is currently available for FRESCO-S" If this is true, I strongly would like to encourage the co-authors, that are developing this product, to provide official documentation to fill this gap for a potential user of the product.

8) page 16, line 9 and following: It seems to me that the statistics of the different data sets are based on different subsets (eg. MODIS versus the others). I strongly suggest to use the same subset to compute statistics in order to avoid biases.

9) page 17, line 27: Would it be possible to asses the effectiveness of this "compensation"?

10) Figure 9: These plots are really not easy to perceive. A particular feature I would like to have discussed is that there seems to be a significant number of C(T)H=0 values for the S5P algorithms as opposed to the OMI data in the upper left plot, which does not show not a single zero reading. What can be the reason behind this behaviour?

Furthermore, why are there significantly more CH0=0 values than CTH=0 values for ROCINN_CAL?

11) The conclusions (page 25) start off with two statements what will be better in the future. Please move these statements to the end of the discussion (compare major comment 1).

Specific comments

page 1, line 7 "were" → "are"

page 2, line 33 omit "easy"

page 3, line 4 omit "fast"

page 3, line 30: NPP-VIIRS etc. are instruments, not satellites

caption of figure 1: omit ", while S5P FRESCO is merely a backup"

page 4, line3: omit "finally"

page 4, line 5: please rephrase this sentence

table 1: the superscrips are appearing at an odd order: b, d, a, c → please sort please also add f_{rc,0.8} to that table

page 5, line 4: A verb is missing somewhere after "In general"

page 5, line 6 and following: please add a short comment on the rationale behind 0.8 as the fixed cloud albedo

page 5, line 16: please use metres to avoid confusion: 7668m

table 2: please sort superscripts please add column with references

page 6, line 6: omit "among else"

page 6, line 8: omit the entire sentence

page 7, line 1: 20% bias on cloud fraction is a lot, please discuss the possible influences on the validation exercises in the paper

page 8, line 2: What is a significant set? How many did you use? Please omit "smart"

page 8, line 8: "spherical particles" → "spheres"

page 8, line 12: What is a "GE_LER"? Please specify.

Page 8, line 14: "RPRO" is not introduced yet

page 10, line 2: Please rephrase so that it easier to perceive, that there are two MODIS instruments of different platforms.

Page 10, line3: omit "from north to south" and "from south to north" as this is redundant information

page 10, line 12: omit "Dutch-Finnish"

page 10, line 13: omit "NASA's"

page 10, line 29: please specify what a "pixel" denotes in the context of a ground based measurement in order to avoid confusion with a "satellite pixel"

page 11, line 6: "much less" → please be more specific

page 11, line 14: Why is there a shift in tenses? "were" → "are"

figure 3: Please inducate the "sharper contrast" in the figures

page 13, line 12 ad following: Please also discuss here, that aerosols may as well have a different impact on the RT than clouds.

Figure 4: Please also discuss why there are steps (depending on the row) in the lower right figure. Is this an artifact/interference or a signal? This is critical, because right now it seems as an error in the proposed FRESCO-A wide algorithm.

Page 15, section 4.2: Which version of OCRA/ROCINN is applied? Is this version 1 or

the proposed future version? If it is version 1, please add a similar figure for version 2 (see major comment 1)

page 17, line 10: omit "as it was stated earlier" → So why state it again?

Figure 7: If this is regridded VIIRS data, I would like to suggest to also show a scatter plot (2D histogram) in order to support a more quantitative comparison.

Figure 8: I guess "TROPOMI" indicates OCRA/ROCINN. Please be more specific in order to avoid confusion with FRESCO. How would these plots look for version 2?

page 20, line 12: Please provide a rationale, why also OMI data are included in this study.

Figure 11: Please improve image quality.

Page 25, line 23: Please be more quantitative.

References

Grzegorski, M., Wenig, M., Platt, U., Stammes, P., Fournier, N., and Wagner, T.: The Heidelberg iterative cloud retrieval utilities (HICRU) and its application to GOME data, Atmos. Chem. Phys., 6, 4461–4476, https://doi.org/10.5194/acp-6-4461-2006, 2006.

Lutz, R., Loyola, D., Gimeno García, S., and Romahn, F.: OCRA radiometric cloud fractions for GOME-2 on MetOp-A/B, Atmos. Meas. Tech., 9, 2357–2379, https://doi.org/10.5194/amt-9-2357-2016, 2016.

Sihler, H., Beirle, S., Dörner, S., Gutenstein-Penning de Vries, M., Hörmann, C., Borger, C., Warnach, S., and Wagner, T.: MICRU background map and effective cloud fraction algorithms designed for UV/vis satellite instruments with large viewing angles, Atmos. Meas. Tech. Discuss., https://doi.org/10.5194/amt-2020-182, in review, 2020.

---

## Referee Comment (RC4) · Anonymous Referee #2 · 9 Aug 2020

The third reviewer touches on two aspects that in my comments have probably remained implicit and have not been sufficiently highlighted.

The first aspect is the accuracy of retrieval for very reflective surfaces. My request to subdivide and categorize biases in function of cloud fraction and surface reflectivity aims to understand two things: (1) what happens for very low CF (2) what happens for very high SA (3) where the algorithms start diverging in performance.

The second is the compactness and clarity of the product naming, having in mind the usability of the data for the typical user. I would appreciate in the conclusions clear and concise guidance on which products to use, for which purposes, and which not. I think

the authors and the production teams should make an effort in this direction.

A reason for further uncertainty is the announcement of new products that would emerge from minimal adjustments of the algorithm. I do not think it is necessary to deem FRESCO-S a new product, only if you change the spectral range within O2A by a few nm. This is just a source of confusion for the reader and certainly cannot be considered a milestone for a typical algorithm development chain.

---

## Author Response (AR1)

**(1.1) This paper presents an evaluation of cloud products retrieved using the Sentinel-5 Pre-cursor TROPOMI instrument. Cloud properties from the Cloudnet, Aura OMI O2-O2, MODIS and Suomi-NPP VIIRS datasets are used for this evaluation. The TROPOMI cloud products are mainly used to correct of filter trace gas retrievals that are performed with TROPOMI, but it is useful to evaluate the products with more established datasets. The paper is of interest for AMTD.**

**The paper is generally well written and structured, although information is somewhat scattered. I recommend publication after the general and specific comments listed below are addressed.**

**Reply:** We thank the reviewer for the positive overall appreciation. Please find our answers and planned actions (including actions to better structure the information) below.

**General comments:**

**(1.2) 1) The different algorithms and datasets are described throughout the paper. Information is somewhat scattered through the introduction, section 2 and section 3. Also, some information is given for one algorithm but not for others. I suggest merging this information in a more consistent manner in a single section. For example, quite specific information on the FRESCO algorithm on the adjustment of the spectral resolution of the database is in the introduction, but is better placed in a section, while it also raises the question how/if these wavelengths shifts are handled in the OCRA/ROCINN.**

**Reply:** Indeed, technical information on the different data products is now in different sections. We will make this more consistent.

Regarding the question of the handling of wavelengths shifts in OCRA/ROCINN: The wavelength shift is one element of the state vector and it is fitted in the OCRA/ROCINN algorithm. Reference and explanation about the wavelength shift fitting will be added to the data set description. The following sentence will be introduced: "Note that during the inversion a wavelength shift for the earthshine spectrum is fitted additionally (Loyola et al., 2018)."

**Planned Action:** We will (i) move technical product information from the introduction to the 'description of the datasets' (old section 3) and (ii) merge information from the old sections 2 and 3. Furthermore, as answer to Reviewer 3 (comment 3.2), differences between major product releases (OCRA/ROCINN CLOUD version 1 and version 2; FRESCO version 1.3 and 1.4) are now more clearly separated in the text, and it is made clear that the versions with the longer data records (CLOUD version 1, FRESCO version 1.3) are the target of the bulk of the analysis. Next, as a response to many aspects raised by reviewer 2, which are interesting but mostly out of scope for this work, we introduce a new section where previous assessments of the cloud algorithms are shortly discussed. Finally, we introduce a new section about the comparability between cloud properties of different cloud products. We outline this in more detail below.

(i) Move technical product information from **Introduction** to **Description of the datasets**

1.a P. 3, line 10-16. "Due to the increase in the spectral resolution [...] response function." The information here will be moved to the FRESCO dataset description.

1.b P. line 17-25 "The OMI instrument … ". The information here will be moved to and merged with the OMCLDO2 dataset description.

(ii) Merge technical product information from section 2 and 3.

We will implement the following structure

- **2. Description of the datasets**

  **2.1 Overview of cloud data products, properties and related terminology**

  -Table 1, 'properties, abbreviation and mathematical symbol' will stay here.

  -General definition of radiometric cloud fraction. (See answer to comment (1.2))

  -General definition of effective cloud height

  **2.2 Satellite data sets**

  **2.2.1 S5P TROPOMI CLOUD OCRA/ROCINN**

  -more rigorous explanation of processing modes (NRTI, OFFL, RPRO) and processor versioning

  -description of processor version 1 (most content of old section 3.2.1)

  -handling of wave length shifts in OCRA/ROCINN

  -Explain that ROCINN_CAL and ROCINN_CRB are disseminated in the same S5P CLOUD files (from old section 2).

  -changes introduced in processor version 2.

  -We also make clear that processor version 1 (the only version of which a full +2-year record is available) is the target of the bulk of the analysis in this work, and that for version 2 only some major impacts are demonstrated.

  **2.2.2 S5P TROPOMI FRESCO-S**

  -more rigorous explanation of processing modes (NRTI, OFFL, RPRO) and processor versioning

  - description of processor version 1.3 (most content of old section 3.2.2)

  -product information from old introduction

  - changes introduced in processor version 1.4.

  - We also make clear that processor version 1.3 (the only version of which a full 2.5-year record is available) is the target of the bulk of the analysis in this work, and that for version 1.4 only some major impacts are demonstrated.

  **2.2.3 Suomi NPP-VIIRS**

  -content of old section 3.2.3

  **2.2.4 Aqua MODIS** (Add platform "Aqua" in the title)

  -content of old section 3.2.4

  **2.2.5 Aura OMI OMCLDO2**

  -content of old section 3.2.5

  -product information from old introduction

  -At the end of this section, the derivation of the OMCLDO2 cloud height (from old section 2, p. 5 line 15).

  **2.3 Ground-based datasets**

  **2.3.1 CLOUDNET**

  -content of old section 3.3.1

  -At the end of this section, the derivation of the 'CLOUDNET cloud mean height (CMH)' (moved from the old section 2, p. 6, line 4-7.)

  **2.4 Note on previous assessments of OCRA/ROCINN and FRESCO algorithms**

  (new section, as action to comment 2.3 and following)

  These will be mainly short descriptions of the limitation, with reference to S5P-SVP (as suggested by reviewer 2) and other works.

  - Plane-parallel approximation (cloud shadow,…)

  - VZA dependence of geom CF for vertical extended clouds

  - Multilayer. Here we will refer to the OCRA/ROCINN_CAL and _CRB tests with double layer simulations of Loyola et al. (2018) and the paper of Joiner (2010).

  - Vertical inhomogeneity

  - across-track dependence of cloud retrievals

**2.5 Notes on intercomparability of cloud properties** (new section)
- Discuss why the different properties are not equivalent.
-- Geometrical CF vs radiometric CF
-- Radiometric CF: different model assumptions
-- Different wave length ranges of OCRA vs FRESCO, OMCLDO2
-- Different cloud height sensitivities due to different wave length or different cloud model (Lambertian vs Mie).
- Introduce the derived property 'scaled radiometric cloud fraction' which can be compared between FRESCO, OMCLDO2, and OCRA/ROCINN_CRB.
- Motivation why an 'effective COT' is needed to enable TROPOMI and VIIRS comparisons (see also comment 1.12)

Discussion of the mission requirements (currently in the old section 2) will be moved to the new section 3.

*Summary of changes (pages and lines refer to revised manuscript):*

- *Technical information on FRESCO-S and on OMI moved from introduction to 'Description of the data sets' (new section 2).*
- *Merge technical product information from old sections 2 and 3 in new section 2 'Description of the data sets'.*
- *Info on wavelength shift OCRA-ROCINN now in section 2.2.1. Page 9, line 17.*
- *Discussion on mission requirements in new section 3.*

**(1.3) 2) A distinction is made between the radiative cloud fraction (RCF) and the scaled RCF (sRCF). However, the RCF should be better defined in the paper. If I understand correctly, the RCF generally is the cloud fraction that leads to the observed reflectance under the assumption of a certain cloud reflection. If the assumed cloud reflection is the correct one then the RCF is equal to the geometric cloud fraction.**

**Reply:** The explanation of the concept of radiometric cloud fraction could indeed have been clearer. We will introduce the following definition in the new section 2.1.

"The RCF is not the geometric cloud fraction of the true cloud, but can be defined as the fraction that has to be attributed to the model cloud to yield (in combination with non-cloud reflectance contributions) a TOA reflectance that agrees with the observed reflectance.

In OCRA, the clear-sky (RCF=0) reflectance is taken from composite maps created from satellite measured reflectances and the fully cloudy (RCF=1) reflectance is defined as 'white' in the color diagram. OCRA then determines the radiometric cloud fraction using the differences between the reflectance (defined as colors in OCRA) of a measured scene and its corresponding clear-sky values.

In FRESCO, the radiometric cloud fraction is the cloud fraction value which, in combination with the assumed cloud albedo and the input surface albedo, yields a TOA reflectance that agrees with the observed reflectance."

*See page 6, line 11 in revised manuscript.*

**(1.4) For FRESCO it is clear what the assumed cloud reflection is (0.8), but it is less clear for OCRA/ROCINN.**

**From the literature I get the impression that OCRA/ROCINN is capable of determining whether a pixel is fully cloudy ('white') or not. But for fractional clouds, I suspect it is biased towards assuming a high cloud**

**reflectance similar to FRESCO. That is also supported by the RCF in Figure 5 being very similar to the sRFC in Fig 6, which are both similarly low-biased compared to VIIRS cloud fraction.**

**Reply:** The UV/VIS spectrometer data from TROPOMI are usually less sensitive to optically very thin clouds, which might be easier detectable with imager data like VIIRS that also include bands in the infra-red. Furthermore, a geometric cloud fraction is independent of the cloud optical thickness while the radiometric OCRA cloud fraction correlates with the cloud optical thickness. If the radiometric OCRA cloud fraction for fractional clouds over- or underestimates the geometric cloud fraction in individual cases strongly depends on the cloud optical thickness. On a global average however, the radiometric cloud fraction was found to be low-biased compared to geometric cloud fractions by about 0.15-0.20.

**Planned action**: The difference between the geometrical cloud fraction (as from VIIRS) and the radiometric cloud fraction from S5P OCRA is now described in more detail in the text.

*See page 6, line 9 and following, on the difference between geometrical and radiometric cloud fraction, and how radiometric cloud fraction is defined for OCRA.*

**(1.5) The histograms in Fig. 8 are filtered to have "only pixels with a VIIRS geometrical cloud fraction, which originates from a cloud mask, above 0.9 contributed to the comparison." Although this sentence is unclear (see specific comments 1.12), my interpretation is that any TROPOMI pixel is included if it is 90% covered by cloud as determined by VIIRS. As mentioned above, these are situations where OCRA/ROCINN is probably doing a reasonable job on identifying a high cloud fraction and retrieving the cloud optical thickness. It masks the performance of determining cloud optical depths for low cloud fractions. Please discuss the interpretation of the retrieved RCF and optical thickness for cases with fractional cloud cover. A 2D histogram of retrieved cloud optical thickness (or albedo) and radiometric cloud fraction of all the OCRA/ROCINN results would be helpful.**

**Reply:**

(i) Regarding the unclarity in the text: Actually, TROPOMI pixels are included only if both VIIRS GCF>0.9 and TROPOMI OCRA RCF>0.9. This will be made clear in the text.

(ii) Regarding the motivation behind the pixel selection. The comparison between TROPOMI and VIIRS makes most sense for near fully cloudy conditions, i.e., when both VIIRS GCF and OCRA RCF are above 0.9. The reason is as follows. The dominating factor for the ROCINN vs VIIRS comparisons should be the cloud optical thickness, but also cloud fraction will play a role. At low CFs the differences between the geometric and radiometric CFs can become much larger than at the high end of the CF range. For partial cloud coverage the comparisons are meaningful (at least to some degree) only if the optical thickness is large enough. Therefore, we believe that the inter-comparability of the ROCINN and VIIRS cloud parameters (CTH, COT) is best justified under (almost) fully cloudy conditions, because there we expect the least deviations between the geometric and radiometric CFs.

(iii) It is indeed true that the performance of OCRA/ROCINN is optimal for high geometric cloud fractions and optically thicker clouds. The combination of low geometric cloud fraction and optically thin cloud is the most challenging situation for any cloud retrieval algorithm in the UVN. However, given the inter-comparability limitations noted above for lower cloud fractions and the different cloud fraction definitions in general, we consider a deeper analysis beyond the scope of this paper.

**Planned action:**

(i) Reformulate as 'Furthermore, only pixels where both VIIRS GCF>0.9 and OCRA RCF>0.9 contributed to the comparison.' Earlier in the paragraph, we explain how the VIIRS cloud mask is converted to a geometrical cloud fraction.

(ii) Make more clear that the high CF threshold was chosen because there the differences between the geometric and radiometric cloud fractions become smaller and hence an inter-comparison between VIIRS and OCRA can be better justified.

(iii) State that for these conditions the performance of OCRA/ROCINN is optimal, but that an analysis of the more challenging conditions (clouds with low GCF and low COT), where the VIIRS vs OCRA/ROCINN inter-comparison is less justified, is beyond the scope of this paper.

*Summary of changes*

- *(i) clarified pixel selection. Page 20, line 28. 'Furthermore, only pixels where both…'*
- *(ii) Page 20, line 30. 'Also, this high CF threshold…'*
- *(iii) Page 22, line 9. 'It should be noted that the performance of OCRA/ROCINN is optimal for…'*

**Specific comments:**

**(1.6) Page 2, line 5: Remove the extra periods after "(SO2)".**

*Removed. Page 3, line 15.*

**(1.7) Page 2, line 27-28: The two references are both relevant for using the PMDs but no reference for using the UV spectral measurements is given. A suggestion is "Van Diedenhoven, B., O.P. Hasekamp, and J. Landgraf, 2007: Retrieval of cloud parameters from satellite-based reflectance measurements in the ultraviolet and the oxygen A-band. J. Geophys. Res., 112, D15208, doi:10.1029/2006JD008155."**

*Reference Diedenhoven inserted. Page 4, line 5.*

**(1.8) Equation 1: In the statement about "if fc=0", I guess that should be "if f_{rc}=0", but this follows from the main equation so seems not needed.**

**Reply:** Thanks for noting the typo. It should indeed be "f_{rc}=0". While mathematically, the second equation follows from the main equation, algorithmically the second line is important, because otherwise cases where $f_{rc}=0$ and $A_c=NaN$ (as zero cloud fraction implies undetermined cloud albedo) lead to $f_{rc,0.8}=NaN$, while $f_{rc,0.8}=0$ is desired. So we want to keep this second line.

**Planned action:** The typo will be corrected. After the equation we will insert "The last line of Eq. (1) is needed to prevent cases where $f_{rc}=0$ and $A_c=NaN$ would lead to $f_{rc,0.8}=NaN$."

*See Eq. (1) and page 10, line 4.*

**(1.9) Equation 2: The other algorithms are using O2 absorption lines and thus are also expected to retrieve cloud (top) pressure instead of height. I guess the conversion represented by Eq. 2 is already done in these products. Please discuss in the paper.**

**Reply:** We explain this now in the text.

**Planned action:**

(i) In the OCRA/ROCINN dataset description, we add 'After the ROCINN retrieval, cloud (top) pressure is obtained from the retrieved cloud (top) height using ECMWF profiles.'

(ii) In the FRESCO dataset description, we add 'In FRESCO, the basic retrieved quantity is cloud height (in km), which is converted to pressure using the AFGL mid-latitude summer profile. (Anderson et al., 1986).'

*Summary of changes:*

- *(i) See page 9, line 26. 'After the ROCINN retrieval…'*
- *(ii) See page 11, line 11. 'In FRESCO, the basic retrieved quantity is…'*

**(1.10) Page 9, line 11-13: The sentence starting with "For scenes with clouds" is very fragmented and hard to follow. I suggest rewriting.**

**Reply:** We will rephrase the sentence.

**Planned action:** We will insert the following sentence: "For scenes with low clouds, i.e. close to the surface, a height that is even closer to the surface will be retrieved. This also holds for low aerosol layers, since the algorithm does not discriminate between the two types of scatterers. In many cases FRESCO then retrieves the surface height, which is incorrect. This defect …."

*See page 12, line 4. 'For scenes with low clouds…'*

**(1.11) Page 10, line 27-31: I am a bit confused what is meant with "pixel" in these sentences. Do you mean "altitude bin" or "range gate" of the radar/lidar measurements or collocated satellite pixel. I suggest rewriting to make this clearer.**

**Reply:** The term "pixel" is indeed confusing here. "Altitude bin" is a good replacement, thanks for the suggestion.

**Planned action:** Replace "pixel" by "altitude bin".

*See page 14, line 7. '…or absence in each altitude bin'.*

**(1.12) Page 17, line 16-17: It is stated that "only pixels with a VIIRS geometrical cloud fraction, which originates from a cloud mask, above 0.9 contributed to the comparison". Are the pixels in this sentence TROPOMI pixels? That is, are only (near) fully cloud covered TROPOMI pixels included? Although the reason given is to reduce artifacts, it is quite a limitation of the comparisons as the performance of OCRA/ROCINN to retrieve optical thickness and height for fractional cloud cover is not evaluated.**

**Reply:** This sentence was indeed unclear. Only pixels where both VIIRS GCF>0.9 and OCRA RCF>0.9 are selected. See our answer to (1.5). Regarding the motivation for this selection: see also (1.5).

**Planned action:** see answer to (1.5).

**(1.13) Page 17, line 25: An effective COT is introduced. Please make sure to identify where this effective COT is used. Are the histograms in Fig 9 showing the effective COT from TROPOMI? If so, I think that important detail is easily missed (If so, it was by me.). If so, how does Fig. 9 look like without that scaling? Also, I would think that this scaling might work with an albedo, but not with COT, since reflectance scales highly non-linear with COT. Please discuss in the paper.**

**Reply:** We think that the reviewer means Fig. 8 (with the histograms), not Fig. 9. We indicate now in the captions of the figures where the effective COT is used. Instead of adding a histogram with the original COT, we chose instead to extend Fig. S5 (which displays part of an orbit) with an original S5P COT, next to the existing S5P effective COT and the VIIRS COT, to demonstrate that only the S5P effective COT and VIIRS COT are comparable.

**Planned action:**
(i) Add note on effective COT in *Fig. 7, Table 4, Fig. S4, Fig. S6 (revised manuscript)*.
(ii) Add original S5P COT orbit part in *Fig. S3 (revised manuscript)*, next to existing S5P effective COT and VIIRS COT.
(iii) Explanation on why the S5P COT is not directly comparable to VIIRS COT is made clearer. *Page 21, line 5 and following: 'The COT from VIIRS is not comparabile directly….'*

**(1.14) Page 17, lines 28-30: Please check the wavelengths mentioned here. VIIRS does not have a 2.16 micron band, so I am quite sure that should be 2.25 micron. Also, I would think the optical thickness is inferred from 0.67 micron over land and 0.86 over ocean.**

**Reply:** The reviewer refers here to "The NASA VIIRS COT is retrieved simultaneously with the effective particle size from the reflection function at 0.75 μm and 2.16 μm. The reflection function at 0.75 μm is in principle sensitive to the COT and the reflection function at 2.16 μm is sensitive to the effective radius (Nakajima and King, 1990)." The wavelengths 0.75 μm and 2.16 μm are the ones mentioned in (Nakajima and King, 1990). However, this two-channel retrieval method can be applied with slightly different combination of wavelengths. So indeed, for VIIRS, the exact wavelength combination in use is 2.2 μm and either 0.65 μm, 0.86 μm or 1.24 μm depending on the surface type. We will clarify this inconsistency in the text.

**Planned action:** We will make the necessary correction on the wavelength combination in the text.

"As demonstrated by Nakajima and King (1990), the reflection function at 0.75 μm is in principle sensitive to the COT and the reflection function at 2.16 μm is sensitive to the effective radius, but this two-channel retrieval method can also be applied with slightly different combination of wavelengths. In the case of VIIRS, the exact wavelength combination in use is 2.2 μm and, depending on the surface type, either 0.65 μm, 0.86 μm or 1.24 μm."

*See page 22, line 3. 'As demonstrated by …'*

**(1.15) Page 19, line 8: "Aggregated" should be "aggregates".**

*Corrected. See page 24, line 1.*

**(1.16) Page 19, line 22: Looking at table S2, I notice that the mean CTH determined by TROPOMI for high clouds is often lower than the threshold for high clouds. I guess the classification is done using the VIIRS cloud properties only. If so, please state this in the paper or otherwise explain. I also suggest moving table S1 in the main text.**

**Reply:** Yes, the classification is done using the VIIRS cloud properties since we consider VIIRS data as a reference for this comparison.

**Planned action:** (i) We add this clarification in the text. (ii) We move the table from the supplement to the main body as suggested.

*See Table 4, with caption 'The classification is done using the VIIRS cloud properties.'*

*See page 24, line 20. ' note that we classify based on…'*

**(1.17) Page 21, Fig 9: This is a very complicated and busy figure. I had some trouble understanding what the x-axis represents, so please explain this better. I still do not understand how the CLOUDNET classes are represented with the pink, white and blue colors. I think a bit more text explaining all the different and dots would be helpful.**

**Reply:** The pink, white and blue colors were used to indicate both the amount of cloudy pixels and the dominant phase (ice or liquid). As this information is not strictly necessary for the main conclusions, and the figure is indeed quite busy, we will remove this information.

**Planned action:** Remove the pink/white/blue background. Replace with grey indicating the cloud occurrence fraction.

*See figure 8 (revised manuscript) and figures supplement S8 to S23.*

**(1.18) Page 23, line 14: What are "zero offset clouds"?**

**Reply:** We meant here clouds with retrieved cloud height equal to the surface altitude.

**Planned action:** we replace this by "cloud heights equal to the surface altitude are obtained."

*See page 29, line 7.*

**Anonymous Referee #2

**(2.1) This paper analyzes properties of clouds derived from TROPOMI measurements and conducts a validation exercise comparing them to retrieval inferred with sensors and algorithms based on different physical approaches.**

**The objectives, methods used and the results must be considered as provisional, as the described algorithms seem not yet mature enough, as often declared by the authors themselves throughout the presentation of the results.**

**Many improvements are undergoing and newer reprocessing of the records are to be expected in the near future. As such, the paper documents the ongoing effort to make TROPOMI cloud data reliable.**

**The science is sound and the text is well written. However, the paper still needs moderate-to-major revisions. I therefore recommend publication only after my comments are addressed.**

**I will collapse all my remarks in one general comment below and leave some minor comments later on.**

**Reply:** We thank referee#2 for the altogether positive evaluation and for the many well-thought suggestions which will improve the quality of the paper.

**Main general comment**

**(2.2) - First and foremost, the present paper is about clouds from TROPOMI. Correction of trace gas retrievals is only one of the many applications.**

**Even if the Sentinel-4 and Sentinel 5 missions, extending the Sentinel 5P record with the same algorithms, share the obvious goal of monitoring atmospheric composition, cloud research is unfortunately shadowed and left in the background.**

**Let us imagine a data user who wants to conduct own cloud search using European data sets instead of American data sets, and specifically TROPOMI data. Is he sufficiently informed about the range of applicability of the retrievals for cloud research itself? I do not think so.**

**Therefore, I find it misleading to begin the introduction by dedicating the entire first paragraph to past missions endeavouring the study of atmospheric constituents only. I understand the logic, but I find it overkill. Clouds are firstly mentioned only at line 19.**

**This does not mean that the paragraph should be removed, but at the end of the introduction I expect a paragraph of equal importance and length that would enable the reader to judge whether TROPOMI's cloud data can do the job for research objectives as: cloud trends, aerosol-cloud interactions, climatology generation, hydrological cycle, climate extremes, input for aviation safety (is a radiative cloud height, or centroid, of any importance to aviation? No. Cloud top is.).**

**Reply**:

The S5P L2 CLOUD OCRA/ROCINN product is not only meant to assist the atmospheric gas retrieval. OCRA/ROCINN is a sophisticated cloud retrieval algorithm which could be used for cloud studies too. The reviewer suggests the scientific topics like aerosol-cloud interactions, cloud trends etc. We agree with

him/her on the topics and we can write some text on this important aspect of the applicability of S5P CLOUD L2 data. In particular, the ROCINN_CAL retrieved properties should be more appropriate for cloud studies but also ROCINN_CRB should be used for scientific cloud topics. Moreover, there is an appropriate reference for relevant cloud studies which we can include: Loyola et al. 2010 (https://doi.org/10.1080/01431160903246741).

S5P FRESCO is primarily developed for cloud correction of trace gas retrievals. The secondary goal is to determine long-term cloud height trends from O2 A-band observations starting with the measurements by GOME in 1995. The importance of the O2 A-band is its independence of temperature as compared to thermal infrared cloud height measurements.

**Planned action**: Using elements of the above, we will introduce a paragraph on the wider scope of cloud studies.

*See page 4, line 26.*

*"We note that applications of the S5P cloud data are not limited to atmospheric composition measurements. As demonstrated by \citet{Loyola-2010aa} for GOME, OCRA/ROCINN can be successfully applied to study global and seasonal patterns and trends of cloud amount, cloud-top height, cloud-top albedo and cloud type, and compares well with the multisatellite international satellite cloud climatology project (ISCCP) D-series cloud climatology.*

*While developed primarily for cloud correction of trace gas retrievals, a secondary goal of S5P FRESCO is the determination of long-term cloud height trends by adding to the \chem{O_2} A-band observations that started with the measurements by GOME in 1995. The advantage over thermal infrared cloud height measurements is its independence of temperature."*

**(2.3) It looks like that the first comparison of TROPOMI cloud algorithms has been already reported in the TROPOMI S5P Science Verification Report (S5P-SVP). As far as I can judge comparing that outcome with that of this paper, the main conclusions are the same. But some methodological approaches deployed for the S5P-SVP, delivering valuable insights, have not been followed up here (e.g., across-track errors, surface errors as function of low cloud fractions, three-dimensional RT).**

**Reply:** Linking to the results of the S5P-SVP, and to other literature, is a good suggestion. But new analysis along these topics, which is often more about algorithmic sensitivity rather than validation, is considered largely out of scope. We discuss most of these points now shortly in a separate section, or we add information in the data set description. See below for the specific aspects.

**So, I invite the authors to elaborate and make explicit the following algorithmic aspects:**

**(2.4) - Errors arising from a plane-parallel approximation (neglection of 3D RT). We know that the improved spatial resolution does have an impact on RT. We are not talking about GOME-2 or SCIAMACHY anymore. Please also link your results to those of the MICRU algorithm when talking about cloud fraction.**

**Reply:** Any detailed study on the neglection of 3D RT, or the comparison with the MICRU, is considered out of scope. We note here that the studies in the S5P SVP was done for pre-launch versions of the TROPOMI cloud algorithms and there have been important changes since then.

**Planned action**: we shortly mention both topics, referring to the S5P-SVP and, for MICRU, to the recent publication of Sihler (AMTD, 2020).

**Reference**:

Sihler, H.; Beirle, S.; Dörner, S.; Gutenstein-Penning de Vries, M.; Hörmann, C.; Borger, C.; Warnach, S. & Wagner, T. MICRU background map and effective cloud fraction algorithms designed for UV/vis satellite instruments with large viewing angles Atmos. Meas. Tech. Discuss., 2020.

*See new section 2.4 'Note on previous assessments of OCRA/ROCINN and FRESCO algorithms'*

**(2.5) - Errors arising from the neglection of cloud multi-layeredness. Are the algorithms capable to flag this? Can a data user expect to be able to use TROPOMI data to investigate turbulent atmospheres? Joiner et al. (2010) shows that the fraction of multi-layer cloudy pixels can be up to 50% or more at OMI spatial resolution.**

**Reply**: No, multi-layeredness is not flagged. To our knowledge multilayer cloud information cannot be obtained from single view observations of the O2 A-band. Only in combination with another sensor like MODIS (Joiner et al., 2010) or from multidirectional O2 A-band observations like POLDER provides (Desmons et al., 2017), multilayer information could be obtained. Note that the impact of double-layered clouds on on OCRA/ROCINN retrievals was tested by Loyola et al. (2018).

**References**:
Desmons et al., A Global Multilayer Cloud Identification with POLDER/PARASOL, J Appl. Met. Clim., 2017, https://doi.org/10.1175/JAMC-D-16-0159.1

Loyola et al., (2018) is already in the list of references.

**Planned action**: we add a note on the neglect of multilayerdness, using the above references (Joiner 2020, Desmons 2017, Loyola 2018). Any further study on this topic is considered out of scope.

*See new section 2.4 'Note on previous assessments of OCRA/ROCINN and FRESCO algorithms'*

**(2.6) - Vertical inhomogeneity of clouds. To what extent are the presented algorithms capable to follow it? Can they be improved to encapsulate different vertical extinction profiles? Will the algorithms be able to reproduce cloud distributions inferred, e.g., from CloudSAT?**

**Reply:** This question is beyond the scope of this paper, since it addresses cloud retrieval algorithm concept and development but not validation. In principle, the FRESCO and CRB algorithms do not assume any profile of cloud extinction since the cloud model is a reflecting surface. The CAL model can in principle be extended to account for several vertical extinction profiles, but this is out-of-scope of this validation paper.

**Planned action**: We only mention the neglect of vertical inhomogeneity, together with the aspect of multilayeredness.

*See new section 2.4 'Note on previous assessments of OCRA/ROCINN and FRESCO algorithms'.*

**(2.7) Ziemke et al., 2009 show that average cloud extinction profiles for tropical deep convective clouds that peak at different pressures depending in general on the total optical thickness. This implies that the ISCCP diagrams from TROPOMI cannot be fully reliable because of the following remark:**
**- Are the CTH/CH retrievals dependent on COT? This is a matter of great concern for cloud research. This aspect is hastily mentioned by the authors only once, but it got my attention. this has to be read and understood in connection with bullet (2) by the first reviewer, which I support. Please, elaborate and make explicit.**

**Reply:** The CTH/CH is retrieved in the absorption peak, whereas the COT in the continuum of the O2 A-band. COT is certainly dependent on other variables (i.e., the radiometric cloud fraction and the surface albedo). From a former analysis of the information content in the Oxygen A-band (Schuessler et al. 2014), the simultaneous retrieval of CTH and COT, using the CAL model, was proved possible. In the same

reference, the errors for the simultaneous retrieval of four parameters (CF, CTH, COT and cloud geometrical thickness (CGT)) and three parameters (CTH, COT and CGT) have been assessed. It was found that even an additional third cloud parameter cannot be retrieved with accepted accuracy.

The simulations presented by Wang and Stammes (2014) show that for optically thin clouds the FRESCO cloud height could be situated inside or above the cloud, depending on the surface albedo. For thick clouds the cloud height is near the optical midlevel.

In conclusion, these works show that CTH/CH and COT are obtained largely from different spectral information, and that CTH/CH can show deviations for optically thin clouds but not for thick clouds.

**Planned action**:
(i) In the data set description of ROCINN, it was already noted that CH/CTH and CA/COT can be retrieved simultaneously. We add here now the Schuessler (2014) reference and make explicit that two independent pieces of information are available for this retrieval.

(ii) in the new section on previous assessments of the cloud retrieval algorithms, we refer now to the simulations of Wang and Stammes (2014).

*Summary of changes*

- *(i) Page 9, line 17 '…from two independent pieces of information (Schuessler et al., 2014).'*
- *(ii) Page 16, line 11 'In agreement with this, simulations with FRESCO (Wang and Stammes, 2014) have shown that for optically thick clouds, the cloud height is near the optical midlevel, while for optically thin clouds and higher surface albedo, a FRESCO cloud height above the cloud can be found.'*

**(2.8) - Surface influence. Looking at the S5P-SVP, Figures 13.28 and 13.29, pages 285- 286, there is a clear CH-dependency on surface reflectivity and cloud fraction. In the present paper only at P23, bias dependence on surface reflectivity is mentioned. So, I appreciate a similar exercise, where the accuracy of CH is subset after surface reflectivity and cloud fraction. It can be done within a Taylor diagram or by other means, but this interdependency must be made explicit.**

**Reply:** Indeed, the surface albedo effect on CH is important, especially for optically thin clouds, so with a low RCF. This is definitely an interesting topic. But this paper is about validation and not retrieval algorithm sensitivities; we consider this beyond the scope of this (already quite comprehensive) paper.  We will however refer to the S5P-SVP results.

**Planned action**: Refer to the S5P-SVP results in the new section on previous assessments of the cloud retrieval algorithms.

*See new section 2.4, page 16, line 9. 'As shown in the S5P-SVP (Richter and the Verification Team, 2015, section 13.4.2.3), cloud height comparisons between ROCINN_CAL or FRESCO with SACURA (Rozanov and Kokhanovsky, 2004) show larger disagreement at scenes with a low RCF and a higher surface albedo, indicating a larger uncertainty in these conditions.'*

**(2.9) - Across-track dependence of cloud retrievals. The authors are encouraged to compare their results with those of Fasnacht et al., 2019. Fasnacht et al., A geometry-dependent surface Lambertian-equivalent reflectivity product for UV-Vis retrievals – Part 2: Evaluation over open ocean, Atmos. Meas. Tech., 12, 6749–6769, 2019.**

**Reply**: Across-track dependence of cloud properties is indeed an interesting topic. Note that version 2.1.3 of OCRA/ROCINN includes the geometry-dependent G3_LER retrieval of surface albedo. In general, concerning across-track dependency, it should also be mentioned here that degradation of the L1b data has an impact. The current L1b data are not degradation corrected and it was found that degradation at the

swath edges is stronger than at nadir. This particularly affects the radiometric OCRA cloud fraction since it directly relies on absolute reflectance. Through the cloud fraction, across-track features originating from degradation may also translate to the other cloud parameters. A degradation correction will be included in the L1b data themselves in a future release.

**Planned action**: (i) We will include across-track plots of sRCF and cloud height. (ii) We will add a short note on the upcoming improvements relevant for across-track errors.

*Summary of changes*

- *(i) New Section 4.2 'Across-track dependence', page 20.*
- *(ii) Page 31, line 20 'With the update to CLOUD version 2, these OMI-based auxiliary data are replaced based on the TROPOMI data themselves and the effects listed below are largely reduced.'*

**Minor comments:**

**(2.10) P3 L18: "used by FRESCO". There should also be "... and ROCINN"?**

**Planned action:** This will be added.

*See page 13, line 12.*

**(2.11) P5 L5: "Note that at maximum RCF and CA, sRCF reaches 1.2 rather than 1." Do the authors have an explanation why a CF must exceed the limit of 1? Clearly the value is not physical. So, please, elaborate and make explicit that a CF=1.2 is needed as an ad-hoc correction for surface and/or trace gas retrieval.**

**Reply:** There are two intertwining things here.
(i) The introduction of a scaled RCF, sRCF=RCF*CA/0.8. This was introduced to better compare the OCRA and FRESCO cloud fractions. When both RCF=1 and CA=1, it follows that sRCF=1.25. (note that the 1.2 was a typo, this should be 1.25).
(ii) The radiometric cloud fraction of a scene viewed in a certain direction can be larger than one if the reflectance of the cloud is larger than one in that direction. This may occur for e.g., large viewing zenith angles and forward scattered light. Please note that this does not violate flux conservation since that holds for the average over all directions. Note however that for OCRA, the RCF is constraint not to exceed one.

**Planned action**: we explain this now better in the text, and correct the 1.2 typo.

*Summary of changes:*

- *(i) Page 10, line 5. 'Note that when both RCF and CA reach unity, sRCF reaches 1.25 rather than 1.'*
- *(ii) Page 11, line 34. 'Note that the RCF of a scene viewed in a certain direction (e.g., at large viewing zenith angle and forward scattered light) can exceed unity if the reflectance of the cloud is larger than unity in that direction. This does not violate flux conservation since that holds for the average over all directions.'*

**(2.12) P8 L28: "Due to the difference in overpass time between GOME-2 (in the morning) and Sentinel 5 precursor (in the afternoon)" Why is the overpass time a source of discrepancy for the surface albedo climatology? I can understand the difference in footprint size, but not a difference of some hours when building a climatology of an object barely evolving within few hours.**

**Reply:** The LER surface albedo climatology based on GOME-2 data implicitly includes the solar zenith angle, since it is a monthly climatology. Because the solar zenith angle is different for Sentinel-5P and GOME-2, the GOME-2 LER climatology can be a source of discrepancy when applied in retrievals for Sentinel-5P.

**Planned action**: see (2.13).

**(2.13) For the spatial resolution, it is not clear to me why the GOME-2 climatology is used and not the MERIS black-sky albedo climatology, which would be a much better choice. Please, elaborate, make explicit and justify.**

**Reply:** The MERIS black sky albedo climatology (Popp et al., AMT, 2011) is covering only the period 2002 – 2006, which is about 15 years ago. In the data base, snow regions contain many missing pixels. Therefore we consider the MERIS data base outdated, and is replaced by the recent GOME-2 climatology.

**Planned action**: We reformulate the text as follows:

*See page 11, line 21. "The FRESCO-S algorithm uses a surface albedo monthly climatology based on GOME-2 \citep{Tilstra-2017}. An important advantage over the MERIS black-sky albedo climatology (based on 2002-2006 data, i.e., about 15 years ago) \citep{Popp-2011aa} is that it is more recent. On the other hand, it is affected by the GOME-2 resolution and the solar zenith angle at overpass time. Due to the difference in overpass time…"*

**(2.14) P13 L2 and ff: Section 4.1.2 S5P FRESCO. I find this section unnecessary in the context of this manuscript. The purpose of the paper is to present a validation and comparison between different cloud products derived from TROPOMI measurements. However, in this section, the inability of FRESCO to discriminate aerosols from clouds is presented. This result, besides not being new (Wang et al, ACP, 2012), is not surprising given the spectral range used by the algorithm which handles cloud and aerosol radiances similarly.**

**Moreover, the authors swiftly interchange between clouds and aerosols in the narrative and this is inconsistent: line 9 should read "For this new product, the sensitivity to low ___aerosols___ in the low atmosphere is improved" and not "the sensitivity to ___low clouds___ is improved".**

**This is because based on the very same evidence provided by the authors themselves in the paper you are not retrieving clouds.**

**So wouldn't it be better to filter out all those pixels that are reasonably aerosol from the data set? I would like to stress that, although one of the possible applications of these data sets is the correction for trace gas retrievals, a cloud data set should serve cloud research too. What if a data user is going to average and assess long-term tendencies or climatology of cloud properties? How much of such missflagged aerosols will be present in the record?**

**Reply**: This case is very important for the paper because it shows the improvement of the FRESCO algorithm which will be implemented in the new version 1.4.0. The previous FRESCO version retrieves the surface height, which is wrong. The capability of the new version of FRESCO to retrieve aerosol and cloud layers close to the surface is essential for accurate NO2 retrievals from TROPOMI. In this case of strong air pollution in winter in China, aerosols and NO2 are strongly dependent. We will therefore keep this section. We will however review the text to correct apparent inconsistencies.

Any study on aerosol filtering or flagging is considered out of scope of this work.

**Planned action**: Review the paragraph, especially on cloud and aerosol aspects.

*The important point here is not that FRESCO cannot discriminate between aerosol and cloud, but that for low cloud and/or aerosols, a height equal to the surface height can be retrieved. So we change*

*Page 34, line 2. "An issue in S5P FRESCO 1.3 is that, at low radiometric cloud fraction, there is a tendency to retrieve a cloud height (or aerosol height, as the algorithm does not discriminate between aerosol and*

*cloud, (Wang et al., 2012)) equal to the surface altitude. Errors in the cloud (or aerosol) height can have an important impact on the retrieval of tropospheric NO2 columns by TROPOMI."*

**(2.15) Additionally, please collapse all FRESCO algorithmic details in one section, as pointed out also by the first referee.**

**Reply**: see our answer to reviewer #1 (1.2). As part of a general reorganization on product information and algorithmic details, FRESCO algorithmic details will be collapsed in one place.

**Planned action**: see (1.2).

**References**

**Joiner, J., Vasilkov, A. P., Bhartia, P. K., Wind, G., Platnick, S., and Menzel, W. P.: Detection of multi-layer and vertically-extended clouds using A-train sensors, Atmos. Meas. Tech., 3, 233–247, 2010.**

**Wang P, Tuinder ONE, Tilstra LG, De Graaf M, Stammes P (2012) Interpretation of FRESCO cloud retrievals in case of absorbing aerosol events. Atmos Chem Phys 12(19):9057–9077, doi:10.5194/acp-12-9057-2012**

**Sentinel-5P TROPOMI Science Verification Report, S5P-IUP-L2-ScVR-RP, Issue 2.1", Sect. 13.4-14.4, https://earth.esa.int/documents/247904/2474724/Sentinel-5P-TROPOMI-Science-Verification-Report, 2015**

**Ziemke, J. R., Joiner, J., Chandra, S., Bhartia, P. K., Vasilkov, A., Haffner, D. P., Yang,K., Schoeberl, M. R., Froidevaux, L., and Levelt, P. F.: Ozone mixing ratios inside tropical deep convective clouds from OMI satellite measurements, Atmos. Chem. Phys., 9, 573–583, 2009.**

**Fasnacht et al., A geometry-dependent surface Lambertian-equivalent reflectivity prod- uct for UV-Vis retrievals – Part 2: Evaluation over open ocean, Atmos. Meas. Tech., 12, 6749–6769, 2019.**

**Anonymous Referee #3

**(3.1) This paper compares different OCRA/ROCINN and FRESCO cloud products to several auxiliary satellite data from other sensors as well as ground based data. It furthermore describes the ongoing efforts to improve the reliability of both official product branches. The content of the paper is clearly in line with the topics of AMT. Publication, however, is only encouraged after the following comments and those detailed by the Referees #1 and #2 are considered.**

**Reply:** We are grateful for the careful review of our manuscript. Please find replies and planned actions to your comments below.

**Major comments:**

**(3.2) 1) Actually, I got the impression to read two merged papers. On the one hand, the actual operational algorithms are described and somehow verified – validation without knowing the actual truth may be exaggerating - and, on the other hand, significant problems of the algorithms are identified and possible fixes are presented. What I miss is the link: Why are the future algorithm versions that are proposed to fix many issues of the actual algorithms not included in the verification exercises? From the manuscript as is, I get the impression that this paper is actually an algorithm presentation of somehow improved algorithms within a paper extensively using outdated data.**

**In order to bridge this gap, I would like to suggest to options: a) Split the paper in two, one verifying the actual operational algorithms and another one (or two for both algorithm branches each) introducing the future versions of the algorithms – then with a verification as well. b) Treat both algorithm versions (actual and future) of both OCRA/ROCINN and FRESCO similarly in the verification section so that an actual user may judge for himself either to use what is already available or to wait for an algorithm update being applied in the future without knowing, when this will be.**

**Reply:** We agree that the presentation of older and newer versions of the cloud algorithms was confusing. As there has been no reprocessing yet, there are currently no long time series of the newer versions of OCRA/ROCINN and of FRESCO available. A verification/validation (in particular the comparison with CLOUDNET) on the same level of the older and newer versions (option b proposed by the referee) is therefore simply not possible. Instead, in the data set description, we make now more clearly the separation between the older product version and a short description of the changes in the newer product version. Furthermore, it is made clearer that the bulk of the analysis in the paper is for the older versions: CLOUD OCRA/ROCINN version 1 and FRESCO 1.3.2. The results in the old section 'Geographical patterns', which actually show improvements in the newer versions, is now put in a separate outlook section after the main results.

See new *Section 5 'Impact of processor version upgrades'*.

**(3.3) 2) I found the structure of manuscript quite confusing. Descriptions of the algorithms appear at several locations. I suggest to first introduce all data, then describe the conducted studies and finally discuss the results.**

**Reply:** We will now bundle the description of the algorithms in one section. See our answer to comment (1.2) of Referee#1.

**Planned action**: see Planned action to comment (1.2).

**Minor comments**

**(3.4) 1) The abstract contains many acronyms, which are not described at their first appearance, e.g. the difference between ROCINN_CRB and ROCINN_CAL are not clear from the beginning. I suggest to restructure the whole abstract. Maybe it is possible to collapse both ROCINN branches to one. Furthermore, the outlook on future mitigations (page 1, line 8) should be moved to the end the abstract as suggested in the guidelines.**

**Reply:** Thanks for the suggestions on restructuring.

**Planned action:** Restructure the abstract.
*Page 1, line 3. 'In this work we assess the quality of the cloud data from three Copernicus Sentinel-5 Precursor (S5P) TROPOMI cloud products: (i) S5P OCRA/ROCINN_CAL (Optical Cloud Recognition Algorithm/Retrieval of Cloud Information using Neural Networks;Clouds-As-Layers), (ii) S5P OCRA/ROCINN_CRB (Clouds-as-Reflecting Boundaries) and (iii) S5P FRESCO-S (Fast Retrieval Scheme for Clouds from Oxygen absorption bands - Sentinel). Target properties of this work are …'*

*'Peculiar geographical patterns are identified in the cloud products, and will be mitigated in future releases of the clouddata products.' Moved to end of the abstract.*

**2) Please add more references:**

**(3.5) page 2, line 6: for the trace gas products, please also add H2O**

H2O is added. *See page 3, line 15.*

**(3.6) page 2, line 16: Please add reference and name pollution**

**Planned action**: We rephrase as "…global measurements of atmospheric species related to air quality (NO2, SO2, CO, tropospheric O3, aerosols), …". As general reference we add 'www.tropomi.eu'.

*See page 3, line 25.*

**(3.7) page 2, line 28: both HICRU and MICRU are also using PMD data, please add references below**

References Grzegorski and Sihler added. *See page 4, line 6.*

**(3.8) page 3, line 7: Please add a reference to the FRESCO algorithm applying "a directional surface albedo". I could not find this feature described in the cited references.**

**Reply:** This is described in the just submitted paper by Tilstra et al., "Directionally dependent Lambertian-equivalent reflectivity (DLER) of the Earth's surface measured by the GOME-2 satellite instruments", submitted to AMT, 2020.

**Planned action:** Add above reference. *See page 4, line 22.*

**(3.9) Page 3, line 13: Ref to FRESCO-S**

**Reply:** FRESCO-S is a specific version of the FRESCO algorithm. The differences with respect to the FRESCO version described by Wang et al. (2008) have to do with the spectral band width selection and interpolation. Details are given in the TROPOMI NO2 ATBD (since FRESCO is part of the NO2 algorithm of TROPOMI) and in the Sentinel-5 Cloud ATBD.

References:

-TROPOMI ATBD for tropospheric and total NO2,  issue 1.4.0, 2019-02-06, S5P-KNMI-L2-0005-RP
-Sentinel-5 L2 Prototype Processor – Algorithm Theoretical Baseline Document for Cloud data product, ref: KNMI-ESA-S5L2PP-ATBD-005; issue: 3.0; date: 2018-12-15.

**Planned action:** Add above references.

*See page 10, line 17. 'FRESCO-S specific information can be found in the S5P NO2 ATBD (KNMI, 2019) and in the S5 CLOUD ATBD (KNMI, 2018).'*

**(3.10) page 4, line 10: Reference to OCRA**

**Reply:** Note that this section underwent extensive reorganization. Where still applicable, references to Loyola (1998), Lutz (2016), Loyola (2018) will be added.

**Planned action:** Add refs Loyola (1998), Lutz (2016).

**(3.11) page 7, line 17: Reference to OCRA**

**Reply:**.We will add refs to Loyola (1998) and Lutz (2016) (already present in the list).

**Planned action:** Add references. *See page 8, line 25.*

**(3.12) page 12, line 2: Are these "geographical patterns" also discussed in Lutz et al., 2016? Please add a reference to the S5P verification report.**

**Reply:** Geographical patterns for GOME-2 have indeed been discussed in Lutz et al. (2016) in terms of zonal and meridional means, sun-glint and scan-angle dependencies.

**Planned action**: We include references here to Lutz (2016) and to the S5P verification report.

*See page 31, line 16.*

**(3.13) 3) Please homogenize the acronyms and formulae. For example, O2 is sometimes written with subscript and sometimes not. Also TROPOMI/S5P is not consistently spelled. Sometimes its lower case and sometimes it is in caps.**

**Reply:** We apologize for the heterogeneity in naming conventions.

**Planned action**:

- Check everywhere: O2 in subscript.
- Consistently use 'TROPOMI/S5P', 'S5P', 'TROPOMI' in all caps.

**(3.14) 4) page 3, line 2: What about GEMS and TEMPO? In what respect is S4 the first of its kind?**

**Reply**: The point here was that S4 will be the first geostationary view *over Europe*, but this was not so clear.

**Planned action**: Remove 'of its kind', leaving, 'the first mission for a geostationary view of air quality over Europe.' *See page 4, line 17.*

**(3.15) 5) page 7, line 27: The choice of resolution seems a bit arbitrary. Please provide a rationale for both.**
**(3.16) Page 7, line 28: What is the rationale behind taking monthly mean reflectance data.**

**Reply:** We will explain both points (3.15, 3.16). See below:

**Planned action:** Include text blocks below for version 1 and 2 of the OCRA/ROCINN algorithm:

*See page 8, line 34. "For version 1 of the algorithm, the cloud-free background is based on three years of OMI data and consists of global monthly composite maps per color with a spatial resolution of 0.2 x 0.4 degrees. This relatively coarse and asymmetric spatial grid choice is due to the relatively large and asymmetric (especially near the swath edge) OMI pixels. Thanks to the monthly temporal resolution, seasonal changes can be covered. For each given day a linear interpolation between two adjacent monthly maps is used."*

*See page 10, line 8. "In the new S5P CLOUD version 2, the OMI-based cloud-free background maps have been replaced by maps based on TROPOMI data (currently 1 year) and the spatial resolution was increased to 0.1 x 0.1 degrees, while keeping the monthly temporal resolution."*

**(3.17) 6) Please include a description of the treatment of snow and ice surfaces in the description of the OCRA/ROCINN and FRESCO.**

**Reply:** For OCRA/ROCINN version 1, there is no special treatment for snow/ice pixels. But it is known that the retrieval is more challenging in such circumstances. Regarding FRESCO-S, the treatment of snow is described in the TROPOMI NO2 ATBD (see reference above).

**Planned action:** For FRESCO-S, we will refer the TROPOMI NO2 ATBD.

*See page 11, line 29. 'Treatment of snow/ice surfaces is described in the S5P NO2 ATBD (KNMI, 2019).'*

**(3.18) 7) page 9, line 3: "no official documentation is currently available for FRESCO-S" If this is true, I strongly would like to encourage the co-authors, that are developing this product, to provide official documentation to fill this gap for a potential user of the product.**

**Reply:** A journal paper on FRESCO-S is being planned. For the time being, the information is given in the two ATBDs mentioned above.

**Planned action:** None.

**(3.19) 8) page 16, line 9 and following: It seems to me that the statistics of the different data sets are based on different subsets (eg. MODIS versus the others). I strongly suggest to use the same subset to compute statistics in order to avoid biases.**

**Reply**: Indeed, different subsets are considered. It is true that it can make an important difference if the common subset is taken or not. However, constructing the common subset for all products in Fig. 4 is considered out of scope. But, as response to comment (2.9) we add now plots showing the across-dependence of sRCF and C(T)H of S5P OCRA/ROCINN_CAL, _CRB and S5P FRESCO. Here we show this for (i) the product-specific pixel screening and (ii) the common subset of S5P ROCINN_CRB and S5P FRESCO. Especially for the mean cloud height it is important if the common subset is taken or not.

**Planned action**: Add lines with the common subset of S5P ROCINN_CRB and S5P FRESCO in the across-track dependency plots (CH and sRCF).

*See figure 5.*

**(3.20) 9) page 17, line 27: Would it be possible to assess the effectiveness of this "compensation"?**

**Reply**: We include now a plot of the original S5P ROCINN_CAL COT, which can be compared to the effective COT (RCFxCOT) and the VIIRS COT. See also answer to comment (1.13).

**Planned action**: See (1.13).

**(3.21) 10) Figure 9: These plots are really not easy to perceive.**

**Reply:** The plots are indeed very busy as remarked by other reviewers.

**Planned action:** We will leave out the blue/white/red background on cloud phase, as it is not really essential to understand the results. Instead, we use a grey background indicating the vertically resolved cloud occurrence fraction.

**(3.22) A particular feature I would like to have discussed is that there seems to be a significant number of C(T)H=0 values for the S5P algorithms as opposed to the OMI data in the upper left plot, which does not show not a single zero reading. What can be the reason behind this behaviour?**

**(3.23) Furthermore, why are there significantly more CH0=0 values than CTH=0 values for ROCINN_CAL?**

**Reply on (3.22, 3.23):** The O2 A-band that S5P FRESCO 1.3 uses is less sensitive to low clouds than the O2-O2 absorption band that OMI OMCLDO2 uses, causing S5P FRESCO to predict a too low cloud height for low clouds, often equal to the surface altitude (see FRESCO data set description in the paper). Note that this situation is expected to improve for S5P FRESCO 1.4, using a wider spectral window with low to moderate absorption in the O2 A-band (as explained in the paper). Regarding S5P ROCINN_CRB and ROCINN_CAL, deeper investigations are needed to conclude under which particular situations these low retrievals happen, or why they are less prevalent for ROCINN_CAL.

**Planned action**: The following text will be inserted.

*Page 28, line 23. 'For S5P FRESCO, these low cloud height retrievals can be attributed to the low sensitivity of the selected window of the O2A band to low clouds (see Sect. 2.2.2; this will be improved for FRESCO 1.4 with the new window selection), while the O2-O2 band employed by OMI OMCLDO2 has a better sensitivity for low clouds (Acarretaet al., 2004). Regarding S5P ROCINN_CRB and ROCINN_CAL, deeper investigations are needed to conclude under which particular situations these low retrievals happen, or why they are less prevalent for ROCINN_CAL.'*

**(3.24) 11) The conclusions (page 25) start off with two statements what will be better in the future. Please move these statements to the end of the discussion (compare major comment 1).**

*Statements are moved to the last items of the bullet list, page 36.*

**Specific comments**

**(3.25) page 1, line 7 "were" → "are"**

Done.

**(3.26) page 2, line 33 omit "easy"**

**Reply**: We prefer to keep this wording.

**Planned action:** None.

**(3.27) page 3, line 4 omit "fast"**

**Reply**: We prefer to keep this wording.

**Planned action:** None.

**(3.28) page 3, line 30: NPP-VIIRS etc. are instruments, not satellites**

Replace 'satellites' with 'instruments'. *See page 5, line 2. '…from other instruments:…'*

**(3.29) caption of figure 1: omit ", while S5P FRESCO is merely a backup"**

**Reply:** We consider this relevant information and therefore prefer to keep it.

**Planned action:** None.

**(3.30) page 4, line3: omit "finally"**

"finally" removed.

**(3.31) page 4, line 5: please rephrase this sentence**

**Reply**: Ok.

**Planned action:** We replace the original sentence with

*Page 6, line 5. "In this work several cloud data products and cloud properties are discussed; here we provide an overview and terminology conventions."*

**(3.32) table 1: the superscripts are appearing at an odd order: b, d, a, c → please sort please also add f_{rc,0.8} to that table**

**Planned action**: Entries are ordered. Instead of f_{rc,a} we now write f_{rc,0.8} as that is the only case we consider.

*See Table 1.*

**(3.33) page 5, line 4: A verb is missing somewhere after "In general"**

**Planned action**: We will insert 'one has'. Note that 'In general' is replaced with 'In most cases'.

*Page 7, line 4. 'In most cases one has RCF … '*

**(3.34) page 5, line 6 and following: please add a short comment on the rationale behind 0.8 as the fixed cloud albedo**

**Reply:** The choice of 0.8 as fixed cloud albedo for satellite spectrometers with large pixels (much larger than the size of clouds) is done to have an optimal cloud correction for trace gases like ozone (Koelemeijer et al., 2001, Stammes et al. 2008).

**Planned action**: Add comment including above reference.

*Page 11, line 8. 'See Koelemijer et al. 2001, Stammes et al. 2008 for the justification.'*

**(3.35) page 5, line 16: please use metres to avoid confusion: 7668m.**

Replaced.

**(3.36) table 2: please sort superscripts. please add column with references**

Done.

**(3.37) page 6, line 6: omit "among else"**

**Reply:** In the revised version of the manuscript this sentence no longer appears.

**(3.38) page 6, line 8: omit the entire sentence**

**Reply:** Agreed, as the relevant section appears right after.

**Planned action**: Remove sentence.

**(3.39) page 7, line 1: 20% bias on cloud fraction is a lot, please discuss the possible influences on the validation exercises in the paper**

**Reply**: A single number as requirement is of course a simplification. The impact of cloud fraction errors depends on the cloud height and on the application. (E.g., cloud correction of trace gases which may or may not be well-mixed, cloud slicing to obtain tropospheric ozone).

**Planned action**: A short note on requirements using the above will be inserted.

*Page 17, line 19. 'It should be noted that single numbers as requirement are necessarily a simplification. The impact of cloud parameter errors depends on the cloud height and on the application (e.g., cloud correction of trace gases which may or may not be well-mixed, cloud slicing to obtain tropospheric ozone,...).'*

**(3.40) page 8, line 2: What is a significant set? How many did you use? Please omit "smart"**

**Reply:** The term "smart sampling" is explained in the reference and the word 'smart' cannot be omitted. The sample size for training was ~200000 samples.

**Planned action**: rephrased as "A significant set (~200000 samples) of simulated radiances…".

*See page 9, line 12.*

**(3.41) page 8, line 8: "spherical particles" → "spheres"**

**Reply:** In our judgment, 'spherical particles' is more appropriate here. A cloud consists of particles, which in this model are assumed to be spherical.

**Planned action**: None.

**(3.42) page 8, line 12: What is a "GE_LER"? Please specify.**

**Reply: "**geometry-dependent effective Lambertian equivalent reflectivity"

**Planned action**: this will be spelled out in the text.

*See page 10, line 13.*

**(3.43) Page 8, line 14: "RPRO" is not introduced yet.**

**Planned action:** insert "The resulting reprocessed ('RPRO' in the file name) files…"

*See page 8, line 11.*

**(3.44) page 10, line 2: Please rephrase so that it easier to perceive, that there are two MODIS instruments of different platforms.**

**(3.45) Page 10, line3: omit "from north to south" and "from south to north" as this is redundant information**

**Reply:** We will rephrase the sentence (3.43), and omit the redundancies (3.44).

**Planned action**: Following sentence will be used: "There is a Moderate Resolution Imaging Spectroradiometer (MODIS) instrument on board of both the Terra and the Aqua satellites, with Terra in descending mode passing the equator in the morning and Aqua in ascending mode passing the equator in the afternoon, respectively."

*See page 12, line 30.*

**(3.46) page 10, line 12: omit "Dutch-Finnish"**

**(3.47) page 10, line 13: omit "NASA's"**

**Reply:** We prefer to keep both specifications (3.45, 3.46).

**Planned action**: None.

**(3.48) page 10, line 29: please specify what a "pixel" denotes in the context of a ground based measurement in order to avoid confusion with a "satellite pixel"**

**Reply**: See our answer to 1.11. We will use the term "altitude bin" instead.

**Planned action**: see (1.11).

**(3.49) page 11, line 6: "much less" → please be more specific**

**Reply**: The physical horizontal extent depends on which lidar was used, but for ceilometers it is 1-2 m at 1 km altitude, and 10-20 m at 10 km altitude. Some other systems may even stay below 2 m at 10 km altitude.

**Planned action**: We will reformulate the sentence as follows:

*See page 14, line 15. 'The physical horizontal extent of the cloud radar measurements is on the order of 20 m at 1 km altitude, and 200 m at 10 km altitude; for the lidar, the physical horizontal extent of the measurements is about an order of magnitude smaller. '*

**(3.50) page 11, line 14: Why is there a shift in tenses? "were" → "are"**

This sentence is no longer present in the current version of the manuscript.

**(3.51) figure 3: Please indicate the "sharper contrast" in the figures**

**Reply:** thanks for the suggestion.

**Planned action**: We will indicate on the plot regions where version 1 has clearly a sharper contrast.

*See figure 13.*

**(3.52) page 13, line 12 ad following: Please also discuss here, that aerosols may as well have a different impact on the RT than clouds.**

**Reply**: For scattering aerosols with little absorption the improvement of FRESCO algorithm is expected to have the same effect as for clouds: the raising aerosol layer height will improve the NO2 column. For (strongly) absorbing aerosols the radiative transfer is more complicated and its effect on NO2 retrievals has to be analysed separately (see e.g. Chimot et al., 2019).

Chimot, J., Veefkind, J. P., de Haan, J. F., Stammes, P., and Levelt, P. F.: Minimizing aerosol effects on the OMI tropospheric NO2 retrieval – An improved use of the 477 nm O2 – O2 band and an estimation of the aerosol correction uncertainty, Atmos. Meas. Tech., 12, 491–516, https://doi.org/10.5194/amt-12-491-2019, 2019.

**Planned action**: Add sentence based on the above.

*Page 34, line 17. "Note that for scattering aerosols with little absorption the improvement of FRESCO algorithm is expected to have the same effect as for clouds: the raising aerosol layer height will improve the NO2 column. For (strongly) absorbing aerosols the radiative transfer is more complicated and its effect on NO2 retrievals has to be analysed separately (see e.g., Chimot et al., 2019)."*

**(3.53) Figure 4: Please also discuss why there are steps (depending on the row) in the lower right figure. Is this an artifact/interference or a signal? This is critical, because right now it seems as an error in the proposed FRESCO-A wide algorithm.**

**Reply:** The steps are an artefact, and have to do with the spectral smile effect of the TROPOMI 2D-spectrometer.

**Planned action**: We will add a short note in the text or caption.

*Page 34, line 14. 'The steps in this figure are an artefact caused by the spectral smile effet of the TROPOMI 2D-spectrometer.'*

**(3.54) Page 15, section 4.2: Which version of OCRA/ROCINN is applied? Is this version 1 or the proposed future version? If it is version 1, please add a similar figure for version 2 (see major comment 1)**

**Reply:** OCRA/ROCINN version 1 is applied here. A comparison with version 2 is out of scope of the current manuscript.

*Version is indicated page 17, line 25; caption fig. 3, caption fig. 4, and other places.*

**(3.55) page 17, line 10: omit "as it was stated earlier" → So why state it again?**

**Reply:** We will simplify the sentences here, thereby omitting 'as it was stated earlier'.

**Planned action**: Replace 'As it was stated earlier […] (Siddans, 2016).' by

*Page 20, line 21. 'To enable a pixel-by-pixel comparison, the original 750 m NASA VIIRS pixels have been regridded to the TROPOMI footprints as explained by the S5P-NPP Cloud Processor ATBD (Siddans,2016).'*

**(3.56) Figure 7: If this is regridded VIIRS data, I would like to suggest to also show a scatter plot (2D histogram) in order to support a more quantitative comparison.**

**Reply**: In this Figure, we only show a comparison of the CTH from TROPOMI and VIIRS (regridded data) over a limited scene from an orbit. It does not refer to the complete dataset that it is used for the comparisons between VIIRS/TROPOMI. Therefore, probably a scatter plot might not be so useful for such a limited number of data. Furthermore, more analysis between VIIRS and TROPOMI is considered out of scope of the current manuscript.

**Planned action**: None.

**(3.57) Figure 8: I guess "TROPOMI" indicates OCRA/ROCINN. Please be more specific in order to avoid confusion with FRESCO. How would these plots look for version 2?**

**Reply:** thanks, this was an accidental omission. It is indeed OCRA/ROCINN_CAL. A comparison with TROPOMI version 2 is left for future work.

**Planned action**: Add OCRA/ROCINN_CAL in the caption.
*See Figure 7 revised manuscript.*

**(3.58) page 20, line 12: Please provide a rationale, why also OMI data are included in this study.**

**Reply:** OMCLDO2 is a well-established product, which has been compared to CLOUDNET before. We want to verify to which extent the new TROPOMI cloud products differ with OMCLDO2 in their relation with CLOUDNET data.

**Planned action**: Explain better rationale. The rationale was already present further down in the paragraph. We now rearrange.

*P 26, line 4. "In this section we discuss the comparison of S5P OCRA/ROCINN_CAL CTH, S5P OCRA/ROCINN_CRB CH and S5P FRESCO CH with ground-based CLOUDNET data. Moreover, we compare also OMI OMCLDO2 with CLOUDNET, using the same methodology, as this allows to make the connection with the work of Veefkind et al. (2016). By comparing the S5P products with the CLOUDNET data on one*

*hand, and OMCLDO2 with CLOUDNET data on the other hand, one learns better how the effective cloud heights of these different products relate to the (vertically re-solved) lidar/radar cloud observations of CLOUDNET, and where they are different."*

**(3.59) Figure 11: Please improve image quality.**

**Reply:** The text has indeed a low resolution.

**Planned action:** Figure will be remade in eps format.

*See Figure 10 in revised manuscript.*

**(3.60) Page 25, line 23: Please be more quantitative.**

**Reply**: These dependencies are largely removed in version 2 and we don't plan to make further quantifications for version 1.

**Planned action**: None.

**References**

Grzegorski, M., Wenig, M., Platt, U., Stammes, P., Fournier, N., and Wagner, T.: The Heidelberg iterative cloud retrieval utilities (HICRU) and its application to GOME data, Atmos. Chem. Phys., 6, 4461–4476, https://doi.org/10.5194/acp-6-4461-2006, 2006.

Lutz, R., Loyola, D., Gimeno García, S., and Romahn, F.: OCRA radiometric cloud fractions for GOME-2 on MetOp-A/B, Atmos. Meas. Tech., 9, 2357–2379, https://doi.org/10.5194/amt-9-2357-2016, 2016.

Sihler, H., Beirle, S., Dörner, S., Gutenstein-Penning de Vries, M., Hörmann, C., Borger, C., Warnach, S., and Wagner, T.: MICRU background map and effective cloud fraction algorithms designed for UV/vis satellite instruments with large viewing angles, Atmos. Meas. Tech. Discuss., https://doi.org/10.5194/amt-2020-182, in review, 2020.

**Anonymous Referee #2

**The third reviewer touches on two aspects that in my comments have probably remained implicit and have not been sufficiently highlighted.**

**(2.16) The first aspect is the accuracy of retrieval for very reflective surfaces. My request to subdivide and categorize biases in function of cloud fraction and surface reflectivity aims to understand two things: (1) what happens for very low CF (2) what happens for very high SA (3) where the algorithms start diverging in performance.**

**Reply:** As noted in (2.8), we consider such an extra analysis beyond the scope of this paper.

**(2.17) The second is the compactness and clarity of the product naming, having in mind the usability of the data for the typical user. I would appreciate in the conclusions clear and concise guidance on which products to use, for which purposes, and which not. I think the authors and the production teams should make an effort in this direction.**

**Reply**: The product names build on a long heritage and cannot be changed.

**Planned action**: Include short paragraph on guidance.

*Page 37, line 1. Typical applications of the TROPOMI cloud products are in the context of cloud impact on atmospheric composition measurement, such as masking of a measurement scene, accounting for modification in radiative transfer (e.g., the air mass factor) or cloud slicing (e.g., to estimate the tropospheric component of ozone). The study of seasonal patterns and trends is another potential application (Loyola, 2010).*

**(2.18) A reason for further uncertainty is the announcement of new products that would emerge from minimal adjustments of the algorithm. I do not think it is necessary to deem FRESCO-S a new product, only if you change the spectral range within O2A by a few nm. This is just a source of confusion for the reader and certainly cannot be considered a milestone for a typical algorithm development chain.**

**Reply:** The name FRESCO-S is not due to a change of the wavelength range but for a more fundamental change. In previous FRESCO versions - in particular those for GOME-2, the observed wavelengths were interpolated to the wavelengths of the LUT. On the Sentinel missions this is no longer possible due to changes in the spectral resolution and the 2D detector, with a moving wavelength grid. Therefore, in FRESCO-S, spectral resolution of the reflectance database was increased to allow for interpolation of the database to the wavelengths of the observation. This information was already present in the text, but we make now more clear that this is peculiar to FRESCO-S.

**Planned action**:

*Page 11, line 16. "The spectral resolution of the reflectance database was increased to allow for interpolation of the database to the wavelengths of the observation. This is in marked contrast to previous FRESCO versions (for the instruments GOME, SCIAMACHY, GOME-2), where the observed wavelengths were interpolated to the wavelengths of the database."*